# Landscape structures regulate the contrasting response of recession along rainfall amounts

Jun-Yi Lee[1,2], Ci-Jian Yang[2,3], Tsung-Ren Peng[1], Tsung-Yu Lee[4], Jr-Chuan Huang[2]

[1]Department of Soil and Environmental Sciences, National Chung Hsing University, Taichung 402202, Taiwan
5   [2]Department of Geography, National Taiwan University, Taipei 106319, Taiwan
[3]German Research Centre for Geosciences (GFZ), Telegrafenberg, Potsdam 14473, Germany.
[4]Department of Geography, National Taiwan Normal University, Taipei 106209, Taiwan

*Correspondence to*: Jr-Chuan Huang(riverhuang@ntu.edu.tw)

**Abstract.**

10   Streamflow recession, shaped by hydrological processes, runoff dynamics, and catchment storage, is heavily influenced by landscape structure and rainstorm characteristics. However, our understanding of how recession relates to landscape structure and rainstorm characteristics remains inconsistent, with limited research examining their combined impact. This study examines this interplay in shaping recession responses, upon 291 sets of recession parameters obtained through the decorrelation process. The data originates from 19 subtropical mountainous rivers and covers events with a wide spectrum of 15   rainfall amounts. Key findings indicate that the recession coefficient (*a*) increases while the exponent (*b*) decreases with the *L/G* ratio (the median of ratios between flow-path length and gradient), suggesting that longer and gentler hillslopes facilitate flow accumulation and aquifer connectivity, ultimately reducing nonlinearity. Additionally, in large catchments, the exponent (*b*) increases with increasing rainfall due to greater landscape heterogeneity. Conversely, in small catchments, it declines with rainfall, indicating that these catchments have less landscape heterogeneity and thus reduced runoff heterogeneity. Our findings 20   underscore the necessity for further validation of how L/G and drainage area regulate recession responses to varying rainfall levels across diverse regions.

## 1 Introduction

Streamflow recession, the falling segment of a hydrograph, represents the rainfall-runoff process and interactions among different flow paths and aquifers during a rainstorm. Therefore, the streamflow recession and its link with flow paths within 25   the landscape and aquifers, is particularly critical for baseflow estimation (Palmroth et al., 2010). A power-law relationship, $-dQ/dt = âQ^b$, between the rate of change in streamflow and streamflow rate $Q$ is widely used to describe the recession at the catchment scale (e.g. Brutsaert and Nieber, 1977). The recession parameters $â$ and $b$ arise from the geometric and hydraulic properties of the aquifer system. The recession coefficient, $â$, is tangled with the unit of streamflow and exponent $b$, which represents the nonlinearity of storage and is the slope of the regression line of $\log(-dQ/dt)$ vs $\log(Q)$ (see section 2.2.2). 30     Since aquifers in various landscape units (e.g., hillslopes, riparian areas, streams, etc.) exhibit different hydraulic properties, theoretical works have shown that the streamflow recession parameters depend on the landscape structure or aquifer properties.

In general, coefficient $\hat{a}$ shows a positive correlation with stream length and aquifer slope (Rupp and Selker, 2006), while it exhibits a negative correlation with drainage area, aquifer depth, aquifer heterogeneity (Rupp and Selker, 2006), and inter-hillslope heterogeneity (Harman et al., 2009). On the other hand, exponent $b$ tends to increase with the number of streams

(Biswal and Marani, 2010), aquifer heterogeneity (Rupp and Selker, 2006), and inter-hillslope heterogeneity (Harman et al., 2009), whereas it decreases with the total stream length (Biswal and Marani, 2010).

Additionally, theoretical studies have demonstrated that streamflow recession parameters are subject not only to the influences of landscape and aquifer systems but also to the interplay with antecedent storage and rainfall events. For example, coefficient $\hat{a}$ is negatively correlated with the recharge rate (Harman et al., 2009), the streamflow rate (Biswal and Nagesh

Kumar, 2014), and the initial groundwater table under unsaturated conditions, while it has a slightly positive correlation under saturated conditions (Rupp and Selker, 2006). The exponent $b$ slightly increases with a wet antecedent condition (Harman et al., 2009). However, drainage network theory indicates that $b$ increases with peak flow while the downstream channel receives more subsurface flow contribution but decreases with peak flow as the downstream channel receives less (Biswal and Nagesh Kumar, 2013). The inconsistent responses in $\hat{a}$ and $b$ among theories indicate a complicated interaction between landscape

structure and rainstorms during recession, implying that the recession mechanics in different regions need more exploration.

Tables 1 and S1 respectively summarize and compile previous empirical recession studies and two main takeaways are addressed: 1) The responses of $\hat{a}$ and $b$ to landscape features and structure are inconsistent. Such inconsistent results might be landscape dependent (e.g. different regional conditions). 2) These inconsistent recession responses might be due to different analysis methods. Most previous studies aggregated long-term data to a point-cloud, a collection of multiple recession curves,

to retrieve representative recession parameters, while some recent studies retrieved parameters from individual events to elucidate the temporal variability of recession. Fewer studies simultaneously addressed recession responses to landscape structure and distinct rainstorm events. For example, Biswal and Nagesh Kumar (2013) found that the structure of drainage networks might result in contrasting directions in the response of $b$ to peak flow. However, they did not specifically identify which landscape characteristics would predominantly influence the directional switch in the response of parameter b to rainfall.

Everything considered, the theory behind streamflow recession is still developing, and it is clear that we need a better understanding of how landscape structure and rainstorm characteristics affect streamflow recession, especially with the necessity of regional recession assessments under climate change. Thus, this study derived the recession coefficient and exponent in 19 mountainous catchments across Taiwan with multiyear records of hourly streamflow (291 events in total). These catchments, with drainage areas of 77–2,089 km$^2$, are characterized by steep, fractured, forested mountains and periodic

typhoon invasions. As a result of these characteristics, Taiwan's rivers have short water residence time and limited water retention capacity (Lee et al., 2020). We addressed three research questions: (1) What are the recession characteristics of typhoon events in small mountainous catchments? (2) How do landscape and rainstorm variables affect recession parameters in different regions? (3) In what way do landscape variables regulate the response of recession parameters to rainfall? In this study, we document the spatial patterns of recession parameters in Taiwan (Sect. 3) and then discuss how recession behaviors

change in different landscape settings (Sect. 4).

## 2 Material and methods

### 2.1 Study area

Taiwan is a mountainous island geographically located at the juncture between the Eurasian and Philippine tectonic plates and climatologically located in the corridor of typhoons. An active mountain belt with frequent typhoons shapes a steep and fractured landscape with verdant forests. The mean annual rainfall is about 2,510 mm, and approx. 40% of annual rainfall is brought by typhoons within a few days. The lowest mean annual temperature is approx. 4°C in montane regions and 22°C in the coastal plains. The mountains of Taiwan reach an elevation of 4,000 m within a short horizontal distance (~75 km) from the coast, creating a steep terrain (Huang et al., 2016). Specifically, the drainage area of most catchments is smaller than ~500 km², and stream lengths are less than ~55 km. The basic catchment descriptions, including landscape variables, can be found in Table S2. Land cover inventories from the Taiwan Ministry of the Interior (www.moi.gov.tw) were reclassified into three major categories, namely water ($C_W$), forest ($C_F$), agriculture ($C_A$), and others. The landscape metric was retrieved from the digital elevation model (DEM) with 20m resolution: $A$ is the drainage area [km²]; $DD$ is the drainage density [km km⁻²], defined as the ratio of total stream length to drainage area; $S_m$ is the gradient of the main stem [%]; $HI$ is the hypsometric integral [-]; $ELO$ is the basin elongation [-], defined as the ratio of the diameter of the circle with the same area as the basin to basin length.

In addition to the primary landscape variables described above, we incorporated flow path-associated variables into our study, as flow path is an explicit proxy for aquifer systems. Within a gridded DEM, the flow path is defined as the route followed by water from a grid cell, following the surface flow direction towards the channel cell (see detail in Tetzlaff et al., 2009). Specifically, flow path length ($l_{fp}$) is the route length from a cell to the nearest channel cell, flow path height ($h_{fp}$) is the elevation difference between the specific cell to the nearest channel cell, and flow path gradient ($g_{fp}$) is calculated as flow path height divided by flow path length. Each cell possesses its own value of $l_{fp}$, $h_{fp}$, $g_{fp}$, and $l_{fp}/g_{fp}$. Since the velocity of gravity-driven flow is typically proportional to the gradient ($v = l_{fp}/T \sim h_{fp}/l_{fp}$), this implies that the time ($T$) is proportional to $l_{fp}/g_{fp}$: $T \sim l_{fp}^2/h_{fp} = l_{fp}/g_{fp}$. Consequently, $l_{fp}/g_{fp}$ could serve as a potential proxy for residence time. Within a catchment, the medians of the $l_{fp}$, $h_{fp}$, and $g_{fp}$ distributions ($L$, $H$, and $G$) served as representative flow path characteristics. Please be aware that we defined the median of $l_{fp}/g_{fp}$ ($L/G$) in our study, differing from McGuire et al.'s (2005) previous study, which directly used the median $l_{fp}$ divided by the median $g_{fp}$. In the hydrological context, $l_{fp}/g_{fp}$ represents the residence time of each flow path, while the median of $l_{fp}/g_{fp}$ reflects catchment-wide residence time. The detailed definition and calculation of the flow path associated variables are illustrated in Table S3 in the supplement.

Streamflow in this steep mountainous island descends quickly after a typhoon invasion. Thus, hourly streamflow records are required to describe the entire streamflow recession since it only lasts a few days after the peak. This study collected hourly streamflow records during 1986-2014 from the Taiwan Water Resource Agency (www.wra.gov.tw) and Tai-Power Company (www.taipower.com.tw). Only the catchments without large water division infrastructures in the upstream area and with total rainfall greater than 30 mm were used to avoid human-manipulated streamflow data. Based on these criteria, nineteen catchments and 291 events were included for further recession analysis. Commensurate with the hourly streamflow, the hourly rainfall dataset from the Taiwan Central Weather Bureau (www.cwb.gov.tw) was collected, and the Thiessen weighted method

was used to estimate areal rainfall in the corresponding catchments. The cumulative rainfall window was defined as the elapsed time from 6 h before the rising flow to the peak flow. We do not consider rainfall amount after peak flow because there is typically less rainfall occurring after the peak flow. Hydroclimate metrics of rainstorm and streamflow, including total precipitation ($P$), duration ($D$), average precipitation intensity ($I_{avg}$), total streamflow ($Q_{tot}$), peak flow ($Q_p$), antecedent streamflow ($Q_{ant}$), and runoff coefficient ($Q_{tot}/P$), were extracted from these datasets (Table S3 and S4).

## 2.2 Recession analysis

The storage-outflow relationship is typically described by a power law if treating the catchment as a black box. The representative storage is, in fact, composed of many aquifers and thus exhibits a non-linear relationship:

$$Q = mS^n$$

(1)

where $S$ is the storage volume within a catchment (in units of volume [L$^3$] or depth [L]), $Q$ is the rate of streamflow ([L$^3$/T] or [L/T]), and $m$ and $n$ are constants (Vogel and Kroll, 1992). Since $S$ is difficult to directly measure, the relationship between the rate of streamflow decline and streamflow could be derived to represent the recession behavior (Brutsaert and Nieber, 1977) in Eq. (2).

$$-\frac{dQ}{dt} = nm^{\frac{1}{n}}Q^{\frac{2n-1}{n}} = \hat{a}Q^b$$

(2)

where $\hat{a}$ is the recession rate and $b$ represents the nonlinearity of storage, which is also the slope of the regression line in the plot of log($-dQ/dt$) vs log($Q$) (the recession plot). Both parameters can be estimated via different assumptions and fitting techniques. Notably, since nonlinearity is dimensionless, $\hat{a}$ is inherently strongly dependent on the units of $Q$ and $b$ via fitting (see details in section 2.2.2). Although the recession plot enables the analysis of streamflow recession and facilitates the

120 derivation of the storage–outflow relationship (Stölzle et al., 2013), the methods of recession segment extraction affect parameter estimation. For example, Stölzle et al. (2013) compared three extraction methods in conjunction with their corresponding parameter estimations. They found that recession characteristics, like recession time ($1/\hat{a}$), varied over 1–2 orders of magnitude, yet nonlinearity, $b$, varied rather narrowly. Their results suggested that the recession characteristics derived from different procedures have only limited comparability. Further, Dralle et al. (2017) found that the relationship

between $\hat{a}$ and antecedent wetness was sensitive to the number of data points and thus the extraction method. Despite the estimated parameters being inconsistent among the procedures, applying the same procedure is still a feasible way to capture the recession responses in a region.

### 2.2.1 Recession segment extraction

In the extraction procedure, two concerns should be addressed: (1) distinguishing between the early and late recession stages,

and (2) eliminating any unexpectedly positive increases in the recession. The early stage (containing pre-storm and surface flow) and the late stage of recession (dominated only by base flow) are indistinguishable and usually determined subjectively.

Some studies have empirically excluded the early-stage recession to eliminate the influence of quick flow (e.g., Brutsaert, 2008; Vogel and Kroll, 1992). Others used a threshold for the minimum length of extraction procedures, which ranged from 2 to 10 days (e.g., Mendoza et al., 2003; Vogel and Kroll, 1992). For eliminating unexpected positive increases during recession, several approaches have been proposed as well, for example, smoothing the hydrograph (Vogel and Kroll, 1992), discarding the segment entirely (Brutsaert, 2008; Kirchner, 2009), and breaking-and-rejoining the recession segments (Millares et al., 2009). Each strategy has its advantages and disadvantages; smoothing the hydrograph may not completely erase the bulges caused by precipitation and discarding the segment loses parts of recession events. Although breaking-and-rejoining the recession, too, disturbs the original streamflow records, the method which maintains the more complete recession event is preferable here.

For the recession segment extraction, first, the recession evolution caused by rainstorms was a main concern, and thus we selected the whole recession segment from the peak flow of all individual rainstorm. The whole recession segment represents the interactive mixing of quick and base flow. Second, we screened and broke down the hydrograph where abrupt bulges emerged, erased positive streamflow increases, and concatenated the remaining segments. This elimination procedure produces a curve quite similar to the master recession curve on a long-term scale (Millares et al., 2009). Third, data points corresponding to extremely low streamflow ($Q < 0.1$ mm h$^{-1}$) or recession ($-dQ/dt < 0.01$ mm h$^{-2}$), being likely affected by the limits of streamflow measurement, were excluded. Forth, rainfall events with an unreasonable ratio of total flow to total rainfall ($Q/P > 1.1$ or $Q/P < 0.1$) were also excluded to guarantee the data quality. Ultimately, a total of 298 rainstorms were selected for further parameter estimation.

**2.2.2 Parameter fitting**

In recession analysis, several fitting methods have been proposed. One approach involves fitting the lower envelope of a collection of multiple recession curves, which is referred to as point-cloud (Brutsaert and Nieber, 1977). Taking the lower envelope can prevent evapotranspiration effect which leads to higher values of $-dQ/dt$. Another is to fit with the entire point-cloud (Brutsaert, 2005; Vogel and Kroll, 1992) as subsoil heterogeneity may overshadow the evapotranspiration effect in larger or steeper catchments (Brutsaert, 2005). Yet another is to fit with the binned means weighted by the square of the standard error of each binned mean (Kirchner, 2009) because the lower values of $-dQ/dt$ could be affected by the measurement errors in the streamflow observations. Recently, a virtual experiment study (Jachens et al., 2020) suggested fitting with individual recession segments to explore the recession responses to individual rainstorms.

The parameter estimation from the retrieved recession segments is described below. Firstly, we corrected low-flow records: The same low flow levels appear frequently in late recession due to the detection limit of instruments, resulting in a series of zero $-dQ/dt$ values that affect parameter estimation, particularly for $b$. To reduce this bias, we applied the exponential time step method (Roques et al., 2017) in which the time step of the moving window for calculating $-dQ/dt$ exponentially increases along the recession. This extended sampling period helps avoid the occurrence of zero $-dQ/dt$ values (Roques et al., 2017).

An important concern in recession parameter estimation is the dependence between $\hat{a}$ and $b$, which confounds the interpretation of parameters (Dralle et al., 2015). The decorrelation method assumes that the observed flow, $Q$, consists of a

scale-free flow $\hat{Q}$ and a constant $k$ ($Q = k\hat{Q}$). Thus, the power law formula can be rewritten as $-dQ/dt = ak^{b-1}\hat{Q}^b$, where $a$ is the scale-free recession coefficient [$h^{-1}$]. For correcting $\hat{a}$ to $a$, the observed flow $Q$ was divided by a constant $Q_0$ (which is ideally equal to $1/k$, see detail in Dralle et al., 2015):

$$Q_0 = \exp\left(-\frac{\sum_{i=1}^{N}(b_i - \bar{b})(\log(\hat{a}_i) - \overline{\log(\hat{a}_i)})}{\sum_{i=1}^{N}(b_i - \bar{b})^2}\right)$$

(3)

where $\bar{b}$ and $\overline{log(\hat{a})}$ is the means of the fitted parameters $b$ {$b_1$, $b_2$, …, $b_N$} and $\log(\hat{a})$ {$\log(\hat{a}_1)$, $\log(\hat{a}_2)$, …, $\log(\hat{a}_N)$}, respectively, across $N$ rainfall events in a given catchment. Although the decorrelation method can reduce the unit effect and dependency on $b$, Biswal (2021) argued that the dependency of $\hat{a}$ and $b$ can't be fully decoupled, and retrieving parameters from the power law and fixing $b$ is preferable. Obviously, decoupling the dependency of $\hat{a}$ and $b$ in recession is unsolved and challenging and necessitates further study. Nevertheless, after the decorrelation process, the number of catchments with a high correlation between $a$ and $b$ ($R^2 > 0.1$) decreased from 9 to 2, apparently mitigating the unit-effect and dependency of $b$. Finally, events with low goodness of fit ($R^2 < 0.5$) were discarded. As a result, 291 events and all watersheds, with 5 to 26 events each (Table S4), were included for exploring the landscape and rainstorm effects. Each individual storm event may not necessarily occur in all catchments.

## 3. Results

### 3.1. Recession parameters from individual and point-cloud fits

The streamflow recession plots of catchments W9, W5, and W8, as examples, are illustrated in Fig. 2. The three catchments have distinct differences in landscape, particularly in drainage area ($A$) and the ratio of flow-path length to gradient ($L/G$). Catchment W9 has a larger $A$ and lower $L/G$, W5 has a smaller $A$ and lower $L/G$, and W8 has a smaller $A$ but higher L/G, see Table S2 for catchment details. Median $b$ values, in descending order, were 2.34 in catchment W9, 1.96 in W5, and 1.63 in W8. The point-cloud-derived $b$ values were 1.45 (W9), 1.37 (W5), and 0.88 (W8), showing that point-cloud-derived $b$ values are smaller than median-derived values (Fig. 2c). Notably, the exponent $b$ decreases with storm magnitude in W5 and W8 but increases with storm magnitude in W9 (Fig. 2b and 2c). The contradictory responses observed in these three catchments might be attributed to variations in their landscape structure and rainstorm characteristics. This apparent association is explored further in the Discussion section.

The frequency distributions of the fitted recession coefficients and nonlinearities from all catchments and event records are shown in Figure 3a-b. Recession coefficient $a$ ranged from 0.003 to 0.273 $hr^{-1}$ with a mean of 0.059 $hr^{-1}$ and median of 0.047 $hr^{-1}$. The large difference between the median and mean reflects a right-skewed distribution. Exponent $b$ ranged from 0.90 to 4.39 with a mean of 1.76 and median of 1.69. The small difference between the median and mean suggests a relatively symmetric distribution. Spatial patterns of recession coefficient and exponent $b$ are illustrated in Fig. 3c-d. Generally, larger recession coefficients were seen in the southwestern plain catchments (Fig. 3c), which have higher $L/G$ ratios. Apart from this,

no other distinct pattern can be found in other, more mountainous catchments. Conversely, the plot of recession nonlinearity shows no clear connection to large-scale landscape features on the island (Fig. 3d).

The recession parameters derived from individual segments and aggregated point-cloud data are illustrated in Fig. 4. The variations of recession responses from individual segments differed greatly among catchments. For parameter $a$, point-cloud-derived values, which aggregate all recession segments in a catchment, are much larger than the coefficients from individual segments. Notably, when the catchment size exceeds approximately 500 km$^2$ (W19), the point-cloud-derived coefficients become similar to the third quantile of the coefficient distribution from individual segments. For exponent $b$, the values derived from the point-cloud are consistently close to the lower limit of the distribution of the individual segment-derived values and the median and interquartile range of exponent $b$ derived from individual segments are not correlated with drainage area. These distinct differences between coefficients and exponent $b$ from the two fitting methods make comparison and interpretation difficult. The details of the recession characteristics for each catchment can be found in Table S5.

### 3.2 Relationships between recession parameters and event/landscape variables

To capture how rainfall forcing affects streamflow recession, correlation analyses were performed. The correlation coefficients between recession parameters and event-associated variables are shown in Fig. 5 and Table 2. The total precipitation ($P$), duration ($D$), total streamflow ($Q_{tot}$), antecedent streamflow ($Q_{ant}$), and runoff coefficient ($Q_{tot}/P$) were negatively correlated with the recession coefficient, $a$. The average precipitation intensity ($I_{avg}$) and peak flow ($Q_p$), both of which represent the rainstorm magnitude, were not significantly correlated to $a$. As for initial event conditions, the 7-day antecedent precipitation, $AP_{7day}$, defined as the seven-day rainfall amount prior to a rainstorm, was not correlated to $a$, nor were other $AP$ period lengths (3-, 5-, 14-, and 30-day). Unlike the recession coefficient, exponent $b$ was only correlated with two, $Q_{ant}$ and $Q_p$, with positive and negative correlations, respectively. This indicates that higher antecedent flow could lead to higher nonlinearity and peak flow to lower. Overall, hydrometric forcing moderately controls the coefficient and only slightly affects nonlinearity.

Regarding landscape variables, the average height ($H$), length ($L$), and gradient ($G$) of the flow-path were approx. 120 m, 252 m, and 0.47, respectively (Table S2). The mean $L/G$ value for our catchments was approx. 951m. Forest coverage ($C_F$), ranging between 11.8-92.1%, was the dominant landscape type for most mountainous catchments. Notably, the catchments in the western plain are characterized by gentle gradients of flow-path, such as catchments W8, W9, W11, W12, W13, and W14, where agricultural are the dominant land cover. The correlations of recession parameters against landscape variables are illustrated in Fig. 5 and Table 2. Most landscape variables ($H$, $L$, $G$, $L/G$, $DD$, $S_m$, $HI$, $C_W$, $C_F$, and $C_A$) are significantly correlated with the coefficient, particularly the flow-path-associated ones ($H$, $L$, $G$, $L/G$, and $DD$). Flow-path height ($H$), length ($L$), and gradient ($G$) were negatively correlated to the coefficient, but $L/G$ and $DD$ were positively correlated. Additionally, the coefficient increases as $HI$ and $S_m$ decrease. Looking at land cover, the coefficient increases with $C_W$ (proportion of water body land cover) and $C_A$ (proportion of agriculture land cover) and decreases with $C_F$ (proportion of forest land cover). Greater water-body or agricultural land area in a catchment lead to a faster recession, yet greater forested land area can slow recession. Correlations between $b$ and the landscape variables were generally weaker and of the opposite sign than the correlations seen with $a$. There were also less significant correlations. In short, most landscape variables are moderately associated with the

coefficient and low-to-moderately with exponent *b*. Perhaps, putting all catchments with various landscape features together would obscure the landscape's control in recession.

## 4. Discussion

### 4.1 Recession parameters in small mountainous rivers

The point-cloud estimates are distinctly different from the estimates from the individual recessions (Fig. 4). The larger *a* and smaller *b* values derived from the point-cloud compared to those derived from individual segments could be expected due to the influence of antecedent flow and superimposition of recession events (Jachens et al., 2020). Since *a* and *b* are inherently dependent and while the decorrelation method might be only valid for some specific cases (Biswal, 2021), the way (e.g. fixing *b*) to obtain the *a* or *b* of an individual event is still goal-dependent (Sharma and Biswal, 2022). Even so, using the median

from individual segments is suggested, compared to the point-cloud derivation (Dralle et al., 2017; Jachens et al., 2020).

Higher median recession coefficients were found in W8, W11, W12, and W14, which we attributed to the landscape features of shorter and gentler flow paths, i.e., dense drainage networks. By contrast, catchments with longer and steeper flow paths, such as W7 and W15, have lower median recession coefficients. Taken together, these data demonstrate how drainage density and flow-path-associated variables can affect the recession coefficient. The findings presented in Table 2 corroborate this

(discussed more in Sect. 4.2). On the other hand, the median of exponent *b*, is approx. 1.69 (Fig. 3b) with a range of 0.90 to 4.39, also comparable to the ranges found in the literature. For example, values of *b* from 0.5 to 2.1 could be found in 220 Swedish catchments with low-flow data (Bogaart et al., 2016), 0.6 to 1.7 for 22 Taiwanese rivers derived from low-flow data (Yeh and Huang, 2019), and 1.5 to 3.2 for 67 USA watersheds with event data (Biswal and Marani, 2010). Non-linear storage-outflow relationships (*b* is not equal to 1.0) are prevalent for most catchments worldwide. In our cases, the highest and lowest

median values of *b* were found in W7 and W19, respectively. Despite the fact that these two catchments have similar landscape structures, their exponent *b* exhibits distinct differences. Perhaps, other controlling factors, such as geological structure (i.e., connectivity between the deep aquifer and the stream, heterogeneous hydraulic properties, and/or the interface slope between the shallow and bedrock layers, see Roques et al., 2022) or land cover (Tague and Grant, 2004), might alter recession behavior as well.

### 4.2 Landscape structure controls on the median of recession parameters

Landscape structure aggregates catchment hydraulic properties, embodying recession parameters conceptually. Therefore, recession behaviors in a catchment could be interpreted from two perspectives, hillslope hydraulics and inter-hillslope heterogeneity (Harman et al., 2009), both of which could be represented by the flow-path-associated variables (e.g., *H*, *L*, *G*, *L/G*, and *DD* in Table 2) and drainage area. Notably, heterogeneity increases with drainage area because of the possibility of

including a wider range of subsurface conditions. Recession nonlinearity also increases with drainage area since a larger area accommodates more possibility of superimposition of multiple linear reservoirs, which has been seen in the 68 km$^2$ Mahurangi watershed, New Zealand (McMillan et al., 2014), and the 41 ha Panola Mountain Research Watershed, USA (Clark et al.,

2009; Harman et al., 2009), though this does not appear to be the case in our study (Fig. 6a). Moreover, the correlation analysis showed that flow-path-associated variables ($H$, $L$, $G$, $L/G$, $DD$) only have a weak correlation with the recession nonlinearity (Table 2).

The weak correlation between recession nonlinearity and those variables may be attributed to two factors. First, there is the scale effect. Some of our catchments are much larger than 500 km$^2$, which far exceeds the extent of rainstorms (usually less than 200 km$^2$). In these large catchments, the limited extent of rainstorms would not bring about a comprehensive recession response in the outflow hydrograph (Huang et al., 2012). Second, the drainage area cannot reflect the unknown number of aquifers (Ajami et al., 2011), making it unclear whether a positive relationship exists between nonlinearity and drainage area in our study. Moreover, Karlsen et al. (2019) argued that the dependence of $b$ on landscape variables would change with the streamflow rate. Specifically, flow path height, $H$, dominates the nonlinearity during high flow, whereas the drainage area, $A$, gains more importance during low flow. The relationship between flow-path-associated variables and drainage area and recession needs to be examined in our catchments.

### 4.2.1 Landscape structure controls on recession coefficient $a$

Since drainage area could not solely explain our recession behaviors (Fig. 6a), the flow-path-associated variables and drainage area were used to classify the catchments. Surprisingly, an inverse relationship between the $L/G$ ratio and drainage area emerged (Fig. 6b). The $L/G$ ratio, in fact, is highly correlated to $DD$ and the topographic wetness index (Beven and Kirkby, 1979) and is apt to represent the hillslope hydraulics at a catchment scale. In Fig. 6b, all catchments could be simply classified into three types: type A are large catchments (area > 500 km$^2$), B are small catchments with low $L/G$, and C are small catchments with high $L/G$. Another correlation analysis was performed between recession parameters and the flow-path-associated variables ($H$, $L$, $L/G$, and $DD$) according to these classifications (Fig. 7). The recession coefficients significantly correlated with the flow-path-associated variables, particularly in small catchments (Type B and C only). The fact that $H$ is negatively correlated with the recession coefficient suggests that groundwater flow paths possess greater depth and length, consequently leading to slower drainage rates. While $H$ is commonly believed to be positively correlated with the velocity of gravity-driven flow at a small spatial scale, the high heterogeneity in geology or soil properties at a larger spatial scale (Karlsen et al., 2019) implies that a large $H$ does not necessarily lead to a large recession coefficient. Besides, high $DD$ and short $L$ indicate shorter flow paths and thus lead to a higher recession coefficient. In our cases, Type C catchments are characterized by short $L$ and very small $H$ and thus have high $L/G$ ratios and recession coefficients (solid orange dots in Fig. 7c). Individually, extended $L$ or gentle $G$ is conducive to flow accumulation. Thus, the $L/G$ ratio, which integrates both length and gradient, serves as a good proxy for estimating residence time (McGuire et al., 2005; Asano and Uchida, 2012). Potentially, the relationship between recession parameters and $L/G$ has the chance to establish a further linkage between recession parameters and water residence time.

#### 4.2.2 Landscape structure controls on recession nonlinearity *b*

The recession nonlinearity conditionally responds to landscape structure (Fig. 7e-7h). If Type A catchments (large area with low *L/G*, gray solid dots in Fig. 7) are excluded, all flow-path-associated variables become significantly correlated with exponent *b*. The positive relationship of *b* with *H* indicates that steeper and rougher hillslopes present non-linear recession behavior. Perhaps with the increase of *L*, subsurface runoff has more chances of flowing through various blocks (e.g., temporarily perched groundwater). The two ratios, *DD* and *L/G*, are negatively related to the value of *b* (Fig. 7g-h). Short-and-gentle hillslopes, like Type C catchments, are conducive to larger saturation areas during rainstorms (Bogaart et al., 2016; Sayama et al., 2011), and the expansion of the saturation area indicates that the whole subsurface becomes saturated and connected well, thus reducing heterogeneity. It suggested that the *L/G* ratio affects the nonlinearity significantly for small catchments; however, it is not the cases of our large catchments, which necessitates further interpretation associated with scale.

### 4.3 Rainfall amount controls on the variation of recession parameters

Recession behavior is a convolutional response starting as rain falls within catchments. Thus, we separately examined the recession parameters against hydrometric variables for the three catchment types (Types A, B, and C) to rule out the influences (Fig. 8). This produced two significant findings: (1) the recession coefficient, *a*, decreases with the rainfall amount in all types and (2) the exponent *b* shows contrasting responses in Type A and B (Type C is statistically insignificant). In heterogeneity-dominated (large) or hydraulics-dominated (small and steep) catchments, exponent *b* increases or decreases with rainfall amount, respectively.

#### 4.3.1 Rainfall amount controls on recession coefficient *a*

Several empirical studies found a positive or independent relationship between *a* and streamflow; for example, Santos et al. (2019) found that higher streamflow produced a greater *a* value, reflecting a quick recession in Switzerland's catchments. In Sweden, annual rainfall variation might be independent of the *a* (Bogaart et al., 2016). However, most studies found a negative relationship between *a* and storage measures (Table 1). For instance, Biswal and Nagesh Kumar (2014) found a negative correlation between *a* and the antecedent flow rate, while Ghosh et al. (2015) found that high peak flow events tend to produce a small value of *a*. In our study, the recession coefficients decreased with rainfall amount in all catchment types (Fig. 8a-c). Harman et al. (2009) demonstrated that the recession coefficient can be expressed as $a = V_0/R^{b-1}$ (where $V_0$ and $R$ represent the mean of the velocity distribution of hillslope flow and rainfall rate, respectively). In the case of heavy rainfall, the increase of $R$ is much larger than that of $V_0$. The effect of this disproportionate rainfall input increase on *a* could offset the increase in flow velocity, resulting in a negative correlation. Moreover, Biswal and Nagesh Kumar (2014) used a geomorphological recession flow model $a \propto c/q^{b-1}$ (where *c* and *q* represent the celerity and rate of channel flow, respectively, and which is similar to Harman's theory) to explain why "*a*" is negatively correlated with "*q*." To sum up, the negative correlation between coefficient *a* and rainfall amount (e.g. peak flow and prior soil moisture) is consistent with the literature and is prevalent in most regions (also see Table 1).

### 4.3.2 Opposing controls of rainfall on recession nonlinearity $b$

Literature covering the variation of recession nonlinearity among events is divergent. Previous studies concluded that nonlinearity is controlled by landscape structure and is static or insensitive to rainfall (Biswal and Marani, 2010; Brutsaert and Nieber, 1977; Dralle et al., 2017). In other studies, nonlinearity decreases with streamflow rate, albeit on different temporal scales (Shaw and Riha, 2012; Karlsen et al., 2019; Santos et al., 2019), while nonlinearity has been shown to increase with antecedent flow (Jachens et al., 2020). Furthermore, some studies have even argued that nonlinearity can change over the duration of an event dynamically (Rupp and Selker, 2006; Luo et al., 2018; Roques et al., 2022). Across all catchments, we observed an augmentation of exponent $b$ with antecedent flows, but a decline with peak flow (Fig. 5). This augmentation can be attributed to the overlay of event recession flows onto antecedent flows, amplifying the value of $b$ (Jachens et al., 2020). The inverse correlation between $b$ and peak flow suggests that in the majority of catchments, the existence of active fast flow paths could potentially reduce the recession nonlinearity.

Further, our exponent $b$ showed a positive, negative, and flat relationship with rainfall in Type A, B, and C catchments, respectively (Fig. 8d-f). Small catchment areas (Type B and C catchments) may be explained by a 2-dimensional hillslope model (Roques et al., 2022). During heavy rainfall, when fast flow pathways are activated, the exponent $b$ would decrease (Type B catchments, Steep slopes), whereas Type C catchments (gentle slopes) with more homogeneous hydraulic conductivity would experience smaller changes in exponent $b$. Conversely, the exponent $b$, increases with the rainfall amount in Type A catchments. In large and heterogeneous catchments, the expansion of the contributing area is less steady and more complicated, and thus the nonlinearity increases with rainfall amount. A contrasting response of exponent $b$ to rainfall similar to the one seen in this study was also found in Biswal and Nagesh Kumar (2013), which attributed it to the change in subsurface flow contributions along the channel that affect the response direction of $b$. Our study revealed that landscape structure (mainly by $A$ and $L/G$) and rainfall amount control the direction and magnitude of recession response, respectively. Future research could consider different landscape structures when modeling the intra-event variation of $b$.

### 4.4 Landscape structure regulates recession behavior

The above two sections have demonstrated the influence of landscape and rainfall amount on streamflow recession behavior. Thus, a perceptual model which demonstrates the interactive regulation of landscape structure and rainfall amount on recession nonlinearity is introduced (Fig. 9). Landscape structure is considered in two contexts, spatial heterogeneity (drainage area) and hillslope hydraulics ($L/G$). The drainage area may correlate to the number of perched storages within the catchments, and the $L/G$ ratio, encapsulating hillslope geometry, can indicate the dynamics of the contributing area associated with runoff generation. Along the spatial heterogeneity dimension (from Type B to A, with increasing drainage area), additional perched storages respond increasingly with rainfall amount and thus enhance the recession nonlinearity. Perched storages are expected to occur where the hydraulic conductivity abruptly decreases due to heterogeneous soil properties or geological structures. The existence of perched storages was found in an experimental forested catchment in Taiwan through an intensive soil water monitoring scheme (Liang, 2020). Large catchments may suffer uneven spatial rainfall, which activates perched storages locally, and thus, the nonlinearity increases. On the other hand, along the $L/G$ dimension (increasing from Type B to C), the

heterogeneities of hydraulic conductivities decrease. Heavy rainfall, causing saturation and expansion of the saturation area, can mediate the heterogeneity of hydraulic conductivity and thereby reduce nonlinearity.

## 5. Summary

Streamflow recession, which reflects the rainfall-runoff process after rainstorms, is crucial for baseflow assessment. This study investigated the effects of landscape structure and rainfall amount on recession using power-law recession analysis for 291

rainfall events in small mountainous rivers. In these catchments, the recession coefficient is moderately correlated to landscape structure while nonlinearity is only weakly correlated to landscape structure. If classifying the catchments in accordance with spatial heterogeneity (drainage area) and hillslope hydraulics ($L/G$), the recession coefficient increases with $L/G$ and nonlinearity decreases with $L/G$ significantly in small catchments. This likely reveals that both spatial heterogeneity and hydraulic properties regulate recession simultaneously. Along the hillslope hydraulics dimension, small catchments with high

$L/G$ attributed to their short-and-gentle hillslopes, have higher recession coefficients. Additionally, $L/G$ is negatively correlated to nonlinearity for small catchments, perhaps because short-and-gentle hillslopes can expand saturation area and connect different aquifers easily, thus reducing nonlinearity. Note that $a$ and $b$ are inherently dependent so that some uncertainty might be involved. Even so, both parameters, whether derived using the point-cloud or individual segments (Fig. 4), present similar fluctuations among catchments, which supports our arguments.

Rainfall amount affects the recession coefficient. It decreases with rainfall amount for all catchments. On the other hand, contrasting response directions of nonlinearity to rainfall amount could be found along the dimension of spatial heterogeneity (drainage area). Larger catchments exhibited an increase in recession nonlinearity with higher rainfall, whereas smaller catchments showed a decrease in recession nonlinearity with higher rainfall. Conjointly, an interactive regulation of recession by landscape structure and rainfall amount was proposed. In summary, landscape structure (spatial heterogeneity and hillslope

hydraulics) may determine the recession behavior via various aquifer settings, and the rainfall amount tunes the magnitude of recession nonlinearity. If the perceptual model is valid, two challenges should be addressed further. First, the contrasting response direction of nonlinearity to rainfall, depending on the predominance of spatial heterogeneity, requires further theoretical validation. Clarifying which environmental factors could represent the spatial heterogeneity and hillslope hydraulics is also an arduous task but is crucial for recession estimation. Second, the careful determination of the response direction of

nonlinearity is crucial to the regional recession assessment. An incorrect direction would strongly affect the interpretation, particularly for climatic scenarios. Validating the landscape structure control in rainstorm scale would aid in completing the understating of recession variations.

*Data availability.* Hourly streamflow data can be obtained from Taiwan Water Resource Agency and Tai-Power company. The authors declare that data supporting the findings of this study are accessible from the article and its supplementary materials.

*Author contributions.* Conceptualization and Methodology: JYL and JCH. Data Curation and Validation: TYL. Formal analysis: JYL and CJY. Investigation and Writing – Original Draft: JYL. Writing – Review and Editing: JCH and TRP.

*Competing interests.* The authors claim no potential competing interests

*Acknowledgements.* This research was funded by the Ministry of Science and Technology, Taiwan (110-2811-M-005-509, and 109-2811-B-002-631) and the NTU Research Center for Future Earth (107L901004). J. Y. Lee and C. J. Yang was supported

by the grants from Ministry of Science and Technology, Taiwan (110-2811-M-005-521, 110-2917-I-564-009).

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

**Table 1: Summary of empirical recession study results. Numbers inside cells correspond to the reference numbers in Table S1. The asterisk (*) represents this study. The "+", "-", and "x" samples represent positive, negative, and no correlation with factors, respectively.**

| Factor | Centrality of recession | | Temporal variability of recession | | | | | |
|---|---|---|---|---|---|---|---|---|
| | Long-term | | Inter-annual | | Inter-seasonal | | Inter-event | |
| | â | b | â | b | â | b | â | b |
| *Climate/Moisture* | | | | | | | | |
| Rainfall | 1, 21 (x) | 1 (+) | 21 (x) | 21 (x) | | | * (-) | * (+) |
| | | | | | | | | * (-) |
| | | 21 (x) | | | | | | * (x) |
| Maximum monthly rainfall | | 2 (+) | | | | | | |
| Antecedent flow | | | | | 4, 5 (-) | | 5, 13, 22, * (-) | * (+) |
| Peak flow | | | | | | | 8, * (-) | 19, * (+) |
| | | | | | | | 6 (x) | 19, * (-) |
| | | | | | | | | 8, * (x) |
| Flow rate after peak | | | | | | | 5, 6, 9 (-) | 23 (-) |
| | | | | | | | 23 (+) | |
| Total storage change | | | | | | 11 (+) | | |
| Water table elevation | | | | | 3 (-) | 3 (x) | | |
| Saturated area | | | | | 3 (-) | 3 (x) | | |
| 60 cm soil moisture | | | | | 3 (-) | 3 (-) | | |
| Baseflow | 1 (-) | | | | | | | |
| Evapotranspiration | 1, 21 (x) | 1 (-) | | | 3, 12, 24 (+) | 24 (-) | | |
| | | 21 (x) | | | | 3 (x) | | |
| Aridity index | 16, 17, 21 (+) | 15, 16, 17, 21 (-) | | | | | | |
| Mean relative humidity | | 2 (+) | | | | | | |
| *Landscape* | | | | | | | | |
| Drainage area | 10, 16, 20 (-) | 1, 7, 16, 18, 24 (+) | | | | | | |
| | 1, * (x) | 20, * (x) | | | | | | |
| Long shape of catchment | 10 (+) | * (+) | | | | | | |
| | * (x) | | | | | | | |
| Flow path height | * (-) | 24, * (+) | | | | | | |
| Flow path length | * (-) | * (+) | | | | | | |
| Flow path gradient | * (-) | * (x) | | | | | | |
| Mean elevation | | 2 (+) | | | | | | |
| Standard deviation of elevation | | 2 (+) | | | | | | |
| Catchment slope | 17, 24 (-) | 2, 17, 24 (+) | | | | | | |

| | | | | |
|---|---|---|---|---|
| Hypsometric integral | * (-) | * (+) | | |
| Coefficient of variation of slope | 16 (+) | 16 (-) | | |
| Topographic wetness index | | 2 (-) | | |
| Ratio of flow-path length to gradient | * (+) | * (x) | | |
| Drainage density | 14, * (+) | 14 (-) | | |
| | 10 (-) | * (x) | | |
| Subsurface flow contact time | | 2 (-) | | |
| *Landcover* | | | | |
| Reforestation | | | 21 (-) | 21 (+) |
| Water management | | | 21 (-) | 21 (+) |
| Plateaus coverage | | 15 (-) | | |
| Young volcano rock coverage | 14 (-) | 14 (+) | | |
| Forest coverage | 18, * (-) | * (x) | | |
| Water bodies coverage | * (+) | 16, 21, 24 (-) | | |
| | 21 (-) | | | |
| | 16 (x) | * (x) | | |
| Flood attenuation due to lakes | 1 (+) | 1 (x) | | |
| *Soil* | | | | |
| Soil depth | | 24 (+) | | |
| Surface hydraulic conductivity | 17 (+) | 21 (+) | | |
| | 18 (-) | 17 (-) | | |
| Field capacity | 16 (-) | 16 (+) | | |
| Moderate infiltration rate soils | | 2 (+) | | |
| Slow infiltration rate soils | | 2 (-) | | |
| Playas with impermeable soils | | 15 (-) | | |
| Organic matter content | | 2 (+) | | |

**Table 2: Spearman correlation coefficients between logarithmic hydrometric characteristics and recession characteristics for all rainfall events at all catchments (n = 291). Bold fonts represent statistically significant at the 99% confidence level (p-value < 0.01).**

| Variable | Meaning | $a$ [hr$^{-1}$] | $b$ [-] |
|---|---|---|---|
| Hydrometric | | | |
| $AP_{7day}$ [mm] | 7-day antecedent precipitation | -0.080 | 0.010 |
| $P$ [mm] | Total precipitation | **-0.524** | -0.083 |
| $D$ [hr] | Duration of precipitation | **-0.432** | -0.054 |
| $I_{avg}$ [mm hr$^{-1}$] | Averaged precipitation intensity | **-0.257** | -0.026 |
| $Q_{tot}$ [mm] | Total streamflow | **-0.609** | -0.154 |
| $Q_{ant}$ [mm] | Antecedent streamflow | **-0.339** | **0.266** |
| $Q_p$ [mm] | Peak flow | **-0.247** | **-0.228** |
| $Q_{tot}/P$ [-] | Runoff coefficient | **-0.337** | -0.097 |
| Landscape | | | |
| $H$ [m] | Flow-path height | **-0.491** | **0.224** |
| $L$ [m] | Flow-path length | **-0.520** | **0.302** |
| $G$ [-] | Flow-path gradient | **-0.453** | 0.189 |
| $L/G$ [m] | Ratio of flow-path length to gradient | **0.470** | -0.181 |
| $A$ [km$^2$] | Drainage area | 0.040 | -0.095 |
| $DD$ [km] | Drainage density | **0.420** | -0.217 |
| $S_m$ [%] | Gradient of main stem | **-0.318** | **0.229** |
| $HI$ [-] | Hypsometric integral | **-0.498** | **0.226** |
| $ELO$ [-] | Basin elongation | -0.209 | **0.319** |
| $C_W$ [%] | Land cover - water bodies | **0.330** | -0.147 |
| $C_F$ [%] | Land cover - forest | **-0.281** | 0.140 |
| $C_A$ [%] | Land cover - agriculture | **0.268** | -0.059 |

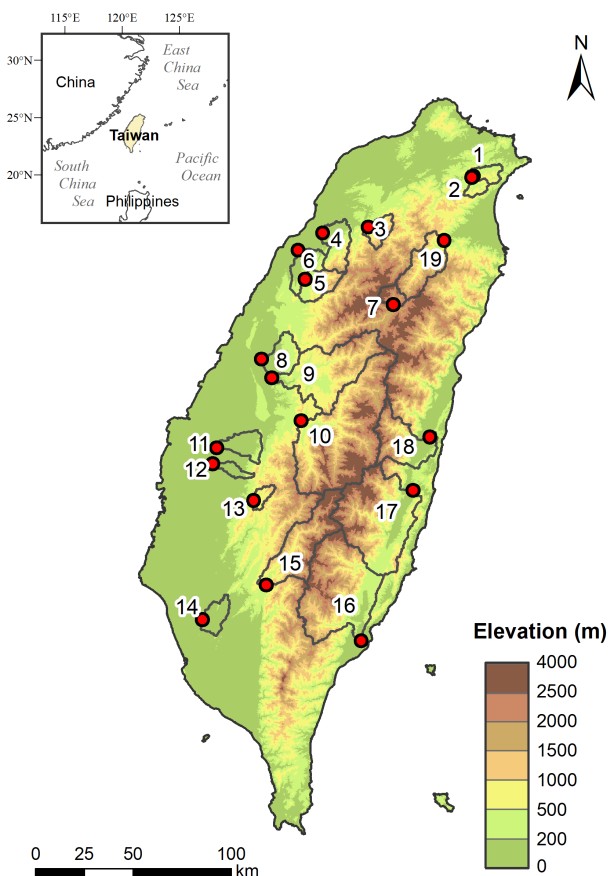

**Figure 1: Topographic map of Taiwan and the locations of the selected catchments (red dots) and associated watersheds (outlines). The catchment IDs correspond to the IDs in Tables S2 and S3, in which the primary descriptions of hydrologic events and landscape variables are listed.**

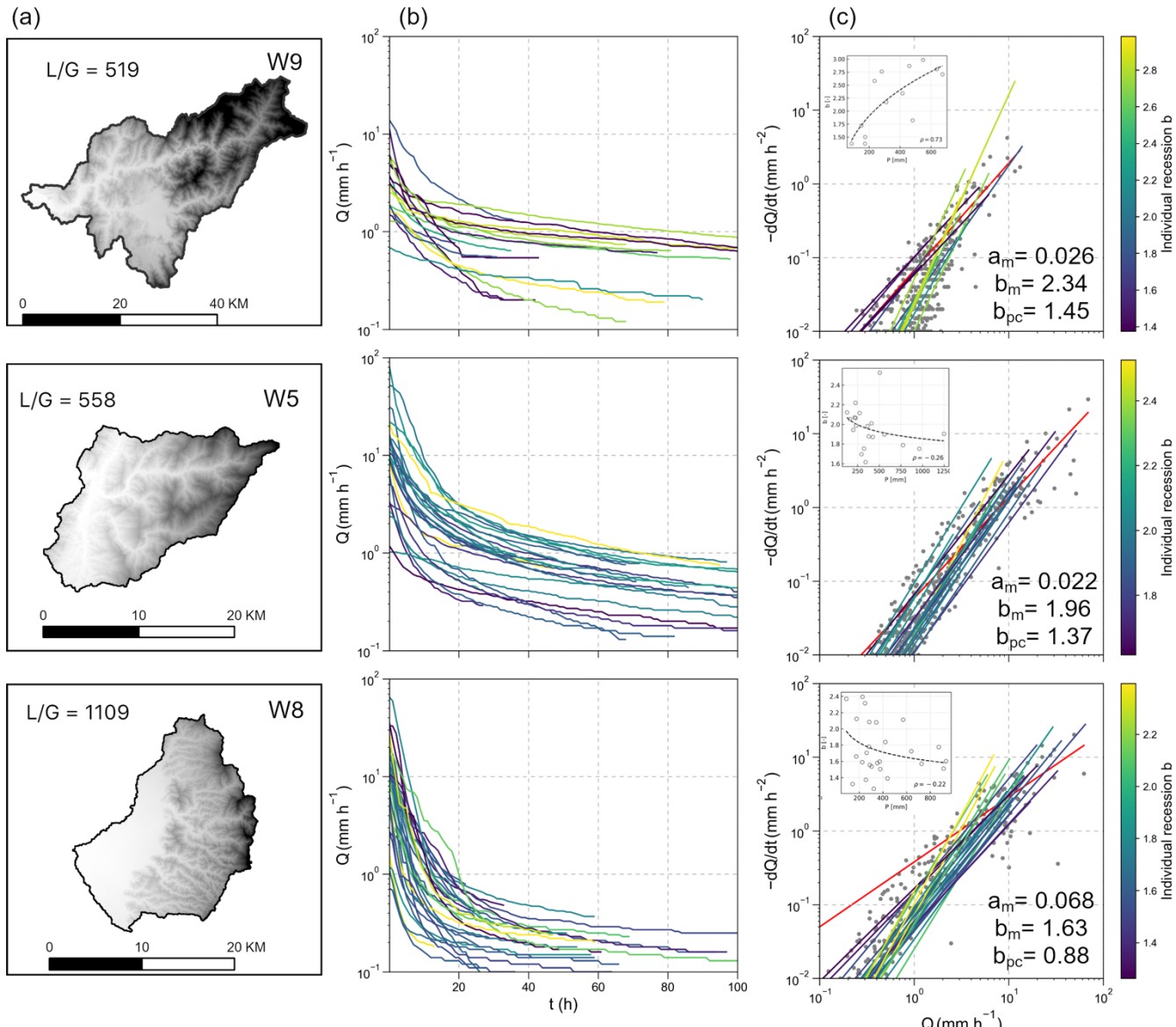

520

**Figure 2: Landscape and recession plots for catchment W9 (row 1), W5 (row 2), and W8 (row 3). Landscape and flow-path topography (L/G) are shown in column (a). Selected recession segments from different rainstorms are shown in (b). Recession plots of all selected rainstorms are shown in column (c). The median of recession parameters *a* and $b_m$ and the parameter $b_{pc}$ derived from the point-cloud are shown in the lower-right corner. The recession *b* from 525 individual segments are colored from purple to yellow with increasing value of *b*, and the red line represents *b* derived using the point-cloud method.**

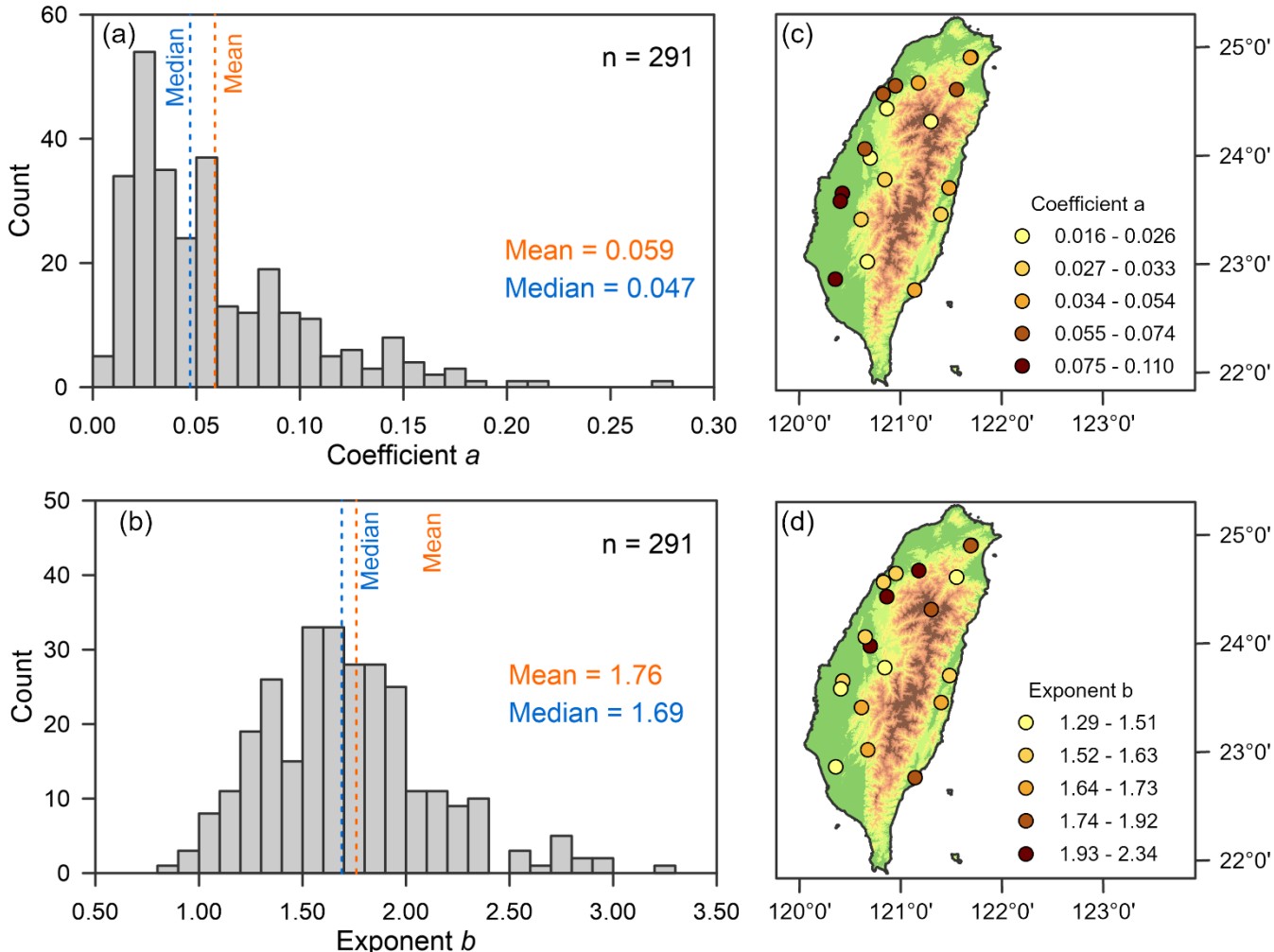

Figure 3: Distributions of recession parameters *a* (a) and *b* (b) estimates in all catchments and events. The spatial distributions of the medians of parameters *a* (c) and *b* (d). The colors of the dots represent the quantiles category.

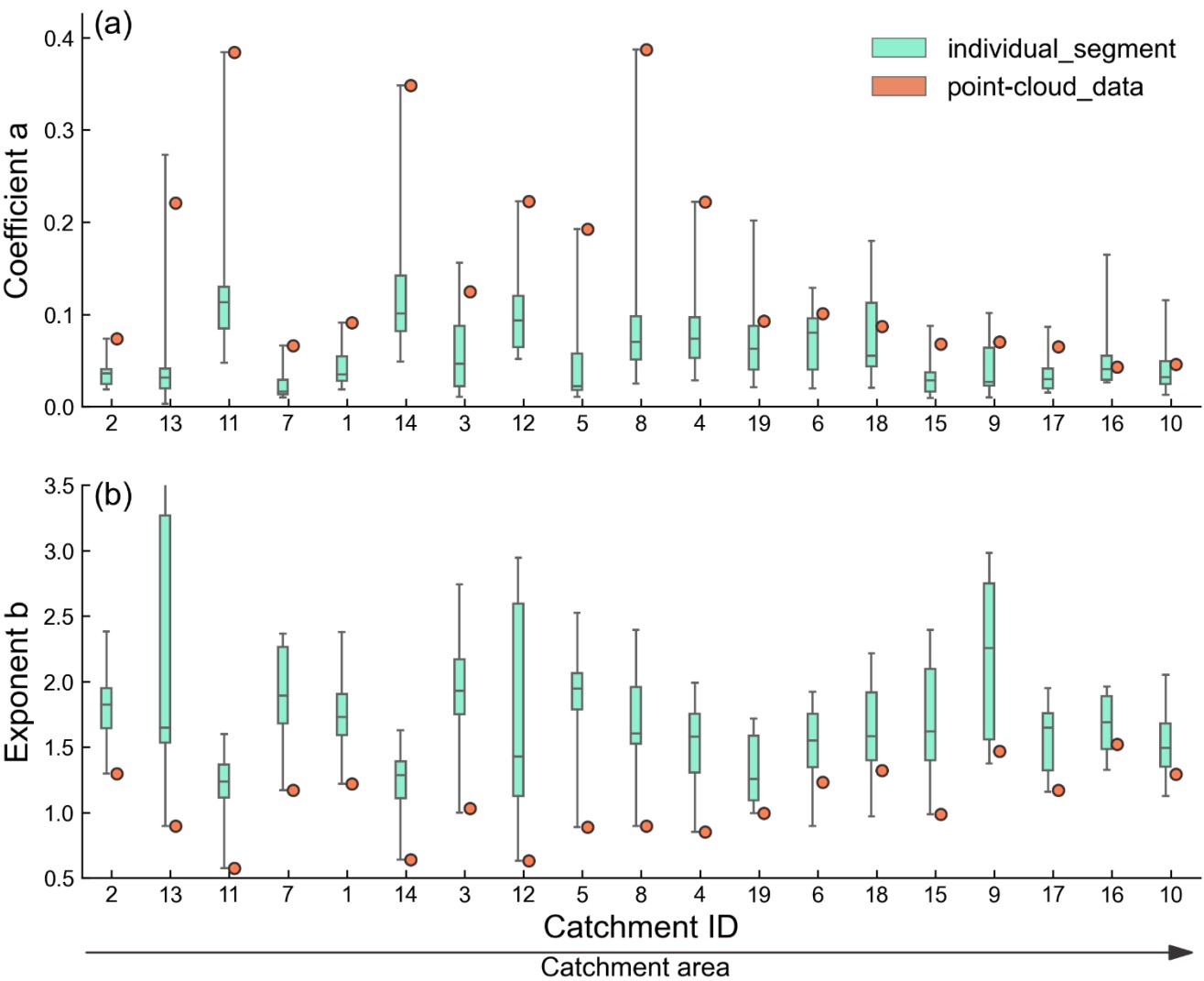

Figure 4: Boxplots of coefficient *a* (a) and exponent *b* (b) derived from the individual recession segments (cyan box) and point-cloud data (orange dot). The catchments are arranged on the *x*-axis in ascending order according to drainage area. Boxes show the interquartile and data range.

535

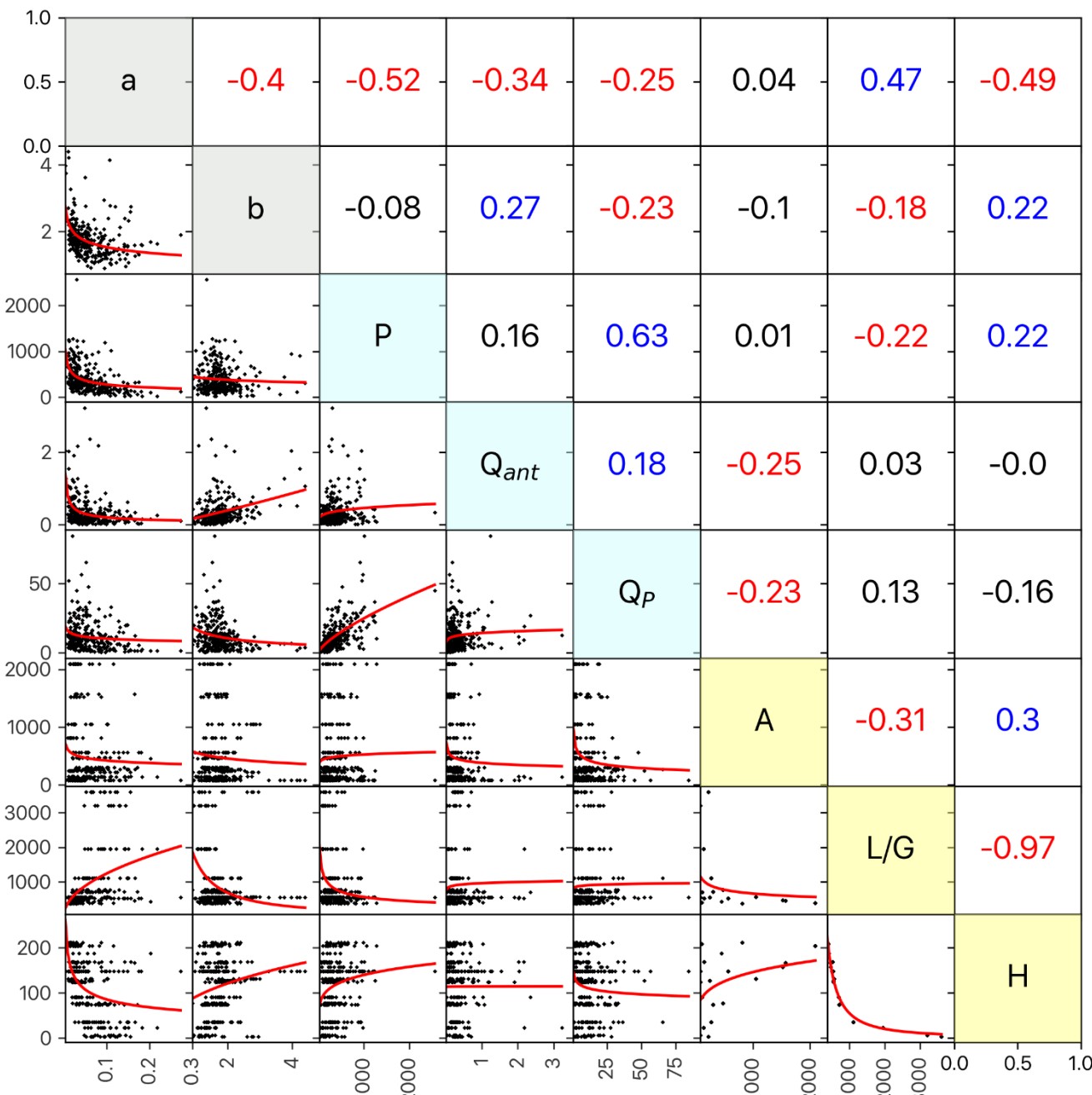

**Figure 5: Correlation of recession parameters *a* and *b* against rainstorm event and landscape variables. Below the diagonal: pairwise scatter plots of the recession parameters and variables with a power-fit regression (red line). Above the diagonal: corresponding Spearman correlation coefficients. Blue and red values indicate statistically significant (*p* < 0.05) positive and negative correlations, respectively. Note that all catchments and events are shown in this figure.**

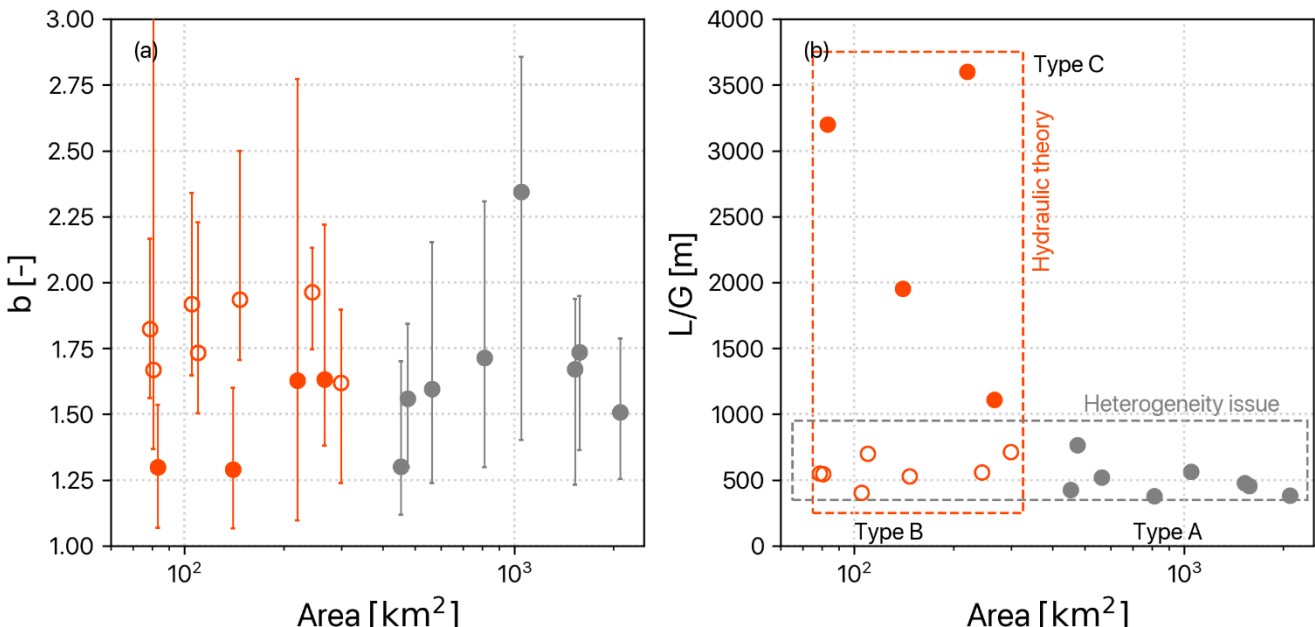

**Figure 6: The relationship between drainage area and the recession exponent *b* (a) and the flow path topography (*L*/*G*) (b). The error bar on (a) is the range of the individual segment recession exponent values of each catchment. The orange and gray dots represent small and large catchments (< and > 500 km², respectively), respectively, and the solid and hollow dots represent large and small L/G. The recession behaviors in small and large catchments could be explained from two perspectives: hydraulic theory (orange box) and heterogeneity issues (gray box).**

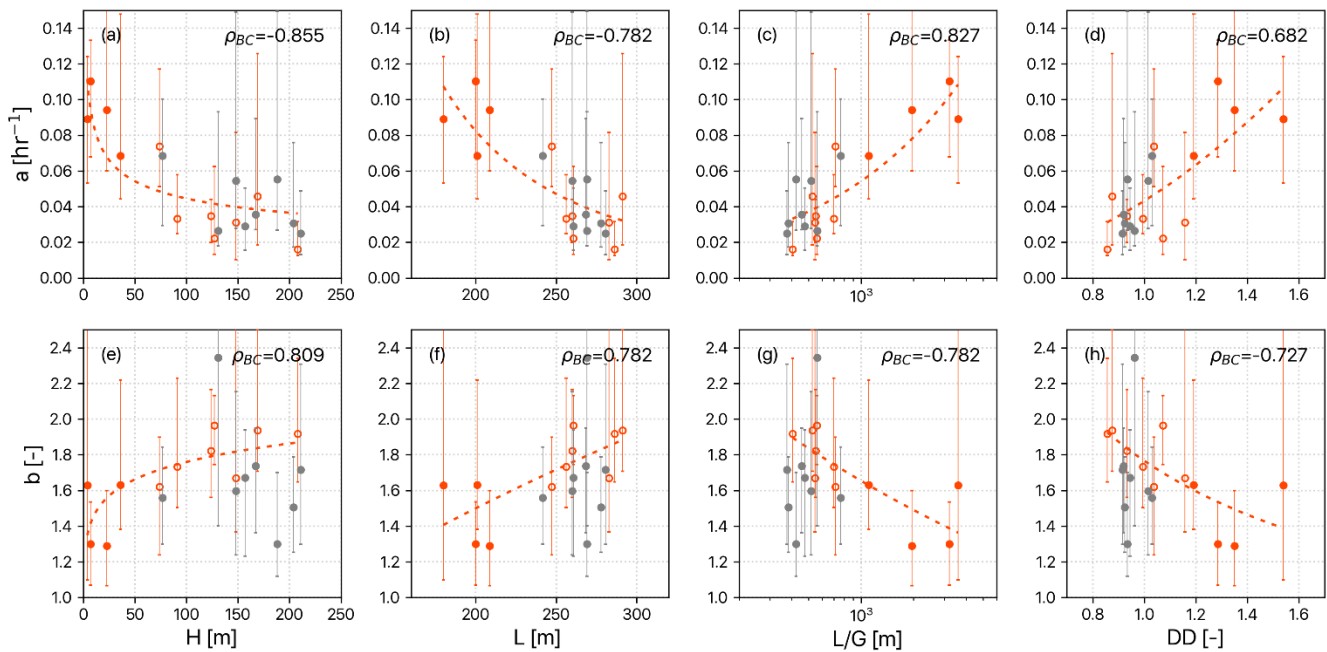

**Figure 7: Scatter plots of the median and the range of 10th-90th percentile of recession parameters at each catchment against landscape variables. Gray solid, orange hollow, and orange solid dots are Type A, B, and C basins, respectively. The orange dash line is the power-law fit for small catchments (Type B and C). The Spearman correlation coefficient ($\rho$) is listed in the upper-right corner of each panel.**

555

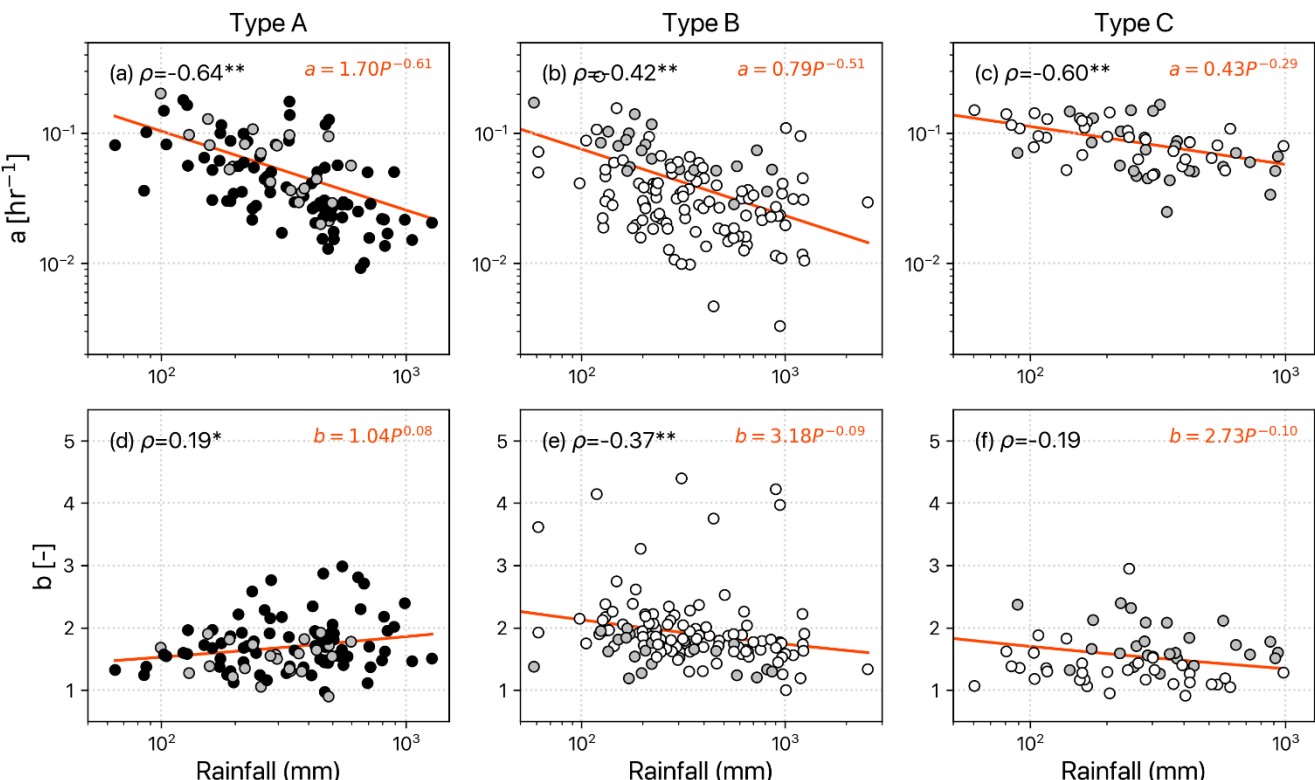

**Figure 8: Scatter plots of recession coefficient, a and exponent, b against total rainfall for recession segments at different catchment types. Type A are large catchments (area > 500 km²), B are small catchments with low *L/G* ratios, and C are small catchments with high *L/G* ratios. Black, gray, and white dot colors represent the low, medium, and large *L/G* catchments, respectively. The orange line is the power-law fitting curve with Spearman correlation coefficients in the upper-left corner of each panel (\* and \*\* denote statistical significance at the 90% and 99% level of confidence, respectively).**

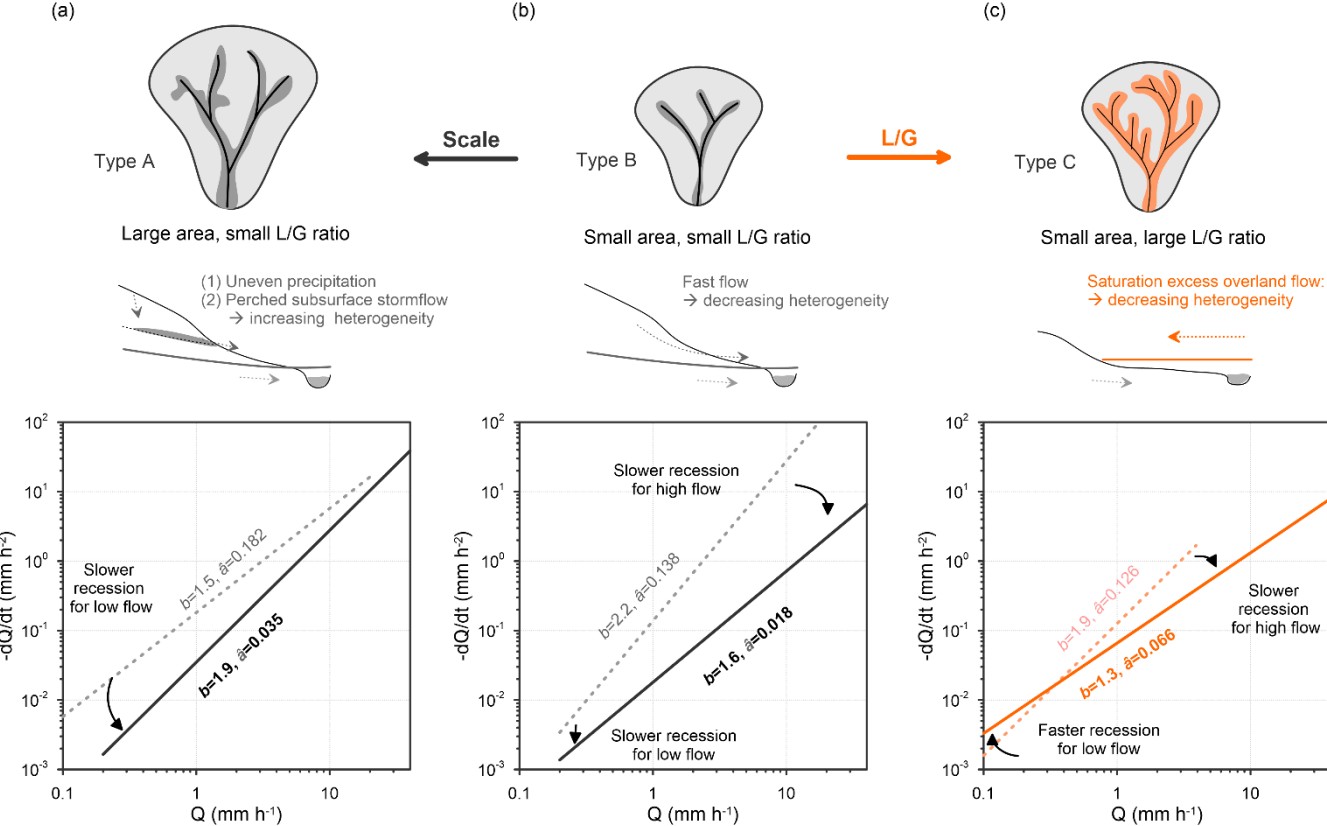

**Figure 9: A conceptual diagram illustrating how landscape variables regulate the recession direction during rainstorms.** The top row presents the drainage area and the stream network of three landscape types of catchments corresponding to Fig. 6b. The middle row presents the cross-sectional valley with descriptions of drainage behavior. Here, (a) type A, large catchment and steep slope, drains water via multiple sources of subsurface flow; (b) type B, small catchment and steep slope, drains water via fewer sources of subsurface flow; and (c) type C, small catchment and gentle slope, drains via the extension of the saturated zone along the riparian zone. Correspondingly, the bottom row shows how their recession parameters (or regressive line) in recession plots would move from light (dashed line) to heavy (solid line) rainstorms.