# Peer review of "Landscape structures regulate the contrasting response of recession along rainfall amounts"

_Hydrology and Earth System Sciences, 2022_

## Referee Comment (RC1)

Landscape structure and rainstorms swing the response of recession nonlinearity

SUMMARY

This paper examines variability in streamflow recessions from and across 19 catchments in Taiwan. Recessions are characterized by the power-law recession parameters $a$ and $b$ in $-\mathrm{d}Q/\mathrm{d}t = aQ^b$. Differences in these parameters among recessions from a single catchment are compared to what are effectively antecedent moisture conditions using, as proxies, variables such precipitation amount and duration, discharge at the beginning of the recession, among others. Differences in parameters across catchments are compared to landscape properties, such as catchment area, shape, drainage density, stream length, among others. Parameter $b$, a measure of nonlinearity (where $b = 1$ indicates linearity) is found to increase with antecedent moisture in some basins but decrease in others. Large basins tend to show the former response, while smaller basins show the latter. In general, the smaller basins show the strongest relationship of $b$ with landscape properties: e.g., $b$ decreases with increasing drainage density in these basins. A hypothesis related to the degree of landscape heterogeneity in a basin is given for the contrasting responses between smaller and larger catchments, and two types of smaller catchments.

GENERAL COMMENTS

The main contribution of this paper is the identification of different responses of recession shape (as characterized by $b$) to antecedent moisture and the apparent connection of these different responses to landscape properties. This is an important finding and may help explain contrasting results from other studies. Ultimately, I think this work could and should be published.

However, the paper would require major revisions as there is one serious issue and a couple of weaknesses, which I describe below.

1. The most serious issue in the paper is the faulty analysis of the parameter $a$. The fact is that no physical significance can be ascribed to changes in $a$ when $b$ also changes concurrently, therefore the paper's interpretation of the variability of $a$ in this paper is flawed.

A problem arises from the units $a$, which change as $b$ changes, as such the paper makes a nonsensical comparison of values of $a$ with different units. One consequence of the scale dependence of $a$ on $b$ is that the reported differences in $a$ among recession events is dependent, even in a relative sense, on the units the authors use for discharge $Q$. If this study were to use units other than mm hr$^{-1}$, not only would the relative magnitudes of the differences change, so could the sign of the difference (while zero change is also possible given the correct units). If the reported differences in $a$ had a physical significance, simply changing the units shouldn't change the physical interpretation. If in doubt, I suggest the authors redo some of their analysis after changing the units of streamflow from mm hr$^{-1}$ to km hr$^{-1}$ and nm hr$^{-1}$ to see the effect.

This unit dependence of $a$ on $b$ is why, for example, Tashie et al. (2020b) and others fix $b$ within a catchment and estimate $a$ for each of the catchment's recessions. However, this does not solve the dilemma of comparing $a$ *across* catchments where the catchments have different values of $b$.

I recommend the authors look to Dralle et al. (2015) and Biswal (2021) for further discussion on the relationship between the power law coefficients.

2. Although the authors reference various papers that have empirically examined the relationships between the power law recession parameters and environmental factors, very much has been published on these topics that is not referenced including relatively recent work (e.g., Tashie et al. 2020b). The paper should have a more comprehensive summary of prior work, followed by a clearer statement of what is still poorly understood, and finishing with what this study proposed to do to address one or more outstanding questions. While the intro does this to some extent, it is not sufficient.

A very valuable contribution would be, possibly in tabular form, a list of those environmental factors considered along with the studies that have found positive/negative/no relationship between these factors with recession parameters. This would clearly illustrate how much has been done and, hopefully, demonstrate why yet another study of this type is still necessary.

Note that while I have referenced numerous papers in this review, they do not include many empirical studies of recession parameters.

3. The paper would greatly benefit from a discussion of what theory would predict for the influence of environmental factors on recession parameters. For examples, Figures 2 and 3 in Rupp and Selker (2006b) show how initial water table height (i.e., antecedent moisture), drainage density, hillslope slope and hillslope length/height ratio determine $a$ and how vertical heterogeneity and initial conditions influence $b$. Are the results of this paper consistent with theory? If not, why not? Perhaps theory breaks down outside of the idealized conditions upon which the theory is based? Theoretical work has also shown how planform shape and downstream boundary conditions (e.g., Troch et al. 2013) as well as a draining vadose zone (Luo et al., 2018) and drainage network geomorphology (Biswal and Marani 2010) can affect recession parameters.

LINE-BY-LINE COMMENTS

L1: The title could be improved. First, the meaning of the word "swing" in this context is unclear. A pendulum swings. I don't think that is what the authors mean to say. Suggested replacements for "swing" are "modify" or "alter". Second, "the response of recession nonlinearity" is also unclear.

L12. Is it 260 sets of recession parameters per catchment, or in total over all 19 catchments?

L29-31: This sentence is out of place and not particularly relevant. The previous sentences are about analyzing individual recession events, while this is about projections of future rare rainfall events. Unless a stronger link is made, I would delete this sentence.

L32: Define $Q$ and $t$ in $-dQ/dt = aQ^b$.

L34. I have an issue with calling $a$ the recession rate. The units of $a$ vary with $b$, so are not universally consistent with a "rate".

L38-41: It should be stated where discharge has been normalized by catchment area. From Figure 2, I take it that is has for the authors' analysis. The authors should be careful when discussing results from other studies that may not have normalized discharge. Brutsaert and Nieber (1977), for example, do not normalize discharge prior to comparing $a$ from different basins with drainage density and network length but clearly dividing by area first would affect their values of $a$. Brutsaert and Nieber (1977) show an inverse relationship of $a$ with total stream length (their Fig. 9) with seems to contradict the attribution to Bogaart et al. (2016) that $a$ has a positive correlation with total stream length. Also, flow-path length and height need to be clearly defined.

L42-43: Please cite the "few studies".

L46-47: I believe all the references except this one concern empirical studies, which makes this reference to theoretical work out of place. The statement is also unclear without more context. How does spatial heterogeneity affect whether increasing the steady-state recharge rate (I assume the authors mean a steady-state recharge rate immediately prior to the beginning of the recession) increases or reduces $a$? Maybe 1-2 paragraphs devoted to theoretical work could be included (see COMMENTS above).

L50: Start new paragraph at "Due to…".

L76-77: Clearer definitions of L, H, and G are needed. Table S1 gives definitions in the footnotes, but they do not appear to be consistent with what is in the main text. The text says "flow-path length [L]" but Tables S1 says L is total length of the drainage networks. Are they the same thing?  If so, the text on line 40 is confusing because total stream length and flow-path length are treated there as if they are distinct measures. The text also says "gradient (G) above the nearest channel" but the Table S1 footnote says "G is the average gradient of the drainage networks". These do not sound like the same thing.

L86: Theissen polygons can be quite poor for interpolation of rainfall, particular in regions with sharp rainfall gradients such as in Taiwan. Is there additional information that can aid the interpolation, even a rainfall vs. elevation relationship? Are there any gridded climatologies (such as PRISM maps) that can be used to improve interpolation? What is the rain gauge density? Can the gauge locations be shown on a map?

L104-106: Dralle et al. (2017) could also be cited as an example of an examination of the effects of methodological choices.

L127-L131: How many recession events were ultimately included per station? Were they the same events in time per station?

L134: How exactly were d$Q$/d$t$ and $Q$ estimated from the data?

L152: Define elongation (ELO). How is it calculated? This should be explained in the methods section.

L145-146: This sentence is confusing. First, please explain what it means that "the two parameters are interactively dependent". Second, why are they "particularly" dependent "when the number of points is huge". Lastly, what does this dependence have to do the ordinary least squares method?

L152-153: If the properties of W8 and W18 are described here, then so should they be for W5. How is W5 distinct from W8 and W18?

L153-154: It is meaningless to rank in descending order of $a$ if all the $a$ do not have the same units. I suggest ranking them in order of $b$.

L154-155: Why are the mean and median of $b$ stated for W8 but not for the others?

L155-156: This sentence is confusing and I wonder if there is an error. Why would the median > mean of the recession rate ($a$) imply that the distribution of nonlinearity ($b$) is right-skewed? Fig 2c doesn't actually give the mean of $a$.

L156-157: A plot of $b$ vs storm magnitude and/or $Q_{ini}$ for each of these three watersheds would be very helpful to illustrate the point being made, and show how strong these relationships actually are.

L157-158: I think the opposite response of W8 and W5 to storm magnitude being associated with differences in landscape properties should be left for the Discussion, but it is OK to foreshadow the discussion here. If this sentence is kept, I would follow it by saying that this apparent association will be explored further in the Discussion section.

L160: Units missing for $a$.

L167-170: Jachens et al. (2020) argue that this is not necessarily true; the point cloud fitting method may not reveal the "general" or average recession.

L187: It should be noted here that P and $Q_{ini}$ are effectively uncorrelated ($r$ is only 0.11 and is not significantly different from zero). This seems like an important point and worth a little more

discussion. How exactly is P calculated for each recession event? Over what time period is it totaled? This should be described in the methods.

L188: What are these "presumed thoughts". I would reiterate the thoughts here or leave this phrase out.

L191: Describe how is L/G is a proxy for the interaction of landscape and climate. References?

L214: Circular logic. The authors' definition of non-linearity is already that the value of *b* is not equal to 1. Also, values less than 1 are non-linear.

L220: It would be clearer to simply say highest and lowest values of *b*. Or "the most and least nonlinear cases are…".

L230-232: But see also Sharma and Biswal (2022).

L238: "Complicates" is not a good term here. I think the authors mean to say that heterogeneity may increase with catchment area because of the possibility of including a wider range of subsurface conditions. This sentence should be rewritten for clarity.

L259: This is the first time any "Type" of catchment is mentioned. The classification of catchments into Types needs to be introduced more clearly. I would start a new paragraph and direct the reader to the upper half of Fig 9 (which will mean reordering the figures).

L271: I suggest rewriting as "The positive relationship of *b* with both H and L…."

L272: I don't understand this sentence. What are these "blocks"? How does higher H and L imply greater prevalence of such blocks?

L273-274: This is an important idea the authors introduce. Do only catchments with short AND gentle hillslopes have large riparian areas? How exactly does a larger riparian area reduce heterogeneity?

L276-277: Do these large basins have something in common other than being large? Almost all these large basins have their headwaters at the highest elevations and most of the smaller basin are on the west side. These smaller basins are also mostly in the rainshadow, whereas the larger basins receive much more rainfall. What role could these factors play?

L279: "The large deviation" of what? Please be explicit.

L303-314: I think it worth noting that *b* is treated as a constant here throughout a single recession event, though it has been empirically shown that it can change over the course of an event (e.g., Rupp and Selker 2006b; Tashie et al. 2020a). Also, groundwater hydraulic theory predicts that *b* can change over time (Brutseart and Nieber 1977). For a horizontal aquifer, this

change depends on the initial conditions (Rupp and Selker 2006a) but the change in *b* happens relatively early and *b* becomes relatively steady as the recession progresses. In a sloping aquifer and/or one that is vertically compartmentalized, *b* can change over a lengthy part of the recession (e.g., Bogaart et al. 2013; Roques et al. 2022). How this theoretical idealized hillslope behavior might manifest in a complex catchment has still not been not well-described, however.

L306: This is an interesting idea that the pervasive saturated overland flow reduces the nonlinearity of recession.  Two issues:
1) What field evidence is there of this pervasive saturated area?
2) I expect this saturated area is decreasing in time, possible very quickly. How would this affect *b*? Thinking along the geomorphological lines of Biswal and Marani (2010), might this not increase *b*?

L309-311: I'm not sure I follow this. How would a large rainstorm "connect" saturated zones of slow reservoirs that were otherwise not draining to a stream? What may be happening is that during large storms there is a wider range of active quickly to slowly draining sources (the fast ones being the ones activated during the large storms).  This heterogeneity of sources can increase *b*. This appears to be what the authors say in the sentences following this one.

L326: Is there any field evidence for these perched storages?

L329: I wouldn't say "unpredictable". Predictability is not the issue here.

L340: I would not say "pretty diverse". Is "inconsistent" what is meant?

L346: What is meant by "higher hillslope hydraulics"?

Figure 2: The panels in column c clearer show discretization artifacts that visually hide the underlying relationship at low flows. A way to remove these artifacts was first proposed by Rupp and Selker (2006b) and modifications were made by Roques et al. (2017) and Guo et al. (2022). I suggest the authors apply one of these methods.

Figure 5: State in caption whether all stations and all events are shown in this plot.

Figure 7: Say in caption which symbols are for Type A, B, and C, basin.

Table S1: Some of these basin average drainage network gradients are very large (as high as 0.75). Hillslope gradients must be yet larger. What are the implications for subsurface flow?

Table S1: L/G is given as having units of $m^2$. If L has units of m and G is unitless, L/G must have units of m.

**References**

Biswal, B. (2021). Decorrelation is not dissociation: there is no means to entirely decouple the Brutsaert-Nieber parameters in streamflow recession analysis. Advances in Water Resources, 147, 103822. https://doi.org/10.1016/j.advwatres.2020.103822

Biswal, B., & Marani, M. (2010). Geomorphological origin of recession curves. Geophysical Research Letters, 37(24). https://doi.org/10.1029/2010GL045415

Bogaart, P. W., Rupp, D. E., Selker, J. S., & Van Der Velde, Y. (2013). Late-time drainage from a sloping Boussinesq aquifer. Water Resources Research, 49(11), 7498-7507. https://doi.org/10.1002/2013WR013780

Brutsaert, W., & Nieber, J. L. (1977). Regionalized drought flow hydrographs from a mature glaciated plateau. Water Resources Research, 13(3), 637-643. https://doi.org/10.1029/WR013i003p00637

Dralle, D. N., Karst, N. J., Charalampous, K., Veenstra, A., & Thompson, S. E. (2017). Event-scale power law recession analysis: quantifying methodological uncertainty. Hydrology and Earth System Sciences, 21(1), 65-81. https://doi.org/10.5194/hess-21-65-2017

Dralle, D., Karst, N., & Thompson, S. E. (2015). a, b careful: The challenge of scale invariance for comparative analyses in power law models of the streamflow recession. Geophysical Research Letters, 42(21), 9285-9293. https://doi.org/10.1002/2015GL066007

Gao, M., Chen, X., Singh, S. K., & Wei, L. (2022). An improved method to estimate the rate of change of streamflow recession and basin synthetic recession parameters from hydrographs. Journal of Hydrology, 604, 127254. https://doi.org/10.1016/j.jhydrol.2021.127254

Jachens, E. R., Rupp, D. E., Roques, C., & Selker, J. S. (2020). Recession analysis revisited: Impacts of climate on parameter estimation. Hydrology and Earth System Sciences, 24(3), 1159-1170. https://doi.org/10.5194/hess-24-1159-2020

Luo, Z., Shen, C., Kong, J., Hua, G., Gao, X., Zhao, Z., ... & Li, L. (2018). Effects of unsaturated flow on hillslope recession characteristics. Water Resources Research, 54(3), 2037-2056. https://doi.org/10.1002/2017WR022257

Roques, C., Rupp, D. E., & Selker, J. S. (2017). Improved streamflow recession parameter estimation with attention to calculation of– dQ/dt. Advances in Water Resources, 108, 29-43. https://doi.org/10.1016/j.advwatres.2017.07.013

Rupp, D. E., & Selker, J. S. (2006a). On the use of the Boussinesq equation for interpreting recession hydrographs from sloping aquifers. Water Resources Research, 42(12). https://doi.org/10.1029/2006WR005080

Rupp, D. E., & Selker, J. S. (2006b). Information, artifacts, and noise in dQ/dt– Q recession analysis. Advances in Water Resources, 29(2), 154-160. https://doi.org/10.1016/j.advwatres.2005.03.019

Sharma, D., & Biswal, B. (2022). Recession curve power-law exponent estimation: is there a perfect approach?. Hydrological Sciences Journal. https://doi.org/10.1080/02626667.2022.2070022

Tashie, A., Pavelsky, T., & Band, L. E. (2020a). An empirical reevaluation of streamflow recession analysis at the continental scale. Water Resources Research, 56(1), e2019WR025448. https://doi.org/10.1029/2019WR025448

Tashie, A., Pavelsky, T., & Emanuel, R. E. (2020b). Spatial and temporal patterns in baseflow recession in the continental United States. Water Resources Research, 56(3), e2019WR026425. https://doi.org/10.1029/2019WR026425

Troch, P. A., Berne, A., Bogaart, P., Harman, C., Hilberts, A. G., Lyon, S. W., ... & Verhoest, N. E. (2013). The importance of hydraulic groundwater theory in catchment hydrology: The legacy of Wilfried Brutsaert and Jean-Yves Parlange. Water Resources Research, 49(9), 5099-5116. https://doi.org/10.1002/wrcr.20407

---

## Author Comment (AC1)

**Reply on Reviewer Comment #1**

SUMMARY

This paper examines variability in streamflow recessions from and across 19 catchments in Taiwan. Recessions are characterized by the power-law recession parameters a and b in $-dQ/dt = aQ^b$. Differences in these parameters among recessions from a single catchment are compared to what are effectively antecedent moisture conditions using, as proxies, variables such precipitation amount and duration, discharge at the beginning of the recession, among others. Differences in parameters across catchments are compared to landscape properties, such as catchment area, shape, drainage density, stream length, among others. Parameter b, a measure of nonlinearity (where b = 1 indicates linearity) is found to increase with antecedent moisture in some basins but decrease in others. Large basins tend to show the former response, while smaller basins show the latter. In general, the smaller basins show the strongest relationship of b with landscape properties: e.g., b decreases with increasing drainage density in these basins. A hypothesis related to the degree of landscape heterogeneity in a basin is given for the contrasting responses between smaller and larger catchments, and two types of smaller catchments.

**Reply:**

We sincerely appreciate Reviewer #1's comments. The reviewer is professional in recession analysis, fully understands our study, and points out the merits and weaknesses in the analysis as well. The main goal of this study attempts to clarify the recession responses to rainfall and landscape. Unlike previous studies which retrieved parameters from the synthesized point clouds or used the median of parameter distributions to discuss the effect of landscape on recession, we retrieved the *a* and *b* from individual event and thus the recession responses to different rainstorms under various landscape settings could be identified. Our results demonstrated that landscape heterogeneity (e.g. drainage area and L/G) in a basin regulates the direction of recession responses. All mentioned flaws in our estimation procedure and some unclear sentences were re-analyzed and rephrased. This reanalysis substantially improved the parameter estimation for the physical interpretation of the relationship between recession parameters on environmental factors. Although the value of recession parameters and correlation coefficients were updated, the contrasting recession responses do not change. The details of the re-analysis and point-to-point reply were described below.

GENERAL COMMENTS
The main contribution of this paper is the identification of different responses of recession shape (as characterized by b) to antecedent moisture and the apparent connection of these different responses to landscape properties. This is an important finding and may help explain contrasting results from other studies. Ultimately, I think this work could and should be published. However, the paper would require major revisions as there is one serious issue and a couple of weaknesses, which I describe below.

**Reply:**

As the reviewer recognized, the major finding of our work is that landscape properties modify different responses of recession shape to the initial moisture. This finding might have

two important implications. One is that the landscape properties should be primarily examined (e.g. drainage area in our case) for determining the direction of recession response as assessing recession at a regional scale. Otherwise, the biased direction would lead to an controversial inference. Secondly, the influence of drainage area on contrasting recession responses needs further developments of a theoretical framework for physical interpretation.

The reviewer also pointed out one serious issue and a couple of weaknesses in our analysis. In this revision, we followed the reviewer's suggestions to improve our analysis procedure in order to make our results more concise and consistent with other studies. The details of the improvements were described below. All the comments are replied carefully and the unclear sentences were rephrased in order to elevate the scientific significance of our study.

1. The most serious issue in the paper is the faulty analysis of the parameter a. The fact is that no physical significance can be ascribed to changes in a when b also changes concurrently, therefore the paper's interpretation of the variability of a in this paper is flawed.

A problem arises from the units a, which change as b changes, as such the paper makes a nonsensical comparison of values of a with different units. One consequence of the scale dependence of a on b is that the reported differences in a among recession events is dependent, even in a relative sense, on the units the authors use for discharge Q. If this study were to use units other than mm hr-1, not only would the relative magnitudes of the differences change, so could the sign of the difference (while zero change is also possible given the correct units). If the reported differences in a had a physical significance, simply changing the units shouldn't change the physical interpretation. If in doubt, I suggest the authors redo some of their analysis after changing the units of streamflow from mm hr-1 to km hr-1 and nm hr-1 to see the effect.

This unit dependence of a on b is why, for example, Tashie et al. (2020b) and others fix b within a catchment and estimate a for each of the catchment's recessions. However, this does not solve the dilemma of comparing a across catchments where the catchments have different values of b.

I recommend the authors look to Dralle et al. (2015) and Biswal (2021) for further discussion on the relationship between the power law coefficients.

**Reply:** We are aware that parameter, *a*, is strongly affected by the unit and concurrently changes with b as fitting the power law equation to observations. In the original manuscript, we simply used runoff depth (mm, discharge normalized by drainage area to eliminate unit effect and keep the consistency among catchments). However, after the unit testing suggested by the reviewer, we found the relationships between *a* and *b* of our 260 cases are still unit dependent, strongly negative for nm, flat for cm, and strongly positive for both m and km for the unit of *a*. In this regard, using runoff depth (mm, the normalized discharge) is insufficient to eliminate the unit dependence between *a* and *b*, even though the relationships between *a* from different units and landscape indices remain unchanged. Therefore, we followed the reviewer's suggested references to re-analyze our cases.

Parameter dependency between *a* and *b* in recession analysis is inherent and entangled, which has no simple method to unravel. Biswal (2021) suggested to fix parameter *b* to obtain

parameter *a* with the same unit for the interpretation of the variability. But, fixed *b* method cannot examine variation in *b* among rainstorms. In a different manner, Dralle et al. (2015) used the corrected (â, b) pairs to interpret the variation in *b*. This method scaled the original flow $\hat{Q}$ by a constant *k*, so the flow had a new value $Q = k\hat{Q}$. The power law relationship could be rewritten as: $-dQ/dt = ak^{b-1}\hat{Q}^b$. Therefore, the fitted *â* is equal to $ak^{b-1}$, showing the correlation of â and $k^{b-1}$. Decorrelating â and $k^{b-1}$ can get a meaningful parameter *a* that is independent to *b*. Finally, rescaling $\hat{Q}$ by *a* value $q_0$ (ideally equal to 1/*k*) leads to *a* free to *b* and *â* = *a*. Since our study attempts to access both variation in *a* and *b*, the decorrelation method is appropriately applied to meaningful parameters, which is advantageous to compare *a* and *b* for different catchments. We added the decorrelation method in the revision for clarifying the calculation of corrected parameter estimation, in section 2.2.2 [Line: 169-173]: "*Secondly, varying units of â with b makes no physical meaning for comparison with other events or catchments. Since our target is to assess the response of dual parameters to rainfall, we used the decorrelation method (Drallet et al., 2015). This method assumes that the observed flow Q consists of a scale-free flow $\hat{Q}$ and a constant k (Q = k$\hat{Q}$). Thus, the power law formula can be rewritten as $-dQ/dt = ak^{b-1}\hat{Q}^b$, where a is scale-free recession coefficient [$h^{-1}$]. For correcting â to a, the observed flow Q was divided by a constant $Q_0$ (ideally equal to 1/k, see details in Drallet et al., 2015).*"

The decorrelated a and b actually changed Fig. 2-Fig. 8, and Table 2. The correlation coefficients between landscape indices to a and b were updated. The corresponding changes, including text in Result and Discussion are also updated synchronously.

2. Although the authors reference various papers that have empirically examined the relationships between the power law recession parameters and environmental factors, very much has been published on these topics that is not referenced including relatively recent work (e.g., Tashie et al. 2020b). The paper should have a more comprehensive summary of prior work, followed by a clearer statement of what is still poorly understood, and finishing with what this study proposed to do to address one or more outstanding questions. While the intro does this to some extent, it is not sufficient.

A very valuable contribution would be, possibly in tabular form, a list of those environmental factors considered along with the studies that have found positive/negative/no relationship between these factors with recession parameters. This would clearly illustrate how much has been done and, hopefully, demonstrate why yet another study of this type is still necessary.

Note that while I have referenced numerous papers in this review, they do not include many empirical studies of recession parameters.

**Reply:** Thanks for this constructive suggestion. Accordingly, a substantial modification was made in the revised introduction. We collected additional 11 empirical and 5 theorical papers involved power law recessions since 2013 and tabulated the relationship between recession parameters and various environmental factors in Table 1 (24 empirical works of power law recession). From this table, most of the studies focused on the relationship between catchment centrality of parameters and environmental settings. Although recent works have examined the temporal variability of recession, their work majorly studied on a seasonal scale, or focused on parameter *a*. In other words, we found two important, but unsolved questions

in our study. First, how do rainfall and physiographic variables affect recession parameters in different landscape regimes? So far, although several studies have explored the dependence of inter-event variably of recession parameters, their study sites located only in the USA. Thus, various responses among theories implying the control of landscape structure and rainfall amount on recession in different regions have room to be improved. Second, how does physiographic variables regulate the response of nonlinearity to rainfall? As we know, Biswal and Nagesh Kumar (2013) was the only work to find different responses of b to peak flow and interpret that the different responses are regulated by the subsurface storage gradient along a river. But what landscape variables control subsurface storage gradient is still unknown. We highlighted the two working hypotheses in paragraph 3 and 4 of the revised introduction [Line: 41- 59]:

> *"Theoretical works also illustrated the temporal dependence of recession parameters on the groundwater table, recharge, and storage. Parameter â has a negative correlation to the initial groundwater table ($h_0$) under unsaturated conditions and slightly positive under saturated conditions ($h_0 \geq Btan\phi$, where B is the aquifer length and $\phi$ is the aquifer angle, Rupp and Selker, 2006). A large recharge rate also reduces parameter â, particularly in homogenous catchments (Harman et al., 2009). On the other hand, hydraulic theories indicate that b decreases from 3.0 to 1.5 during the transition from early to late recession, as the groundwater is vertically sourced from different hydraulic properties in wet conditions (e.g., Rupp and Selker, 2006). The spatial heterogeneity theory demonstrates that the b only slightly increases with the wet antecedent condition (Harman et al., 2009). However, the drainage network theory indicates that b increases/decreases with storage while reaches in downstream are contributed by more/less subsurface storages (Biswal and Nagesh Kumar, 2013). The various responses among theories imply that the control of landscape structure and rainfall amount on recession in different regions should be explored further.*

> *However, in empirical studies, we would argue that while there have been numerous empirical works of the power law recession analysis (summarized in Table 1 and S1), little understanding has been established in different regions and scarce interpretation on dual parameters of inter-event studies. First, USA studies account for nearly three-fourths of the prior works; even all studies of the inter-event variability were on sites in the USA (Table S1), which may ignore other different recession behaviors. For example, empirical recession parameters have inconsistent responses to several physiographic variables (drainage area, drainage density, water bodies coverage, and surface saturated conductivity), implying that different landscape regimes may have distinct recession responses. Secondly, most inter-event studies just analyzed the single parameter (â) that decreases with the catchment wetness, which ignored the temporal variability of b. Only Biswal and Nagesh Kumar (2013) found that b may response to peak flow in different directions, but which landscape variables would control the direction is still unclear."*

3. The paper would greatly benefit from a discussion of what theory would predict for the influence of environmental factors on recession parameters. For examples, Figures 2 and 3 in Rupp and Selker (2006b) show how initial water table height (i.e., antecedent moisture), drainage density, hillslope slope and hillslope length/height ratio determine a and how vertical heterogeneity and initial conditions influence b. Are the results of this paper

consistent with theory? If not, why not? Perhaps theory breaks down outside of the idealized conditions upon which the theory is based? Theoretical work has also shown how planform shape and downstream boundary conditions (e.g., Troch et al. 2013) as well as a draining vadose zone (Luo et al., 2018) and drainage network geomorphology (Biswal and Marani 2010) can affect recession parameters.

**Reply:** Our original edition only took the spatial heterogeneity of flow velocity (Harman et al., 2009) to interpret the catchment variability of recession parameters. Indeed, including other theories could benefit the depth of discussion. In the hillslope hydraulic theory (Rupp and Selker, 2006), the shape of aquifer (length, depth, and gradient) can predict parameter *a*, which can be the analogy of our catchment-scale hillslope variables (flow-path length, height, gradient). The catchment-scale parameters (drainage area and total stream length) are the theorical predictor of *a*, but our results did not show the dependence. We suggest that the hillslope hydraulic theory can be used for interpreting the dependence of *a* on catchment-scale hillslope variables, but not for the actual catchment-scale variables. In the drainage network theory (Biswal and Marani 2010), parameter b can be predicted by the stream order law, but not for our cases.

Theories also show that rainfall/moisture among catchments would affect the responses of recession parameters. In hillslope hydraulics, the ratio of aquifer depth to groundwater table regulates the relationship between a and water table. While the initial water level is smaller than the aquifer depth, parameter a drastically decreases with the rising water table; while the water level is larger than the aquifer depth, parameter a is insensitive to the rising water table. Our results also showed this pattern. As for b, it is regulated by the vertical heterogeneity of hydraulic conductivity. Higher water table has more combinations of velocities in different storages, resulting in a larger b. In drainage network theory, the water table dominates extent of drainage network, controlling b increase or decrease with moisture. The different responses of b to rainfall are regulated by the subsurface storage contribution in each channel segment. Our empirical data showed that different responses of b to rainfall are related to the area, implying the area could be a proxy of subsurface storage change along the channel. The discussion has been updated as follows:

1.  Section 4.2.1, Line [299-306]: "*Outlining among theories, flow-path variables could be regarded as the aggregation of aquifers with various geometries, or vertical heterogeneity of aquifer (Rupp and Selker, 2006). Flow-path variables L, H, G can be the proxy of Bcosϕ, Bsinϕ+Dcosϕ, and D/B+tanϕ, respectively. Large B and tanϕ aquifers have a small coefficient a (Fig. 3 in Rupp and Selker, 2006). Catchments with long total stream length could have a large coefficient. While the flow-path has large H or G, its vertical heterogeneity of aquifer is probably high, also implying that a steep hillslope has a high n value (i.e., high vertical heterogeneity). Our inverse relationship between H and a confirms the theory (Rupp and Selker, 2006). Our catchment-scale flow-path variables could confirm the hillslope-scale aquifer geometry from the groundwater hydraulic theory.*"

2.  Section 4.2.2, Line [318-322]: "*Theoretical b increases with aquifer heterogeneity (Rupp and Selker, 2006), inter-hillslope heterogeneity (Harman et al., 2009), and the number of stream (Biswal and Marani, 2010), yet decreases with the total stream length (Biswal and Marani, 2010). As mentioned before, steep catchments may*

*lead to higher heterogeneity in aquifer and inter-hillslope, increasing the value of b. Also, the relationship between DD and b is consistent with the theories. However, these theories could not be valid in our large catchments, suggesting that prior theories were developed in hillslope or small catchment scale.*"

3. Section 4.3.1, Line [346-351]: "*This phenomenon can be explained by Figure 12 in Rupp and Selker (2006). While the initial water table $h_0$ < aquifer length × gradient (Btanϕ), parameter a is negative to the water table; when the initial water table > Btanϕ, parameter a is insensitive to the water table. Three types of catchments follow a pattern that parameter a drastically decreases until approx. 250 mm and changes to constant. Interestingly, Type C has a higher intercept in the rainfall-a relationship like the theorical curve of h0/D=1, suggesting that the lower H of type C tends to be saturation and have a quick recession.*"

4. Section 4.3.2, Line [367-370]: "*Notably, the contrasting response of b to rainfall was only found in Biswal and Nagesh Kumar (2013) that attributed the change in subsurface flow contribution along the channel to the response direction of b. Our empirical data showed that drainage area regulates different responses of b to rainfall, implying the area could be a proxy of subsurface flow contribution to the channel.*"

LINE-BY-LINE COMMENTS

L1: The title could be improved. First, the meaning of the word "swing" in this context is unclear. A pendulum swings. I don't think that is what the authors mean to say. Suggested replacements for "swing" are "modify" or "alter". Second, "the response of recession nonlinearity" is also unclear.
**Reply:** Yes, the two terms, "swing" and "response of recession nonlinearity", are unclear. We rephrased the title as: *Landscape structure regulates the contrasting responses of recession along rainfall amount*". It clearly elucidated that the direction of recession response would be altered by landscape structure.

L12. Is it 260 sets of recession parameters per catchment, or in total over all 19 catchments?
**Reply:** Yes, it is the total number over all 19 catchments. We rephrased as: "*We derived a total of 291 pairs of recession coefficient, a, and nonlinearity, b, from power-law recession (-dQ/dt = aQb) over all 19 subtropical catchments with a broad rainfall spectrum.*" in Line 12.

L29-31: This sentence is out of place and not particularly relevant. The previous sentences are about analyzing individual recession events, while this is about projections of future rare rainfall events. Unless a stronger link is made, I would delete this sentence.
**Reply:** We removed this irrelevant sentence.

L32: Define Q and t in -dQ/dt = aQb .
**Reply:** Revised as: "*A power-law relationship between streamflow declines (streamflow rate Q recesses with a timestep t) with streamflow rates (-dQ/dt = âQb) can describe the recession characteristics at the catchment scale (Brutsaert and Nieber, 1977).*" in Line 31.

L34. I have an issue with calling a the recession rate. The units of a vary with b, so are not

universally consistent with a "rate".

**Reply:** Yes, the unit of parameter *a* varies with flow and *b*, so "rate" is improper. We used *recession coefficient* in this revision. We checked this term and replaced recession rate with recession coefficient in the revision.

L38-41: It should be stated where discharge has been normalized by catchment area. From Figure 2, I take it that is has for the authors' analysis. The authors should be careful when discussing results from other studies that may not have normalized discharge. Brutsaert and Nieber (1977), for example, do not normalize discharge prior to comparing *a* from different basins with drainage density and network length but clearly dividing by area first would affect their values of *a*. Brutsaert and Nieber (1977) show an inverse relationship of *a* with total stream length (their Fig. 9) with seems to contradict the attribution to Bogaart et al. (2016) that a has a positive correlation with total stream length. Also, flow-path length and height need to be clearly defined.

**Reply:** Thanks for the reminder. In this revision, the discharge was normalized by drainage area and then used in the decorrelation method. The sentences, "*In this study, the stream discharge has been normalized by drainage area, and the unit of Q, â and b is [mm/h], [h$^{-1}$ (mm/h)$^{1-b}$] and [-], respectively.*", could be seen in Line: 115 to 116. Also, the comparison of a with other studies were carefully checked, seeing Table 1. The unit of *a* and discharge from each empirical study was marked. The recession responses to landscape indices were also indicated in Table 1. Thus, the consistency and contradiction from literature could be examined and discussed.

Finally, we defined *L* and *H and added the following sentences*, "*the flow-path is defined as the hillslope grid point following the surface flow direction toward the channel. Flow-path length (L) is the length of this path, and height (H) is the height difference along this path.*", in Line 89-91. We agree that unit of *a* may influence the response of *a* to environmental factors, other details are also important. In Brutsaert and Nieber (1977), the total stream length has a negative relation to *a* in the early part of recession and a positive relation in the late part of recession. As for Bogaart et al. (2016), they focused on the late part of recession and found a positive relationship between *a* and drainage density, which was not really contradict with Brutsaert and Nieber (1977). We put those into our discussion in this revision.

L42-43: Please cite the "few studies".

**Reply:** In this revision, we have included new references listed in Table 1. Event-scale studies account for one quarter of the previous works. Currently, "few studies" is no more a proper description. We replaced this sentence with the new one [Line: 59-60]: "*Only Biswal and Nagesh Kumar (2013) found that b may response to peak flow in different directions, but which landscape variables would control the direction is still unclear.*"

L46-47: I believe all the references except this one concern empirical studies, which makes this reference to theoretical work out of place. The statement is also unclear without more context. How does spatial heterogeneity affect whether increasing the steady-state recharge rate (I assume the authors mean a steady-state recharge rate immediately prior to the beginning of the recession) increases or reduces a? Maybe 1-2 paragraphs devoted to theoretical work could be included (see COMMENTS above).

**Reply:** Thanks to the reviewer, it's a good suggestion to include a paragraph focusing on theoretical work in introduction. We re-organized the introduction thoroughly. Now, the

introduction has five paragraphs. The second paragraph described a basic background of recession parameters and their controlling factors from recession theories. The third paragraph elucidated the changes of recession parameters with catchment moisture from theoretical perspective. The fourth paragraph mainly described less contributions in prior empirical studies (with the new compiled table, Table 1). The new paragraphs read as follows [in Line: 30-59]:

*"A power-law relationship between streamflow declines (streamflow rate Q recesses with a timestep t) with streamflow rates ($-dQ/dt = \hat{a}Q^b$) can describe the recession characteristics at the catchment scale (Brutsaert and Nieber, 1977). Parameter $\hat{a}$ is approximate to the recession rate but influenced by the unit of flow and b (see section 2.2.2), and parameter b represents the nonlinearity of storage. Recession parameters are often linked to the aquifer geometries, landscape, and spatial heterogeneity. Since the aquifer in various landscape units (e.g., hillslope, riparian, stream) exhibits different hydraulic properties, landscape structure, which presents the geometry of catchments and aggregates catchment hydraulic properties, apparently reflects various recession parameters. In theories, parameter $\hat{a}$ has a positive correlation with drainage density (total stream length/drainage area) and aquifer slopes but a negative correlation with aquifer depths, aquifer heterogeneity (of conductivity) (e.g., Brutsaert and Nieber, 1977; Rupp and Selker, 2006) and inter-hillslope heterogeneity (of celerity) (Harman et al., 2009). Parameter b increases with the number of streams (Biswal and Marani, 2010), the heterogeneity of the aquifer (Rupp and Selker, 2006) and the inter-hillslope (Harman et al., 2009), yet decrease with the total stream length (Biswal and Marani, 2010).*

*Theoretical works also illustrated the temporal dependence of recession parameters on the groundwater table, recharge, and storage. Parameter $\hat{a}$ has a negative correlation to the initial groundwater table ($h_0$) under unsaturated conditions and slightly positive under saturated conditions ($h_0 \geq B\tan\phi$, where B is aquifer length and $\phi$ is aquifer angle, Rupp and Selker, 2006). A large recharge rate also reduces parameter $\hat{a}$, particularly in homogenous catchments (Harman et al., 2009). On the other hand, hydraulic theories indicated that b decreases from 3.0 to 1.5 during the transition from early and late recession, as the groundwater is vertically sourced from different hydraulic properties in wet conditions (e.g., Rupp and Selker, 2006). The spatial heterogeneity theory demonstrated that the b only slightly increases with the wet antecedent condition (Harman et al., 2009). However, the drainage network theory indicated that b increases/decreases with storage while reaches in downstream are contributed by more/less subsurface storages (Biswal and Nagesh Kumar, 2013). The various responses among theories implying the control of landscape structure and rainfall amount on recession in different regions should be improved.*

*However, in empirical studies, we would argue that while there have been numerous empirical works of the power law recession analysis (summarized in Table 1 and S1), little understanding has been established in different regions and scarce interpretation on dual parameters of inter-event studies. First, USA studies account for nearly three-fourths of the prior works; even all studies of the inter-event variability were on sites in the USA (Table S1), which may ignore other different recession behaviors. For example, empirical recession parameters have inconsistent responses to several physiographic variables (drainage area, drainage density, water bodies coverage, and surface saturated conductivity), implying that different landscape regimes may have distinct recession responses. Secondly, most inter-*

*event studies just analyzed the single parameter (â) that decreases with the catchment wetness, which ignored the temporal variability of b. Only Biswal and Nagesh Kumar (2013) found that b may response to peak flow in different directions, but which landscape variables would control the direction is still unclear."*

L50: Start new paragraph at "Due to…".
**Reply:** The paragraph was rephrased in [L64-67]: *"Due to frequent tropical cyclones (alias: typhoon) and mountainous landscapes, Taiwan's rivers generally have short water travel time and limit water retention capacity in catchments (Lee et al., 2020). Most typhoon rainwater falls in summer and elevates water level dramatically but diminishes quickly within 2-3 days (Huang et al., 2012)."*, and in [L72-74]: *"Understanding the recession behaviors after typhoons are vital to water resource management, particularly when global warming likely increases the frequency and magnitude of flood and drought (Shiu et al., 2012; Huang et al., 2014)."*

L76-77: Clearer definitions of L, H, and G are needed. Table S1 gives definitions in the footnotes, but they do not appear to be consistent with what is in the main text. The text says "flow-path length [L]" but Tables S1 says L is total length of the drainage networks. Are they the same thing? If so, the text on line 40 is confusing because total stream length and flow-path length are treated there as if they are distinct measures. The text also says "gradient (G) above the nearest channel" but the Table S1 footnote says "G is the average gradient of the drainage networks". These do not sound like the same thing.
**Reply:** Sorry for the unclear descriptions of the landscape characteristics. In this revision, we described all definitions in the main text and supplementary materials. The sentences of definitions are now added as "*Flow-path length (L) is the length of this flow-path (a flow strip from divide to stream), and flow-path height (H) is the height difference along this path. G is the flow-path gradient [-].*", in [L90-91]. The footnote of Table S2 was revised: "*Here, H is the flow-path height [L], L is the flow-path length [L], G is the flow-path gradient [-], A is the drainage area [L$^2$], DD is the drainage density [L/L$^2$], S$_m$ is the gradient of mainstream, HI is the hypsometric integral [-], ELO is the basin elongation [-], CW, CF, CA is the land cover area of water, forest, and agriculture to total catchment area [-].*"

L86: Theissen polygons can be quite poor for interpolation of rainfall, particular in regions with sharp rainfall gradients such as in Taiwan. Is there additional information that can aid the interpolation, even a rainfall vs. elevation relationship? Are there any gridded climatologies (such as PRISM maps) that can be used to improve interpolation? What is the rain gauge density? Can the gauge locations be shown on a map?
**Reply:** We fully understand this issue and we had some experiences in the influence of spatial rainfall pattern to total flow and hydrograph (Huang et al., 2011, 2012). The grid-based rainfall (radar-based resolution $\cong$ 1.1 km) in Taiwan was available since 2002, while our events were derived from 1970s. Both the PRISM and the TRMM (resolution $\cong$ 5 km) also provided rainfall after 1990s, which do not meet the demand of this study. Due to the data limitation, only the rain gauges with sufficient historical records were used. In general, the rain gauge density in Taiwan is approximately 50 km$^2$ per gauge. Our previous studies showed that more dense gauges can describe the rainfall distribution, but the gauge requirement for total rainfall amount is relatively lower than for rainfall distribution

Huang, J.C., Kao, S.J., Lin, C.Y., Chang, P.L., Lee, T.Y., Li, M.H. (2011) Effect of subsampling

tropical cyclone rainfall on flood hydrograph response in a subtropical mountainous catchment, *Journal of Hydrology*, 409 (1-2): 248-261, doi: 10.1016/j.jhydrol. 2011.08.037.

Huang, J.C., Yu, C.K., Lee, J.Y., Cheng, L.W., Lee, T.Y., Kao, S.J. (2012) Linking typhoon tracks and spatial rainfall patterns for improving flood lead time predictions over a mesoscale mountainous watershed, *Water Resources Research*, 48: W09540, doi:10.1029/2011WR011508.

L104-106: Dralle et al. (2017) could also be cited as an example of an examination of the effects of methodological choices.

**Reply:** We cited this work as suggested. This work examined the influence of the method choice to parameter estimation, which is very convincing and suitable to cite here. [Line: 125-127]: "*Dralle et al. (2017) also agreed with the above statement but they found that the relationship between â and antecedent wetness are sensitive to length of data.*".

L127-L131: How many recession events were ultimately included per station? Were they the same events in time per station?

**Reply:** For clarification, we added a sentence to the end of this paragraph: "*Ultimately, each watershed had 5 to 26 events selected for analysis (see Table S3), of which events were not necessarily the same*". [Line: 176-178]

L134: How exactly were dQ/dt and Q estimated from the data?

**Reply:** We thoroughly re-wrote the method section. Now, it is: "*Instead, the exponential time step method (Roques et al., 2017) was applied here to reduce the bias, in which the time step of the moving window exponentially increases along the recession. Each point within the moving window was used for the computation of the (Q, -dQ/t) pair; Q is the average discharge, and -dQ/dt is the slope of linear regression.*" [Line: 166-169]

L145-146: This sentence is confusing. First, please explain what it means that "the two parameters are interactively dependent". Second, why are they "particularly" dependent "when the number of points is huge". Lastly, what does this dependence have to do the ordinary least squares method?

**Reply:** In the original version, we have recognized the dependence between a and b and so we stated that the two parameters are interactively dependent. Besides, with the increase of points (high probability to include extreme events), the regression slope would be strongly biased by extreme events. In this revision, the section of material and method was thoroughly revised. The original sentence has been removed.

L152: Define elongation (ELO). How is it calculated? This should be explained in the methods section.

**Reply:** In this revision, all landscape characteristics were clearly described. Elongation, the ratio of the diameter of circle (same area with basin) to the basin length, can be expressed as ELO = 2 $(A/\pi)^{0.5}/L_B$. We added it in [Line: 88-89]: "*ELO is the basin elongation [-] defined as the ratio of the diameter of the circle (same area with the basin) to basin length.*"

L152-153: If the properties of W8 and W18 are described here, then so should they be for W5. How is W5 distinct from W8 and W18?

**Reply:** We took the low flow correction and decorrelation method into account and reanalyzed the dataset as the reviewer suggested. In this revision, we found that the L/G is more significant than other landscape indexes. Thus, the three samples became W9, W5, and W8. All three catchments were described [Line: 183-184]: "*Catchment W9 has a larger A and lower L/G, W5 has a lower A and lower L/G, and W8 has a smaller A and higher L/G.*"

L153-154: It is meaningless to rank in descending order of *a* if all the *a* do not have the same units. I suggest ranking them in order of *b*.
**Reply:** Ranking them in order of b in this revision [Line: 184-185]: "*In descending order, the ranking of median recession b is catchment W9, W5, and W8.*"

L154-155: Why are the mean and median of b stated for W8 but not for the others?
**Reply:** As replied before, in this revision, the recession parameters of the three catchments were all described [Line: 184-186]: "*In descending order, the ranking of median recession b is catchment W9 (2.34), W5 (1.96), and W8 (1.63). The point-cloud derived b are 1.45 (W9), 1.37(W5), and 0.88(W8), showing all point-cloud b are smaller than median ones (Fig. 2c).*"

L155-156: This sentence is confusing and I wonder if there is an error. Why would the median > mean of the recession rate (a) imply that the distribution of nonlinearity (b) is right-skewed? Fig 2c doesn't actually give the mean of a.
**Reply:** It was our mistake. Recession coefficient (*a*) should be the nonlinearity (*b*). But due to the mean recession *b* was not discussed later, we demonstrated the point-cloud b here [Line 185-186]: "*The point-cloud derived b are 1.45 (W9), 1.37(W5), and 0.88(W8), showing all point-cloud b are smaller than median ones (Fig. 2c).*". Additionally, we would state the median a and b and point-cloud b in Fig 2c.

L156-157: A plot of *b* vs storm magnitude and/or $Q_{ini}$ for each of these three watersheds would be very helpful to illustrate the point being made, and show how strong these relationships actually are.
**Reply:** As suggested, we added plots of *b* vs Rainfall and inserted them into Figure 2c. It is a very useful and convincing suggestion. In the new inserted plot within Fig. 2c, it clearly showed the contrasting response of recession. Many thanks.

L157-158: I think the opposite response of W8 and W5 to storm magnitude being associated with differences in landscape properties should be left for the Discussion, but it is OK to foreshadow the discussion here. If this sentence is kept, I would follow it by saying that this apparent association will be explored further in the Discussion section.
**Reply:** Thanks for this suggestion. We added this suggested sentence, "*This apparent association will be explored further in the Discussion section*" in Line: 188.

L160: Units missing for a.
**Reply:** The unit of $\hat{a}$ is [hr$^{-1}$ (mm/hr)$^{1-b}$] and *a* is [hr$^{-1}$]. We added the unit here [Line 190]: "*Coefficient, a, ranges from 0.003 to 0.273 hr$^{-1}$ with mean = 0.058 hr$^{-1}$ and median = 0.047 hr$^{-1}$.*"

L167-170: Jachens et al. (2020) argue that this is not necessarily true; the point cloud fitting method may not reveal the "general" or average recession.
**Reply:** Yes, the term, "general", is not truly right. It just presents the bias description of

common recession [Line: 197-200].

L187: It should be noted here that P and Qini are effectively uncorrelated (r is only 0.11 and is not significantly different from zero). This seems like an important point and worth a little more discussion. How exactly is P calculated for each recession event? Over what time period is it totaled? This should be described in the methods.
**Reply:** We guess the reviewer may misunderstand $Q_{ini}$ (the initial flow or antecedent flow of rainfall event) for the flow at the begin of recession (i.e., $Q_p$ in our manuscript). Thus, we replaced $Q_{ini}$ with $Q_{ant}$ (antecedent flow). The rainfall period was defined as the elapse time from 6 hr before the rising flow to the peak flow. We described it in Methods [Line: 120-123].

L188: What are these "presumed thoughts". I would reiterate the thoughts here or leave this phrase out.
**Reply:** Re-think about it, we leave this phrase out.

L191: Describe how is L/G is a proxy for the interaction of landscape and climate. References?
**Reply:**
Rainfall-runoff is the main driver to shape the landscape and regulate landform evolution via erosion. Retrospectively, the geometry of landscape left by climate's watermark could identify the climate features (Seybold et al., 2017). In erosion, hillslope length and gradient are the key factors to erodibility, Therefore, L/G is a proxy for the interaction of landscape and climate. Moreover, McGuire et al. (2005) suggested that the flow path length and gradient distribution reflect the hydraulic driving force of catchment-scale transport (i.e. Darcy's law) and thus some description of topography (e.g. L, G, or L/G) provides a first-order control on flow processes and water residence time. We put some of the above descriptions in Line: 210-214.

Seybold, H., Rothman, D. H., & Kirchner, J. W. (2017). Climate's watermark in the geometry of stream networks. Geophysical Research Letters, 44(5), 2272-2280.

L214: Circular logic. The authors' definition of non-linearity is already that the value of b is not equal to 1. Also, values less than 1 are non-linear.
**Reply:** We believe that the reviewer's comments are about the sentence, "*Nonlinearity higher than 1.0 indicated….*", in the original manuscript, Line: 219. We rephrased it as, "*Non-linear storage-outflow relationship (b is not equal to 1.0) is prevalent for most catchments worldwide*". [Line: 234]

L220: It would be clearer to simply say highest and lowest values of b. Or "the most and least nonlinear cases are…".
**Reply:**   As suggested, we used "*the highest and lowest median values of b*" instead. [Line: 250]

L230-232: But see also Sharma and Biswal (2022).
**Reply:**   Thanks for suggesting this paper. This comment is similar to the previous one. We added a sentence, "*Notably, there is no single value of b preferable for all practical purposes.*" [Line: 260-262]

L238: "Complicates" is not a good term here. I think the authors mean to say that heterogeneity may increase with catchment area because of the possibility of including a wider range of subsurface conditions. This sentence should be rewritten for clarity.
**Reply:** We used the reviewer's sentence in Line: 267. Thanks.

L259: This is the first time any "Type" of catchment is mentioned. The classification of catchments into Types needs to be introduced more clearly. I would start a new paragraph and direct the reader to the upper half of Fig 9 (which will mean reordering the figures).
**Reply:** We reorganized the paragraph according this comment. Now, it is, "*According to Fig. 6b, all catchments could be simply classified into three types. Type A is large catchments (area > 500 km$^2$), B is small catchments with low L/G ratio, and C is small catchments with high L/G ratio.*" [Line: 291-293]

L271: I suggest rewriting as "The positive relationship of b with both H and L…."
**Reply:** Thanks for the rewording. The sentence has been rephrased as you wrote in Line 321-322: "*The positive relationship of b with H and L indicates that steeper and rougher hillslope present non-linear recession behavior.*"

L272: I don't understand this sentence. What are these "blocks"? How does higher H and L imply greater prevalence of such blocks?
**Reply:** Rephrased as, "*The positive relationship of b with H and L indicates that steeper and rougher hillslope present non-linear recession behavior. A possible interpretation is that with the increase of flow path, subsurface runoff has more chances flowing through various blocks (e.g. temporarily perched groundwater)*". [Line: 321-323]

L273-274: This is an important idea the authors introduce. Do only catchments with short AND gentle hillslopes have large riparian areas? How exactly does a larger riparian area reduce heterogeneity?
**Reply:** Rephrased as, "*Short-and-gentle hillslopes, which means their topographic wetness indices vary smoothly. The smooth distribution of topographic wetness indices would present the gentle expansion of saturation area during rainstorms from the perspective of TOPMODEL (Huang et al., 2009). The expansion of saturation area indicates the whole subsurface is getting saturated and connected and thus reduces heterogeneity.*" [Line: 323-324]

Huang, J.-C., Lee, T.-Y., and Kao, S.-J.: Simulating typhoon-induced storm hydrographs in subtropical mountainous watershed: an integrated 3-layer TOPMODEL, Hydrol. Earth Syst. Sci., 13, 27–40, https://doi.org/10.5194/hess-13-27-2009, 2009.

L276-277: Do these large basins have something in common other than being large? Almost all these large basins have their headwaters at the highest elevations and most of the smaller basin are on the west side. These smaller basins are also mostly in the rainshadow, whereas the larger basins receive much more rainfall. What role could these factors play?
**Reply:** Reviewer is right that most orographic, conventional, and frontal rainfall are strongly affected by landscape and form rainshadow. It's another interesting issue. But, our dataset with limited spatial resolution in rainfall can't support to test this hypothesis. Notably, typhoon, alias of tropical cyclone in Pacific Asia, has quick moving velocity with different trajectories. Moreover, it rotates counterclockwise quickly (depends on pressure gradient). In

this context, the rainshadow regions vary dynamically. We can't exclude the effect of rainfall distribution on recession raised by reviewer. This comment also likely interprets why the recession response to rainfall in large catchments are more non-linear. We put above descriptions into the revision [Section 4.3.2].

L279: "The large deviation" of what? Please be explicit.
**Reply:** Large deviation in the value of *a*. [Line: 338]

L303-314: I think it worth noting that *b* is treated as a constant here throughout a single recession event, though it has been empirically shown that it can change over the course of an event (e.g., Rupp and Selker 2006b; Tashie et al. 2020a). Also, groundwater hydraulic theory predicts that *b* can change over time (Brutseart and Nieber 1977). For a horizontal aquifer, this change depends on the initial conditions (Rupp and Selker 2006a) but the change in *b* happens relatively early and *b* becomes relatively steady as the recession progresses. In a sloping aquifer and/or one that is vertically compartmentalized, *b* can change over a lengthy part of the recession (e.g., Bogaart et al. 2013; Roques et al. 2022). How this theoretical idealized hillslope behavior might manifest in a complex catchment has still not been not well-described, however.
**Reply**: Yes, *b* might be time-variant during an event, since the saturation degree and hydraulic connectivity vary dynamically. We added that *b* is constant through a single recession event, in Line 378-379: "*Note that although b was found it can change over the course of an event* (Luo et al., 2018; Rupp and Selker 2006), *this study treated b as a constant and the inter-event variability is discussed as the following.*"

L306: This is an interesting idea that the pervasive saturated overland flow reduces the nonlinearity of recession. Two issues: 1) What field evidence is there of this pervasive saturated area? 2) I expect this saturated area is decreasing in time, possible very quickly. How would this affect b? Thinking along the geomorphological lines of Biswal and Marani (2010), might this not increase b?
**Reply:**
The first issue has been replied in previous ones [L273-274] and [L326]. We don't have comprehensive field evidence, but do have some local experience. The second issue is also not easily replied. Yes, the saturated area decreases in time. But, our recession *b* was treated as a constant during an event; in other words, the saturated area that we indicated is at the beginning of recession (i.e., peak flow). Large saturated area, like more water bodies, would behave like a linear reservoir, resulting in a smaller *b*. Although the geomorphological lines of Biswal and Marani (2010) implied that *b* increase with the extent of drainage network (i.e., large rainfall), the uniform flow contributions along river are often not meet in real systems. Their revised model (Biswal and Nagesh Kumar, 2013) stated only large flow contribution in downstream could meet the positive relationship between *b* and peak flow.

L309-311: I'm not sure I follow this. How would a large rainstorm "connect" saturated zones of slow reservoirs that were otherwise not draining to a stream? What may be happening is that during large storms there is a wider range of active quickly to slowly draining sources (the fast ones being the ones activated during the large storms). This heterogeneity of sources can increase b. This appears to be what the authors say in the sentences following this one.

**Reply:** Thanks for making it more clearly. What we want to say is that with the connection of saturated zones, the large storms can activate different draining sources, mixing them downstream and result in the decrease of b (as Type C demonstrated). The above descriptions have been updated.

L326: Is there any field evidence for these perched storages?
**Reply:** Although there is no comprehensive observation on a larger scale, an experimental forested watershed in northern Taiwan was observed having perched subsurface water bodies (Liang, 2020). With the increasing rainfall, heterogenous subsurface saturations might be activated to contribute into the stream. We added a sentence: "*The existence of perched storages have been found in an experimental forested catchment in Taiwan by intensive pore water monitoring (Liang, 2020).*" [Line: 360]

Liang, W.-L. (2020). Hydrological responses in a natural forested headwater before and after subsurface displacement. Journal of Hydrology, 591: 125529. https://doi.org/10.1016/j.jhydrol.2020.125529.

L329: I wouldn't say "unpredictable". Predictability is not the issue here.
**Reply:** Eliminated.

L340: I would not say "pretty diverse". Is "inconsistent" what is meant?
**Reply:** As suggested, we replaced "*pretty diverse*" with "*inconsistent*" in Line: 370.

L346: What is meant by "higher hillslope hydraulics"?
**Reply:** Revised as "*higher L/G*". Other terms of hillslope hydraulics were also replaced with L/G.

Figure 2: The panels in column c clearer show discretization artifacts that visually hide the underlying relationship at low flows. A way to remove these artifacts was first proposed by Rupp and Selker (2006b) and modifications were made by Roques et al. (2017) and Guo et al. (2022). I suggest the authors apply one of these methods.
**Reply:** Thanks for this suggestion to discretize artifacts during low flow. For improving the estimations of recession parameters, in this revision, we applied the exponential time step method (Roques et al., 2017) to remove the discretization. We added the above descriptions in the section of material and method to clarify our estimation procedure [Line: 110-115].

Figure 5: State in caption whether all stations and all events are shown in this plot.
**Reply:** Now the sentence was phrased as, "*Recession parameter a and b from all catchment-events against landscape variables.*" in the caption of Fig. 5.

Figure 7: Say in caption which symbols are for Type A, B, and C, basin.
**Reply:** We added the descriptions of Type A, B, and C in Fig. 7. The sentence, "*Type A is large catchments (area > 500 km$^2$), B is small catchments with low L/G ratio, and C is small catchments with high L/G ratio.*" was added in the caption.

Table S1: Some of these basin average drainage network gradients are very large (as high as 0.75). Hillslope gradients must be yet larger. What are the implications for subsurface flow?

Table S1: L/G is given as having units of m2. If L has units of m and G is unitless, L/G must have units of m.

**Reply:** Sorry for the confusion. *G* is the flow-path gradient [-], not the average drainage network gradient. Therefore, the unit of *L/G* is [L]. We updated the definitions of *L* and *G* as, "L is the flow-path length [L], G is the flow-path gradient [-]" in the revised Table S1.

---

## Author Comment (AC2)

**Reply on Reviewer Comment #2**

This overall well written paper intends to relate the classical a and b recession parameters to stream network, rainfall and antecedent moisture conditions. As discussed in the review by Anonymous Referee #1, there is a methodological problem: the presented analysis investigates the relationship between the marginal distributions of the parameters and possible explanatory variables, i.e. the analysis omits that a and be are not independent; a solution would be to first model the relation between a and b but as far as I see from fig. 5, there is no evident relationship between a and b.

**Reply:**

We regret that Referee #2 did not see the true merits of our efforts behind the presentation with certain flaws. Referee #1 pointed out one methodological issue and a couple of weaknesses. The methodological issue is solvable. The weaknesses are basically suggestions to deepen the discussion in this revision. So, through Referee #1's insightful review and our thorough endeavor the significance of our study has been better manifested.

The goal of our study attempts to present landscape structure that can regulate the contrasting recession responses along rainfall amounts. Therefore, is there a simple and obvious relationship between a, b and landscape variables (Fig. 5), which is actually the question we are trying to explore. There have been many studies which pointed out the relationship between landscape and recession variables, but why can't we sort out a simple relation in our cases? After classifying the catchments via drainage area and L/G, we found that the catchments present clear contrasting responses (Fig. 8). Our findings might have two important implications. One is that the landscape properties should be primarily examined (e.g. drainage area in our case) for determining the direction of recession response as assessing recession at a regional scale. Otherwise, the biased direction would lead to an opposite inference. Secondly, the influence of drainage area on contrasting recession responses needs developments of a theoretical framework for physical interpretation.

We are aware that parameter, a, is strongly affected by the unit and concurrently changes with b as fitting the power law equation to observations. In this revision, we used the "decorrelation" method (Dralle et al., 2015) to resolve the independence between a and b (see revised section 2.2.2 [Line: 169-173]): "_Secondly, varying units of â with b make no physical meaning for comparison with other events or catchments. Since our target is to assess the response of dual parameters to rainfall, we used the decorrelation method (Drallet et al., 2015). This method assumes that observed flow Q consist of a scale-free flow $\hat{Q}$ and a constant k ($Q = k\hat{Q}$). Thus, the power law formula can be rewritten as $-dQ/dt = ak^{b-1}\hat{Q}^b$, where a is scale-free recession coefficient [$h^{-1}$]. For correcting â to a, the observed flow Q was divided by a constant $Q_0$ (ideally equal to 1/k, see detail in Drallet et al., 2015)._"

Besides, it is unclear what the main contribution of the paper is beyond a state-of-the-art case study (which is probably no enough to justify publication in HESS). A clear presentation of what we could learn from a case study in the selected hydroclimatic area would be of key importance. The paper would also strongly benefit from a concise synthesis of known factors influencing recession properties

and a better justified selection of the potential explanatory variables that are retained.

For all above reasons, I suggest rejecting the paper.

**Reply:** The main contribution of our study is to propose a hypothesis that the degree of landscape heterogeneity regulate the contrasting recession responses. Additionally, we identified that different responses of recession shape (parameter, a and b) to rainfall amounts appealing the connection of these different responses to landscape properties. This finding will bridge the gap between conceptual-physical model (Biswal and Marani, 2010 and 2013) to practical application.

Detailed comments:

There is a lack of references for the theoretical aspects of how recession properties depend on landscape properties

**Reply:** We included several theoretical papers in this revision, including aquifer/hillslope geometry, vertical heterogeneity of aquifer (Rupp and Selker, 2006), draining vadose zone (Luo et al., 2018), drainage network (Biswal and Marani, 2010), and inter-hillslope heterogeneity of celerity (Harman et al., 2009). In our paper, the drainage area and the ratio of flow path length to gradient are the most important landscape variables, which was discussed with the above theories. The discussion has been updated as follows:

1. Section 4.2.1, Line [299-306]: "*Outlining among theories, flow-path variables could be regarded as the aggregation of aquifers with various geometries, or vertical heterogeneity of aquifer (Rupp and Selker, 2006). Flow-path variables L, H, G can be the proxy of $B\cos\phi$, $B\sin\phi+D\cos\phi$, and $D/B+\tan\phi$, respectively. Large B and $\tan\phi$ aquifers have a small coefficient a (Fig. 3 in Rupp and Selker, 2006). Catchments with long total stream length could have a large coefficient. While the flow-path has large H or G, its vertical heterogeneity of aquifer is probably high, also implying that a steep hillslope has a high n value (i.e., high vertical heterogeneity). Our inverse relationship between H and a confirms the theory (Rupp and Selker, 2006). Our catchment-scale flow-path variables could confirm the hillslope-scale aquifer geometry from the groundwater hydraulic theory.*"

2. Section 4.2.2, Line [318-322]: "*Theoretical b increases with aquifer heterogeneity (Rupp and Selker, 2006), inter-hillslope heterogeneity (Harman et al., 2009), and the number of stream (Biswal and Marani, 2010), yet decreases with the total stream length (Biswal and Marani, 2010).  As mentioned before, steep catchments may lead to higher heterogeneity in aquifer and inter-hillslope, increasing the value of b. Also, the relationship between DD and b is consistent with the theories. However, these theories could not be valid in our large catchments, suggesting that prior theories were developed in hillslope or small catchment scale.*"

3. Section 4.3.1, Line [346-351]: "*This phenomenon can be explained by Figure 12 in Rupp and Selker (2006). While the initial water table $h_0 <$ aquifer length × gradient ($B\tan\phi$), parameter a is negative to the water table; when the initial water table $> B\tan\phi$, parameter a is insensitive to the water table.*"

*Three types of catchments follow a pattern that parameter a drastically decreases until approx. 250 mm and changes to constant. Interestingly, Type C has a higher intercept in the rainfall-a relationship like the theorical curve of h0/D=1, suggesting that the lower H of type C tends to be saturation and have a quick recession.*"

4. Section 4.3.2, Line [367-370]: "*Notably, the contrasting response of b to rainfall was only found in Biswal and Nagesh Kumar (2013) that attributed the change in subsurface flow contribution along the channel to the response direction of b. Our empirical data showed that drainage area regulates different responses of b to rainfall, implying the area could be a proxy of subsurface flow contribution to the channel.*"

There is no discussion of active drainage density (the actual drainage network can vary strongly seasonally)
**Reply:** We have included the drainage network theory in this revision. This theory states that the recession parameter b is positive to the number of stream and negative to the total stream length (Biswal and Marani, 2010). In their revised model (Biswal and Nagesh Kumar, 2013), the Strahler stream order number was included; they used the bifurcation ratio and length ratio replace the original ones. In temporal variation, they attributed the response of b to the difference of flow contribution in various order stream. In our case, drainage area might be the apparent landscape variable for the difference of flow contribution. We updated our introduction [Line 41] and discussion [Line 367-370].

the literature review should be improved; the previous findings are summarized but not yet synthesized; we also do not know where the previous work has been done (catchments size, climate, region etc); is this study the first in a tropical area?
**Reply:** As suggested, we compiled recent recession studies into Table 1 of this revision. We collected additional 11 empirical and 5 theorical papers involved power law recessions since 2013 and tabulated the relationship between recession parameters and various environmental factors. From this table, most of the studies focus on the relationship between catchment centrality of parameters and environmental settings. All six inter-event variability studies are located in the USA, which means other landscape regimes have not been surveyed. As we know, only Biswal and Nagesh Kumar (2013) found the contrasting response of b to rainfall. They explained this contrasting response by the gradient of subsurface storage along the channel. Thus, our paper aims to (1) explore environment factor under different landscape regimes; and (2) investigate how landscape variables can explain the contrasting response of b to rainfall. Besides, this study is indeed the first study that explored the influence of environmental factors to the inter-event variability of recession in subtropical/tropical catchments

When talking about travel times, it is important to be more specific weather this is in the channeled or the unchanneled state (i.e. in-stream or in the hillslopes), (e.g. Rinaldo et al., 2006)

**Reply:** The travel time of our previous work is defined as the time water traveling through a control volume (i.e., catchment) from the sky to the outlet. Yes, specifying travel time within different geomorphic states (i.e., hillslope or channel) may be greatly beneficial for understanding rainfall patterns controlling travel time distributions (Rinaldo et al., 2006). This comment is important for travel time studies, whereas it is not much relevant to our manuscript.

1: attention some units are wrong, the same units should be on both sides of the equation
**Reply:** Thanks for reminding. We checked the units in equations and text.

There are not enough details on how the explanatory variables of Table 1 are computed for the 260 events (what is total precip, what is Qtot (including or excluding baseflow?), how is peak flow identified if there are several peaks etc. etc.)
**Reply:** In this revision, the clear information of all variables is replenished in Table 2. Besides, we also added the descriptions associated with calculation for all variables for clarification [in Sect. 2.1]: "$AP_{7day}$ (mm) is antecedent 7-day rainfall. D (hr) is the duration of rainfall event defined as the elapse time from 6 hr before the rising flow to the peak flow. P (mm) is the total rainfall during rainfall event. $I_{avg}$ (mm hr$^{-1}$) is the total rainfall divided by duration of rainfall event. $Q_{tot}$ (mm) is total discharge (including baseflow) during the rising flow to the end of recession. $Q_{ant}$ (mm) is antecedent discharge at before 1h of rising flow. $Q_p$ (mm) is the maximum discharge over the rainfall event. $Q_{tot}$/P (-) is total discharge divided by total rainfall."

---

## Referee Report (RR1)

[referee-annotated manuscript omitted]

---

## Author Response (ED1)

**Reply to Editor's Comment**

Dear Authors,

both reviewers agree that you have significantly improved the manuscript. I am glad to see that!

However, there are still some issues concerning the wording/language as well as some other issues which should be addressed in another round of (minor) revisions. One of the reviewers has provided detailed comments and suggestions but please also check for wording issues beyond these comments. In the abstract you refer to a resulting bias, it might be necessary to provide some clarification here - what type of bias are you talking about?

Please revise your manuscript according to the reviewer's suggestions.

Wishing you all the best and looking forward to your revised manuscript,

Theresa Blume

**Reply:**

Dear Theresa,

   Thank you for your email and for sharing the feedback from the reviewers. We are delighted that both reviewers acknowledge the significant improvements we have made to the manuscript.

   We greatly appreciate the invaluable suggestions provided by the two reviewers during the three rounds of review. We are particularly grateful for Reviewer 1's meticulous editing recommendations. In this revision, we have diligently incorporated Reviewer 1's suggestions and added a new Table S3 for describing the definition and calculation of landscape and rainstorm variables. Also, we conducted a comprehensive review of the vocabulary and grammar throughout the entire manuscript.

   In relation to your mention of "bias" in the abstract, we have taken steps to address this concern and provide further clarification. Specifically, we have revised the sentence [L18-20] as follows: "Without considering this contrasting response, which is contingent upon landscape structure, it leads to a misjudgment of the recession nonlinearity in response to rainfall amount and needs further clarification, particularly for use in assessing regional recession in ungauged catchments under climate change."

   Many thanks for your patience and handling. If any further questions, please tell us for corrections.

Best regards,

Jr-Chuan (River) Huang

**Reply to Reviewer 1 Comments**

GENERAL COMMENTS

The third version of this paper, while improved, still contains some unclear statements, and requires additional editing. I provided some editorial suggestions but did not attempt to be thorough.

I recommend this paper be returned to the authors for minor revisions.

**Reply:** We appreciate the reviewer for providing such constructive and editorial suggestions. We made point-to-point modifications based on these suggestions. In this review, Reviewer #1 raised some comments associated with the definition and calculation of landscape variables. Therefore, we added a new Table S3 in supplementary for keeping the manuscript concise. At the same time, we reconfirmed the tenses and grammar of all the verbs in this text.

LINE-BY-LINE COMMENTS

9-19: The abstract does not sufficiently summarize the study and the significance of the results. It should, for one, give the physical significance of the relationships between the recession parameters (a, b) and the various factors (e.g., L/G, catchment area). For example, it is not enough to only state that the value of b decreases as L/G increases. What is the significance of L/G? What does it represent with regards to what influences the flow of water through the catchment What does it imply in terms of subsurface flow that b decreases as L/G increases?

**Reply:** The landscape structures can lead to the contrasting response of recession is what we want to highlight. In this regard, we replaced the L/G with the landscape structure to keep the abstract concise. Meanwhile, we understand this comment raised by the reviewer and admit that the significance of L/G to recession is quite important. Therefore, we addressed it in L81-88 and added a new Table S3 in supplementary for describing the definition and calculation of landscape and rainstorm variables.

15-16: Flow-path length and gradient should be defined. Subsurface flow path length? River channel flow path length? Gradient of the river channel? Mean gradient of the catchment? Mean hillslope gradient?

**Reply:** As mentioned, we removed L/G from the abstract, but addressed the explanation of L/G in L81-88. Table S3 in the Supplementary clarified the definition

and calculation of the variables.

17: Non-linearity of what? "Non-linearity" needs to be defined.
**Reply:** Rephrased as "recession nonlinearity" for clarification [L17].

21-33: The English usage in the first two paragraphs is often poor and the information is presented in a manner that is difficult to follow. These paragraphs provide very important context for the authors' study so I think more care should be given towards clearly relaying the information. Currently, it reads as a somewhat unorganized list of results from 4 – 5 studies. Separate paragraphs for effects of landscape structure and rainfall/antecedent storage conditions might help. Separate paragraphs discussing the recession coefficient and non-linearity might also help.
**Reply:** Based on the influences of topography and rainfall/antecedent moisture conditions, we reorganized this section into two paragraphs as suggested [L29-37] and [L38-46]:

Paragraph 1: "Since aquifers in various landscape units (e.g., hillslopes, riparian areas, streams, etc.) exhibit different hydraulic properties, theoretical works have shown that the streamflow recession parameters depend on the landscape structure or aquifer properties. Specifically, from the aspect of aquifer hydraulics (Rupp and Selker, 2006), spatial heterogeneity (Harman et al., 2009) and drainage network (Biswal and Marani, 2010) have been observed that these recession parameters are influenced by the aforementioned factors. In general, parameter $\hat{a}$ shows a positive correlation with stream length and aquifer slope (Rupp and Selker, 2006), while it exhibits a negative correlation with drainage area, aquifer depth, aquifer heterogeneity (Rupp and Selker, 2006), and inter-hillslope heterogeneity (Harman et al., 2009). On the other hand, parameter $b$ tends to increase with the number of streams (Biswal and Marani, 2010), aquifer heterogeneity (Rupp and Selker, 2006), and inter-hillslope heterogeneity (Harman et al., 2009), whereas it decreases with the total stream length (Biswal and Marani, 2010)."

Paragraph 2: "Additionally, theoretical works have shown that the dependence of streamflow recession parameters on antecedent storage or rainstorms. Parameter $\hat{a}$ is negatively correlated with the recharge rate (Harman et al., 2009), the streamflow rate (Biswal and Nagesh Kumar, 2014), and initial groundwater table under unsaturated conditions (Rupp and Selker, 2006), while it has a slightly positive correlation under saturated conditions (Rupp and Selker, 2006). For parameter $b$, it slightly increases with a wet antecedent condition (Harman et al., 2009). However, drainage network theory indicates that $b$ increases with peak flow while the downstream receives more subsurface flow contribution but decreases with peak flow as the downstream

receives less (Biswal and Nagesh Kumar, 2013). The inconsistent responses in $\hat{a}$ and $b$ among theories indicate a complicated interaction between landscape structure and rainstorms during recession, implying that the recession mechanics in different regions need more exploration."

25: That "the recession coefficient, a, approximates the recession rate" is not a generally true statement.
**Reply:** Thank you. We removed the clause, "approximates the recession rate" [L27].

27-33: Which of these relationships were determined from theory and which were shown empirically?
**Reply:** Replied above.

51: Not yet clear at this point what "point-cloud" means. Please define.
**Reply:** We added the clause, "a collection of multiple recession curves" for clarification [L50].

51-54: I don't think it is correct to say that recession responses are dependent on "a" and "b". "a" and "b" are a convenient way for us to characterize recession curves. Real-world behavior does not depend on these artificial values.
**Reply:** The reviewer is right. We fully agree with this suggestion. We have revised the sentence as [L52-53]: "Fewer studies simultaneously addressed recession responses to landscape structure and distinct rainstorm events."

54-55: This statement appears to be very relevant to the authors' study. The authors should elaborate: Why do Biswal and Nagesh Kumar conclude that "b" may respond to peak flow differently depending on the structure of the drainage networks?
**Reply:** Thank you. We added a sentence in [L54-55]: "However, they did not specifically identify which landscape characteristics would predominantly influence the directional switch in the response of parameter b to rainfall."

80-83: It is not perfectly clear to me what exactly the flow-path length (L) is, which makes it harder for me to assess the physical significance of L/G. Is L the length of the hillslope, as in the distance from a drainage divide to a stream? How is it calculated for an entire catchment?
**Reply:** This comment and the following four are highly relevant with L/G ratio. Thus, we replied all comments and all corresponding modification in the revision. The flow path is defined as the hillslope grid point following the surface flow direction toward

channel. The revised sentences were: "The flow path is defined as the hillslope grid point following the surface flow direction toward the channel (see detail in Tetzlaff et al., 2009). Specifically, flow-path length ($L$) is the length of this path, flow-path height ($H$) is the height difference along this path, and $G$ is the flow-path gradient [-]. Therefore, each grid cell can have its own $L$, $H$ and $G$. The median value of these flow-path metrics in a watershed was calculated as the representative value for the catchment. Among them, the composite ratio of $L/G$, which represent the distance effect of flow-path under different gradient holds hydrologic significance as it can serve as a proxy for water residence time (McGuire et al., 2005; Tetzlaff et al., 2009). Therefore, these flow-path metrics are widely used as proxies for understanding the interaction between landscape and climate (Seybold et al., 2017). The detail definition and calculation of the flow-path associated variables are illustrated in Table S3 in supplementary." in L80-88.

Is G = H/L? Is L/G, therefore, simply L/(H/L) = L^2/H? If so, what does L^2/H say about landscape structure that simply G (or 1/G = L/H) alone does not?

**Reply:** Replied above. The composite $L/G$ ratio represents the distance effect of flow-path under different gradient. Both distance and gradient are highly relevant with water flow. In fact, the $H$, $L$, $G$, and $L/G$ are significantly correlated to a (see Table 2). However, L/G is the best one to classify the catchment types (see Figure 6).

Given how much of the discussion on the results centers on L/G, it is important that its geomorphic/hydrologic/hydraulic significance be stated.
**Reply:** Replied.

How does a large value of L/G imply a "short-and-gentle" hillslope (as stated on line 364)? I understand that the "gentle" part comes from a low value of G leading to a high value of L/G. However, a "short" hillslope would have a lower L, which leads to lower value of L/G.
**Reply:** As reviewer mentioned, large $L/G$ can be result from a large $L$ or gentle $G$. However, in Taiwan, the median value of L among the catchments are quite small (see Table S2). Most median L values less than 300m indicate the length of flow paths in Taiwan is relatively short.

107: Actually, "b" is the slope in a plot of log(-dQ/dt) vs log(Q).
**Reply:** Thank you, we revised according to the suggestions [L112].

179: "This contrasting response coincided with a difference in drainage area and was relatively consistent across all catchments". I don't fully understand this sentence. What exactly was consistent across all catchments?

**Reply:** We rephrased in [L183-184]: "The contradictory responses observed in these three catchments can be attributed to variations in their landscape structure and rainstorm characteristics."

192-193: Why make a distinction for catchments with drainage area larger than 800 km2?

**Reply:** We observed that when the drainage area larger than 800 km$^2$, the point-cloud-derived coefficients become similar to the third quantile of the coefficient distribution from individual segments [L197].

200-201: The Methods section should describe how each of these variables was calculated.

**Reply:** We added a new table, Table S3, which described the calculation methods for these variables. Additionally, we updated the numbering of the supplementary tables throughout the entire manuscript.

206-207: What might cause the opposite response of "b" to antecedent flow and peak flow? This seems to be an inconsistent result if both factors are related to storage at the beginning of the recession event. This topic should be revisited in Section 4.3.2.

**Reply:** We have followed the reviewer's suggestion and added content to section 4.3.2 in L325-328: "In our study, we observed an increase in recession nonlinearity with antecedent flows but a decrease with peak flow. This phenomenon can be attributed to the superimposition of recession events on antecedent flows, which amplifies the value of $b$ (Jachens et al., 2020). The negative correlation between $b$ and peak flow does not necessarily imply a consistent response across all catchments."

228: I would rephrase this to say the point-cloud estimates are distinctly different from the estimates from the individual recessions. Both may have systematic biases since we don't know the "true" values.

**Reply:** We used the reviewer's sentence "Notably, the point-cloud estimates are distinctly different from the estimates from the individual recessions."[L233]

228-230: I don't think it is a given that a skewed distribution of flood peaks is the primary cause. Jachens et al. (2020) provide an example where the distribution of

peaks is not skewed and the point-cloud value of b is still less than those of the individual events.

**Reply:** We agree with the reviewer's comment. We rephrased as "The larger *a* and smaller *b* values derived from the point-cloud  from individual segments could be expected due to the influence of antecedent flow and superimposition of recession events (Jachens et al., 2020)." [L233-235]

235: Sharma and Biswal (2022) is missing from the References section.
**Reply:** Include in the reference [L476].

237-239: I don't think the authors' ranges of the recession coefficient can be compared directly to other studies because the units are not the same. Moreover, if the other studies did not apply the decorrelation method, the comparison may not be valid.
**Reply:** Thank you for your feedback. We removed the related sentences.

253: Roques et al. (2022) is missing from the References section.
**Reply:** Included in the reference section [L465].

281-284: I do not follow the authors' logic here. A high value of H does not necessarily lead to slower drainage. On the contrary, a steep hillslope implies a stronger hydraulic gradient and faster drainage.
**Reply:** We rephrased as [L280-282]: "Flow-path height, *H*, does not necessarily correspond to hydraulic gradients due to the geologic and soil setting in different regions (Karlsen et al., 2019). Our H, negatively correlated to the recession coefficient, likely indicated we have a deeper and longer groundwater flow system and thus drainage slowly."

SUGGESTED EDITS
**Reply:** We greatly appreciate the numerous editing suggestions provided by the reviewer, including the rephrasing of words and sentences.

50: Replace "treatments" with "analysis methods" and delete the parenthetical phrase that follows.
**Reply:** Rephrased as suggested [L50].

51: Delete "centrality of recession". It's a strange term and may cause confusion.
**Reply:** Removed [L50].

60: Replace "period typhoon invasions" with "periodic typhoons".
**Reply:** Revised [L60].

64-65: Check for consistency in verb tense through paper. For example, "document" (present tense) and "discussed" (past tense) are used in the same sentence.
**Reply:** Thank you and checked [L65].

109: "unit" should be "units".
**Reply:** Revised [L113].

111: Replace "manipulate" with "affect".
**Reply:** Revised [L115].

150: Replace "was" with "is".
**Reply:** Revised [L154].

154: The material that is covered beginning with "Secondly," should be its own paragraph, as it is a distinct issue from data resolution. The new paragraph could begin with "An important concern in recession parameter estimation is the dependence between…"
**Reply:** Thanks and revised [L159].

155: Replace "blurs" with "confounds".
**Reply:** Revised [L159].

188: I would replace "presents a vague pattern…", with something like "shows no clear connection to large-scale landscape features on the island".
**Reply:** Revised as suggested [L193].

190: Replace "fluctuated" with "differed".
**Reply:** Revised [L195].

195: By "irrelative of drainage area", do the authors mean "not correlated with drainage area"? The latter phrase would be clearer.
**Reply:** Revised [L200].

263: Replace "vague" with "weak".

**Reply:** Revised [L262].

319: Replace "prevalently" with "prevalent".
**Reply:** Revised [L318].

346: Replace "hydrological" with "hydraulic".
**Reply:** Revised [L349].

355-359: I suggest deleting the entire part of this paragraph beginning with "Despite the power-law…" This paper is not about the impact of methodological choices.
**Reply:** Removed as suggested.

360: Replace "In our cases" with "In these catchments…"
**Reply:** Revised [L358].

360: Rearrange the sentence to read that the recession coefficient is moderately correlated to landscape structure while nonlinearity is only weakly correlated to landscape structure. Otherwise, it reads as if, for example, landscape structure is dependent on the recession coefficient. I would look for other places in the paper where the order should be reversed.
**Reply:** Thanks. We rephrased as suggested [L358-359].

360, 361: It would be clearer to always write "recession coefficient" and not simply "coefficient".
**Reply:** Revised [L359, 361].

369: Delete "Further". It is redundant to write both "further" and "also".
**Reply:** Revised [L369].

369: Rewrite the first sentence. Rainfall amount effects the recession coefficient, it does not affect the "estimating" of it.
**Reply:** Revised [L368].

373: Replace "In sum" with "In summary".
**Reply:** Revised [L372].

**Reply to Reviewer 2 Comment**

The reviewers' comments have been carefully addressed, the English substantially improved and the paper now conveys the results of the study and what it adds to the literature in a much clearer way.

**Reply:** We are grateful for the positive affirmation given by the reviewer. The suggestions provided in the past two rounds have significantly improved this manuscript.

[revised manuscript text omitted]

---

## Author Response (AR2)

**Department of Geography, National Taiwan University, Taipei, Taiwan**

No. 1 Sec. 4, Roosevelt Road, Taipei, Taiwan
* * *
**Dr. Jr-Chuan (River) Huang**

Tel & Fax: (886-2) 3366-5825; E-mail: riverhuang@ntu.edu.tw

Dear Editor,

Enclosed, please find the manuscript entitled "**Landscape structures regulate the contrasting response of recession along rainfall amount**" by Lee et al. for the second review.

First of all, we would like to express our sincere appreciation for the editor and reviewers' patience and constructive feedback. In this revision, we have rewritten the discussion and summary sections to concisely express our thoughts. We carefully checked and clarified any unclear or confusing statements, and also had a native English editor polish the manuscript. We hope that these revisions will clearly convey our findings.

In this review, referee #1 asked us to address the decorrelation method's ability to separate the dependency between recession parameters completely, in addition to solving the unit effect between a and Q. We addressed this point in sections 2.2, 4.1, and the summary, and made revisions accordingly. We also want to thank referee #2 for his/her help in clarifying our findings and pointing out confusing statements. In due course, we comprehensively rewrote the discussion and summary sections, fixed any typos and grammatical errors, and rephrased unclear statements.

Please feel free to contact me if you have any questions. Thank you for your assistance.

Sincerely,

Jr-Chuan (River) Huang, riverhuang@ntu.edu.tw
Professor, Department of Geography, National Taiwan University, Taiwan

**Reply to Reviewer #1's Comments**

GENERAL COMMENTS

This a much-revised version of the paper I previously reviewed entitled "Landscape structure and rainstorms swing the response of recession nonlinearity".

My main criticism of the original paper was of the interpretation of the causes of the variability in the recession parameter "a", which I said was flawed due to variability of "a" being unit-dependent as a consequence of the dependence of "a" on the power-law exponent "b". To address this issue, the authors applied the "decorrelation" method recommended by Dralle et al. (2017). I am skeptical that the decorrelation really solves the underlying problem such that causal mechanisms can confidently be attributed to variability in "a" when "b" varies. Biswal (2021) describes how the decorrelation does not completely dissociate "a" from "b" and concludes that the method of fixing "b" as a constant is preferable. I think the question that remains is, for the specific cases that the authors analyze, to what degree does the decorrelation method not solve the underlying issue but yet does improve the situation sufficiently that the interpretations that the authors make are generally valid, albeit with some uncertainty. This topic I believe is still an open research question, therefore I am open to having this paper be published. However, the authors should both acknowledge the arguments made by Biswal (2021) and acknowledge that there is methodological uncertainty in their analysis.

Overall, I am impressed with additional work that has been done. The introduction is much more thorough. The review summarized in Table 1 is, by itself, is a valuable contribution to the literature.

I think the English does not meet the standard of a HESS article. I sometimes guessed at the authors intended meaning. Improvement is needed throughout the entire paper, so I did not attempt to suggest edits. I am recommending major revisions primarily largely on the need to improve the English but also on some undeveloped arguments.

**Reply:** We appreciate that the Referee #1 recognizes our efforts on the revision, particularly for the application of the decorrelation method. We fully agree that this issue of dependence between a and b in power-law is quite open and needs more studies to solve. We added the following sentences in sections 2 and 4, and the summary to clarify the potential uncertainty and why the decorrelation method can't dissociate a and b completely.

In section 2.2 [L109-110], we added, the sentence, "Notably, since nonlinearity is

dimensionless, $\hat{a}$ is inherently strongly dependent on the unit of $Q$ and $b$ via fitting (see details in section 2.2.2)." to clarify the originality of the dependence.

In section 2.2.2 [L163-167]: We added the sentences, "Although the decorrelation method can reduce the unit effect and dependency on $b$, Biswal (2021) argued that the dependency of $\hat{a}$ and $b$ can't be fully decoupled, and retrieving parameters from the power law and fixing $b$ is preferable. Obviously, decoupling the dependency of $\hat{a}$ and $b$ in recession is unsolved and challenging and necessitates further study. Nevertheless, after the decorrelation process, the number of catchments with a high correlation between $a$ and $b$ ($R^2 > 0.1$) decreased from 9 to 2, apparently mitigating the unit-effect and dependency of $b$."

In summary [L371-373]: We added, "Note that $a$ and $b$ are inherently dependent, so some uncertainty might be involved. Even so, both parameters, whether derived using the point-cloud or individual segments (Fig. 4), present similar fluctuations among catchments, which supports our arguments." We introduced Biswal's point to the readers and demonstrated the improvement in dependence through decorrelation in our study.

In this revision, the undeveloped or unclear arguments raised by the editor and referees were carefully reviewed and revised for accuracy of wording. Also, our manuscript and response were fine-tuned by a native speaker. We have also carefully checked all unclear arguments and grammatical errors in order to meet the standard of the high-ranking journal, HESS.

LINE-BY-LINE COMMENTS

45-47: The transition from 3 to 1.5 is not due to groundwater being "vertically sourced from different hydraulic properties". The transition is due to the influence of the upstream boundary condition becoming a factor as the aquifer drains.

**Reply:** Yes. We rephrased as "For parameter $b$, hydraulic theories indicate that $b$ decreases from 3.0 to 1.5 during the transition from early to late recession as the influence of the upstream boundary condition becomes a factor when the aquifer drains in wet conditions (e.g., Rupp and Selker, 2006)." [L39-41]

211 and 229: Should this be Table 2?
**Reply:** Yes. We corrected it, please see [L200] and [L215].

255: See also Roque et al. (2022) who discuss the role of contrasting shallow and deep geologic layers on the recession parameters.
**Reply:** Thanks for providing the reference that can help deepen the discussion of the

geological aspect. We rephrased the statement as follows, "Perhaps, other controlling factors, such as geological structure (i.e., connectivity between the deep aquifer and the stream, heterogeneous hydraulic properties, and/or the interface slope between the shallow and bedrock layers, see Roques et al., 2022) or land cover (Tague and Grant, 2004), might alter recession behavior as well." [L250-254]

301-302: I do not see why, necessarily, the "inverse relationship between H and 'a' confirms that the hydraulic parameters vary markedly with depth". H is the vertical distance between the highest and lowest points in the basin, correct? It is a measure of the surface topography, not of the underlying geology. This statement requires further explanation.

**Reply:** We have revised the section, and we would like to point out that the inverse relationship between H and a has been explained by the following sentences, "Flow-path height, $H$, is directly linked to the water table depth in the homogeneous hillslopes. A steeper hillslope corresponds to permeable soils with higher $H$, leading to a deeper and longer groundwater flow system and slower drainage (Karlsen et al., 2019)." [L281-283].

325-326: I do not follow the explanation of why "a" would decrease with large storms. Why would drainage be slower from a major typhoon? What is meant by "overwhelm the effect of flow velocity?" Could not expect the opposite? During a heavy storm, there is more drainage from the upper layers, which are likely to be more conductive. There will also be more surface storage being drained, which I would think would be associated with a high "a". Table 1 also show more studies found negative than positive relationships of "a" with measures of initial storage. How did those studies interpret the negative relationship? I think Section 4.3.1 requires additional explanation.

**Reply:** We apologize for the confusing statements in the section. We have revised it to clarify the explanation of why "a" may decrease with large storms. The revised sentences are, "Harman et al. (2009) demonstrated that the recession coefficient can be expressed as $a = V_0/R^{b-1}$ (where $V_0$ and $R$ represent the mean of the velocity distribution of hillslope flow and rainfall rate, respectively). In the case of heavy rainfall, the increase of $R$ is much larger than that of $V_0$. The effect of this disproportionate rainfall input increase on $a$ could offset the increase in flow velocity, resulting in a negative correlation. Moreover, Biswal and Nagesh Kumar (2014) used a geomorphological recession flow model $a \propto c/q^{b-1}$ (where $c$ and $q$ represent the celerity and rate of channel flow, respectively, and which is similar to Harman's theory) to explain why "$a$" is negatively correlated with "$q$." To sum up, the negative correlation between coefficient $a$ and rainfall amount (e.g. peak flow and prior soil moisture) is

consistent with the literature and is prevalently in most regions (also see Table 1)."
[L312-319]

Table 1: It would be helpful to the see the authors' study placed in this table, too.
**Reply:** We added our study in Table 1.

Figure 9: It is not clear to me what the arrows are for in the lower three panels.
**Reply:** We have made the arrows more prominent and added a caption to explain their significance as follows, "Correspondingly, the bottom row shows how their recession parameters (or regressive line) in recession plots would move from light (dashed line) to heavy (solid line) rainstorms."

**Reply to Reviewer #2's Comments**

The revised version of the paper did take into account the methodological recommendations of the reviewers. However, the summary and discussion remain almost unchanged, i.e. the methodological changes are not reflected in these parts of the paper, which is surprising. In particular, the numbers in the discussion are not updated, which seems strange. The discussion and the summary of the paper do not convey, which of the findings are new, which ones are specific to the chosen case study and how the findings could be related namely to the climatic conditions. There are no clear conclusions, i.e. we do not know if the study found the same results as at other locations or if there are specific new insights. Some more comments are in the annotated pdf.

The language is critical, at instances hard to understand. In the annotated version, I highlighted some instances but those are by no means exhaustive.

**Reply:** We appreciate Referee #2's constructive comments and kind help in clarifying our unclear statements. Our response to this main concern is as follows:

In this revision, the discussion and summary were totally rewritten to express our work precisely, but the main story remains. Although we updated the parameter values using the decorrelation method, their response to various environmental variables has not changed significantly. Therefore, our original discussions and conclusions are not greatly affected. Nevertheless, we have updated the opinions raised by referee #2 in our discussions and summary. The main contribution of this study is to demonstrate that: "The results showed that $a$ and $b$ respectively increase and decrease with $L/G$ (the ratio of flow-path length to gradient), particularly in small catchments. Additionally, corroborating previous studies, $a$ decreased significantly with rainfall amount. However, nonlinearity increases with rainfall amount in larger catchments but decreased in small catchments."[L15-18] Additionally, we have compiled the literature on empirical recession behavior and identified the relationships between recession behavior and environmental factors, presented in Table 1, which should be important for further recession studies.

We would like to apologize for any confusion caused by our limited English proficiency, which may have made it difficult to understand our work. We are grateful to Referee #2 for drawing our attention to sentences with unclear meaning. We have not only rephrased those sentences but also thoroughly revised most of the text. Additionally, we have enlisted the help of native speakers to extensively edit the manuscript, and we hope that it now presents our arguments more clearly.

COMMENTS (**\*Line number in the previous annotated pdf**)

12-14: you do not mention the methodological changes between the first and the revised version.

**Reply:** We point out the decorrelation process in L15

65-67: not good writing practice

**Reply:** Rephrased.

67-68: unclear synthesis of the different theories, so what is actually the contradiction?

**Reply:** We have revised the paragraph in [L39-44] to better convey the contradiction. It now reads as follows: "For parameter $b$, hydraulic theories indicate that $b$ decreases from 3.0 to 1.5 during the transition from early to late recession as the influence of the upstream boundary condition becomes a factor when the aquifer drains in wet conditions (e.g., Rupp and Selker, 2006). Spatial heterogeneity theory demonstrates that b only slightly increases with a wet antecedent condition (Harman et al., 2009). However, drainage network theory indicates that $b$ increases with storage while the downstream receives more subsurface flow contribution but decreases with storage as the downstream receives less (Biswal and Nagesh Kumar, 2013)."

73: what is this?

**Reply:** We replaced "landscape regimes" with "regions" [L63].

86: you mean landscape characteristics?

**Reply:** Certainly. In fact, using the term "characteristics" is also an appropriate way to convey the broad sense in this context. However, given that we already mentioned L/G in the abstract and summary, we will retain the term "variables" here.

88: how? such a general hypothesis is not very useful.

**Reply:** After careful consideration, we realized that the general hypothesis may not be particularly useful in this context. As such, we have decided to eliminate it from our study. Thanks for bringing this to our attention.

135: why do you bring in the water balance equation here? this was not mentioned in the first round but in fact, it does not add anything here; it is rather confusing since E influences how streamflow reacts to a precip event and how much water will percolate and recharge groundwater, so it clearly has an influence on recessions; but: recession analysis is supposed to analyse how baseflow recedes in absence of input;

simply remove the comment on P and E

**Reply:** Thanks for the valuable feedback regarding our use of the water balance equation in this section. We have removed the water balance part and revised the beginning of section 2.2 to better reflect the storage-outflow relationship. It now reads: "The storage-outflow relationship is typically described by a power law if treating the catchment as a black box. The representative storage is, in fact, composed of many aquifers and thus exhibits a non-linear relationship". [L97-98]

138: all should have same units

**Reply:** Evapotranspiration has been removed accordingly.

139:  Q=mSn, always or only during the recession?

**Reply:** Please see above.

141: well, P is small or zero by definition of a recession, but E is definitively not!

**Reply:** Comment on P and E has been removed.

177-178: this does not add anything and is wrong, recession (the true recession part) is only baseflow; but if you let the recession start at peakflow, part of the segment contains subsurface flow that is not baseflow (groundwater)

**Reply:** The treatment of recession segments depends on the purpose of the study. To ensure that groundwater dominates the recession signal, the recession segment is typically chosen to lag a few days after the peak flow (e.g., Biswal and Marani, 2014). However, other studies consider starting the recession segment from the peak flow, as fast-flow processes (such as subsurface flow) may also contribute to power law recession behavior (e.g., Dralle et al., 2017). Upon summarizing the empirical power law recession studies in Table 1, we carefully checked the initial time of the recession segment in Table S1. Table S1 indicates that out of the 24 studies, 15 of them had a lag time of less than 1 day for the recession segment.

195: English

**Reply:** Eliminated

205-207: reference?

**Reply:** We moved the reference, Roques et al. (2017), from [L155] to here [L156].

207-208: reference? how is k estimated, over all recessions or per recession? cannot simply refer to another paper, we need this information here; if it is

estimated accross all recessions, how are they pooled together? if it is estimated per recession, then, of course, we have a new time-varying parameter.

**Reply:** The decorrelation method can be applied to either all recessions (point-cloud) or individual segments. In our study, we applied this method to both cases. For further details and a more comprehensive explanation of our methodology, please refer to Dralle et al. (2015), which is included in the reference list [L159].

216: English

**Reply:** Rephrased. Now it reads, "The streamflow recession plots of catchments W9, W5, and W8, as examples, are illustrated in Fig. 2." [L173]

232-233: small = asymetric; before you said skewed, what is the difference?

**Reply:** I apologize for the confusion. Now it reads, "The small difference between the median and mean suggests a relatively symmetric distribution." [L184-185]

240: English

**Reply:** Please see below.

248: deviation from what?

**Reply:** Please see below.

249: english!

**Reply:** It appears that Referee #2 had some concerns with the clarity of our original paragraph. We have since rephrased the paragraph to provide a clearer explanation. The revised paragraph now reads as follows: "Notably, when the drainage area is larger than 800 km$^2$ (W19 and larger), the point-cloud-derived coefficients become similar to the third quantile of the distribution of individual segments. For nonlinearity, the values derived from the point-cloud are consistently close to the lower limit of the distribution of the individual segment-derived values and the median and interquartile range of nonlinearity derived from individual segments are irrelative of drainage area. These distinct differences between coefficients and nonlinearities from the two fitting methods make comparison and interpretation difficult. The details of the recession characteristics for each catchment can be found in Table S4." [L192-197]

252-253: is this a common finding or not?

**Reply:** This finding is getting recognized recently (perhaps since 2017). We addressed this point in section 4.1. [L235-236]

254: referred

**Reply:** corrected.

256: english, the title does not mean anything

**Reply:** We rephrased it as "*Relationships between recession parameters and event/landscape variables*" [L198]

258: hydrometric forcing: what is this? hydrometry refers to the measurement of streamflow

**Reply:** You are correct that "hydrometry" refers to the measurement of streamflow. However, in the given sentence, it seems that "rainfall forcing" would be a more appropriate term to use [L199].

259-261: english: the correation is not significant, not the values themselves

**Reply:** We've rephrased it, now it reads, "As for initial event conditions, the 7-day antecedent precipitation, $AP_{7day}$, defined as the seven-day rainfall amount prior to a rainstorm, was not correlated to $a$, nor were other $AP$ period lengths (3-, 5-, 14-, and 30-day)." [L203-205]

274: on what?

**Reply:** Typo and corrected [L209-210]

274-276: English

**Reply:** This is a redundant sentence. We've removed it.

294: this 4.1 does not discuss any link to "subtropical"

**Reply:** We apologize for any confusion. It seems that there have been a mistake. We've eliminated "subtropical".

295: coefficients

**Reply:** Thank you. The whole discussion has been rewritten.

299-301: what is new here? compared to literature?

**Reply:** While the concept that landscape characteristics could influence recession behavior is not a new idea, our study provides a novel approach for explicitly representing the aquifer and assessing the relationship between landscape structure and recession coefficients. Specifically, we introduced the L/G ratio as a useful index for analyzing recession behavior. We rephrased the sentence and it now reads "Taken

together, these data demonstrate how landscape structure, particularly drainage density and flow-path-associated variables, can affect the recession coefficient. The findings presented in Table 2 corroborate this (discussed more in Sect. 4.2)." [L242-243]

311: this part is not specific to the the selected catchments but a general methdological consideration; and it is unclear how it relates to text above

**Reply:** This paragraph has been rewritten and the comment has been addressed in this revision.

348: English

**Reply:** Sorry for the any confusion. The statements were revised.

435: English

**Reply:** Please see below.

435-441: your perceptual model or a hypothesis developed previously ?

**Reply:** The perceptual model was developed by the authors of this study. We've rephrased the statement as follows, "The above two sections have demonstrated the influence of landscape and rainfall amount on streamflow recession behavior. Thus, a perceptual model which demonstrates the interactive regulation of landscape structure and rainfall amount on recession nonlinearity is introduced (Fig. 9)." [L339-341]

442: Type B to A: what is this?

**Reply:** We rephrased the sentence as follows, "Along the spatial heterogeneity dimension (from Type B to A, with increasing drainage area), additional perched storages respond increasingly with rainfall amount and thus enhance the recession nonlinearity." [L346-347]

452: new summary does not reflect new methods
**Reply:** Replied above. Also, the summary was thoroughly revised.

455: what is this "catchment events"? where, in what climate?
**Reply:** Please see below.

458 and 459: english!
**Reply:** We've rephrased the statement as follows, "This implies that it is not possible to infer recession characteristics by comparing the parameters found in the literature.

460: inference is reserved to specific techniques, not correct use
**Reply:** Replaced "inference" with interpretation.

462: avoid abreviations of this case in the summary, make it more stand-alone
**Reply:** Thanks for the suggestion. In this revision, the only abbreviation is "L/G".

462-463: what is new here compared to previous work?
**Reply:** Please see below.

463: in the summary, should be made clear again what the recession coefficient is (the nonlinear recession has two parameters)
**Reply:** Suggestion accepted. Please see below.

470-471: at this stage, we would like to see answers, not hypotheses
**Reply:** Agreed and revised accordingly.

471: english! landscape cannot have preferences
**Reply:** Corrected.

473: response to what?
**Reply:** Please see below.

473-474: do not understand
**Reply:** The above 5 comments are related to the summary. The previous version of the summary was not clear and organized, which may have made it difficult for the referees to understand the arguments presented. We have revised the summary to address these concerns and provide a more detailed explanation of our findings.

1. The coefficient and nonlinearity derived from point-cloud are considerably larger and smaller, respectively, than the median of individual segments.
2. the coefficient increases with *L/G* and nonlinearity decreases with *L/G* significantly in small catchments. This likely reveals that both spatial heterogeneity and hydraulic properties regulate recession simultaneously.
3. Further, rainfall amount also plays a dominant role in estimating parameter *a*. It decreases with rainfall amount for all catchments.
4. The contrasting response directions of nonlinearity to rainfall amount could be

found along the dimension of spatial heterogeneity (drainage area)

5. landscape structure (spatial heterogeneity and hillslope hydraulics) may determine the recession behavior via various aquifer settings, and the rainfall amount tunes the magnitude of recession nonlinearity.

---

## Editor Decision (ED2)

Dear Authors,

Thank you for your corrections to the previous version of the manuscript. I spent quite some time going over your manuscript again and found that while your revision is an improvement, several issues still need to be clarified:

L/G: median L /median G or median L/G? There is a difference. In McGuire 2005 they use median L/median G. You say in Table S3 that you use median (L/G) – why? Explain what this means for your results. Especially as you are now using a different measure than previous studies.

Also – you should state in the methods that you are using median values. Throughout the text it is not clear when you are talking about L or G or H if you are referring to the median or not. This needs to be clarified.

Why is L/G a good proxy for residence time? If you base this on Darcy's law it would be good to explain this better than the currently slightly confusing sentence: "Note that G can also be regarded as a surrogate of flow velocity (most equations used for estimating flow velocity needs gradient to represent the conversion from potential to kinetic energy). Therefore, the composite ratio, L/G [m], can be a proxy for residence time (McGuire et al., 2005; Tetzlaff et al., 2009) and as a means to comprehend the interplay between landscape features and climate impacts on residence time (Seybold et al., 2017)." Instead I would suggest something similar to this (if this is indeed what you mean): As the velocity of gravity-driven flow is usually proportional to the gradient: $v = L/T \sim H/L$ this results in time T being proportional to L/G: $T \sim L^2/H = L/G$. Therefore, L/G could be a potential proxy for residence time. In terms of a catchment this relates to the ratio of the medians of L and G.

What do you mean by composite ratio? What is the difference between composite ratio and ratio? Why is a simple ratio not enough? This needs to be explained.

You are citing that Harman et al. 2009 found that heterogeneity between hillslopes increased with catchment area. However, they compare a single hillslope to a catchment of 10 ha and a catchment with a catchment area of 41 ha, so a maximum scale of not even 0.5 km². In your study, catchment areas only begin at 77 km². Does this relationship still hold at this very different spatial scale? This should be discussed.

Table 3 in the supplement – here the language can be simplified/clarified and there are also still some expressions that should be corrected:

- Definition of DD: fast not faster
- Water body coverage
- Forest coverage
- Agricultural land coverage

Please also see my comments to the supplementary material and within the pdf of your response.

I am glad to see the major improvements in this manuscript over the course of the review and am hoping that we can resolve these issues with a last round of minor revisions. Given that there are still some issues with the language I have been in touch with the English copy-editing department of Copernicus and they will assist you with final improvements on this front. Unfortunately, even your revised abstract still needs work, as the sentences have the tendency to be convoluted and confusing.

Looking forward to bringing this review process to a close and moving your manuscript forward towards publication!

All the best,

Theresa

**Reply to Editor's Comment**

Dear Authors,

unfortunately quite a few of your recent corrections in response to the reviewer's comments still require some work. In several instances it is not really possible to understand the meaning of what you are trying to say.

I have added some comments in your response as well as at the beginning of the track changes document, but please also go over the remaining changes and clarify and correct the english where necessary. Some of my comments appear only in the response but also apply to the manuscript.

Please also make sure that the corrections are made not only in the one line the reviewer references but also in the other instances throughout the manuscript.

Please note that you need to download the pdf to see all the comments as not all of them appear when viewing the pdf in the browser

Once these things have been clarified and corrected I will review the manuscript again.

All the best,

Theresa

**Reply:**

Dear Theresa,

We extend our gratitude for your meticulous review of our manuscript. We have addressed all the comments you highlighted in our previous communication and made necessary revisions to the manuscript. We thoroughly revised the manuscript and clarified the ambiguous sentences. Mainly revisions are in the abstract, landscape variable [L82-92], and discussion 4.2.1 [L279-289]. Also, Table S3 which illustrated the definitions and calculations of hydrologic-event and landscape variables was clarified and revised. We have also ensured that all responses in the manuscript are correspond to the appropriate corrections.

Best regards,

Jr-Chuan (River) Huang

**SPECIFIC COMMENTS**

(in response)

*"Also, we conducted a comprehensive review of the vocabulary and grammar throughout the entire manuscript."*

This still needs to be improved, especially in the newly added text.

**Reply:** We have diligently refined the revised manuscript and engaged a native speaker to further enhance the precision of our text.

*"Without considering this contrasting response, which is contingent upon landscape structure, it leads to a misjudgment of the recession nonlinearity in response to rainfall amount and needs further clarification, particularly for use in assessing regional recession in ungauged catchments under climate change."*

This sentence is not clear and needs to be rephrased.

**Reply:**   In this version, we rephrased the abstract thoroughly and highlighted our three new findings. For the last two sentences, now it reads, "Our finding that $L/G$ and drainage area might regulate the contrasting response of recession along rainfall amounts requires additional validation in different regions since recession response is crucial when assessing regional recession in ungauged catchments under the influence of climate change." [L20-24]

*"For example, it is not enough to only state that the value of b decreases as L/G increases. What is the significance of L/G? What does it represent with regards to what influences the flow of water through the catchment What does it imply in terms of subsurface flow that b decreases as L/G increases?"*

This should be answered in the manuscript and explained in more detail.

**Reply:**

To clarify the composite L/G ratio, we have revised several sections of the manuscript, including the abstract, material and methods, and discussion.

In the abstract [L16-17], we added a short definition of *L* and *G*. Now it reads, "(*L*: median of flow-path lengths within a catchment; *G*: median of flow-path gradients within a catchment)".

In the material and methods [L82-92], we rewrote the paragraph to provide a clear definition, explanation, and simple calculation of flow path variables. It is, "In addition to the primary landscape variables described above, we incorporated flow path associated variables into our study, as flow path is an explicit proxy for aquifer systems. Within a gridded DEM, the flow path is defined as the route followed by water from a grid cell, following the surface flow direction towards the channel cell (see detail in Tetzlaff et al., 2009). Specifically, flow path length *L* [m] is the length of this route, flow path height *H* [m] is the elevation difference between a specific cell to the channel cell, and *G* is the flow-path gradient [-], defined as  the flow path height divided by flow path length. As such,

every grid cell possesses distinct values for *L*, *H* and *G*. Within a catchment, the medians of the *L*, *H*, and *G* distributions serve as representative flow path characteristics. Note that *G* can also be regarded as a surrogate of flow velocity (most equations used for estimating flow velocity needs gradient to represent the conversion from potential to kinetic energy). Therefore, the composite ratio, *L/G* [m], can be a proxy for residence time (McGuire et al., 2005; Tetzlaff et al., 2009) and as a means to comprehend the interplay between landscape features and climate impacts on residence time (Seybold et al., 2017). The detailed definition and calculation of the flow path associated variables are illustrated in Table S3 in the supplement."

In discussion 4.2.1 [L279-289], now it reads, "Our variable *H*, which exhibits a negative correlation with the recession coefficient, likely suggests that our groundwater flow paths possess greater depth and length, consequently leading to slower drainage rates. Although flow path height, *H*, denoting potential energy is a component of gradient, it does not necessarily correspond to hydraulic gradient due to the geologic and soil settings varying across regions (Karlsen et al., 2019). Besides, high *DD* and short *L* indicate shorter flow paths and thus lead to a higher recession coefficient. In our cases, Type C catchments are characterized by short *L* and very small *H* and thus have high *L/G* ratios and recession coefficients (solid orange dots in Fig. 7c). Individually, extended *L* or gentle *G* is conducive to flow accumulation. Thus, the *L/G* ratio, which integrates both length and gradient, serves as a good proxy for estimating residence time (McGuire et al., 2005; Asano and Uchida, 2012). While the equivalent composite ratio can result from either *L* or *G*, the relationship between recession parameters and *L/G* has the potential to establish a further linkage between recession parameters and water residence time."

Hope the revised text expresses the intended meaning more effectively.

*"landscape structure"*
Not clear.
**Reply:** Landscape structure in our study is interpreted by hillslope hydraulics (L/G) and inter-hillslope heterogeneity (A). While we have introduced this term in the introduction to convey the general concept, we employed the terms L/G and A in the main body of the text to provide specific emphasis.

*"Therefore, we addressed it in L81-88 and added a new Table S3 in supplementary for describing the definition and calculation of landscape and rainstorm variables."*
Not enough.
**Reply:** Please see reply above. Besides, Table S3 has been revised.

*"Specifically, from the aspect of aquifer hydraulics (Rupp and Selker, 2006), spatial heterogeneity (Harman et al., 2009) and drainage network (Biswal and Marani, 2010) have been observed that these*

*recession parameters are influenced by the aforementioned factors."*

Sentence does not work, needs to be fixed.

**Reply:** We have removed this sentence since the following sentences explained the works.

*"Additionally, theoretical works have shown that the dependence of streamflow recession parameters on antecedent storage or rainstorms."*

Needs to be fixed.

**Reply:** Fixed. Now it reads, "Additionally, theoretical studies have demonstrated that streamflow recession parameters are subject not only to the influences of landscape and aquifer systems but also to the interplay with antecedent storage and rainfall events" [L39-40]

*"the downstream"*

Not a noun.

**Reply:** We replaced "the downstream" with "the downstream channel" [L44] and [L45].

*"Replied above."*

Where? should be stated in the manuscript text

**Reply:** In the introduction section of this version, paragraph #2 and #3 demonstrated the theoretical works and paragraph #4 expressed the empirical studies.

*"point-cloud"*

this also needs to be explained in the methods section, where you suddenly switch from single recession curves to the so-called point cloud.

**Reply:** In the methods section, we have revised the sentence introducing the term "point-cloud": "One approach involves fitting the lower envelope of a collection of multiple recession curves, which is referred to as point-cloud (Brutsaert and Nieber, 1977)."[L150-151]

*"The flow path is defined as the hillslope grid point following the surface flow direction toward channel."*

Not clear. A point cannot be a path

**Reply:** Please see reply above. We rephrased the paragraph [L82-92] to interpret the definition and calculation of flow path associated variables.

*"Among them, the composite ratio of L/G, which represent the distance effect of flow-path under different gradient holds hydrologic significance as it can serve as a proxy for water residence time (McGuire et al., 2005; Tetzlaff et al., 2009)."*

Explain why this can serve as a proxy for residence time and what you mean by composite ratio.

**Reply:** We rephrased the sentences in [L88-91]: "Note that $G$ can also be regarded as a surrogate of

flow velocity (most equations used for estimating flow velocity needs gradient to represent the conversion from potential to kinetic energy). Therefore, the composite ratio, *L/G* [m], can be a proxy for residence time (McGuire et al., 2005; Tetzlaff et al., 2009) and as a means to comprehend the interplay between landscape features and climate impacts on residence time (Seybold et al., 2017)."

*"detailed"*
**Reply:** Revised as suggested [L92].

Table S3: This table contains errors in units and definitions. The explanations for how variables are calculated are often not clear and need to be improved.
**Reply:** Revised thoroughly. Please see the new Table S3.

*"the supplement"*
**Reply:** Revised as suggested [L92].

*"Is G = H/L? Is L/G, therefore, simply L/(H/L) = L^2/H? If so, what does L^2/H say*
*about landscape structure that simply G (or 1/G = L/H) alone does not?"*
I have the same question and don't feel that you answer it below. We need a better explanation for the calculation and meaning of L/G.
**Reply:**
L and G are the length and gradient of the flow path. Since the gradient is an important variable for estimating velocity, the composite L/G ratio could be imagined as a kind of time due to distance over velocity. Therefore, L/G represent the hillslope hydraulics and it explained why L/G is a good proxy for water residence time at catchment scale.

For variable selection, the variables of H, L, L/G and DD shown in Fig. 7 are highly correlated with recession parameters. Our focus on the composite L/G ratio aims to underscore its potential in establishing a linkage between recession parameters and water residence times.

*"The composite L/G ratio represents the distance effect of flow path under different gradient"*
I don't understand this.
**Reply:** Replied above.

*"Given how much of the discussion on the results centers on L/G, it is important that*
*its geomorphic/hydrologic/hydraulic significance be stated. Reply: Replied."*
Need a better explanation than what you gave above
**Reply:** Replied above.

How does a large value of L/G imply a "short-and-gentle" hillslope?

This does not answer the question and the correction in the ms is missing.

**Reply:** Apologies for the unclear interpretation. The large L/G value certainly is derived from a long flow path or gentle G. However, our Type C catchments are characterized by short *L* (Fig. 7b) and very small *H* (Fig. 7a) and thus have high *L/G* ratios (Fig. 7c). We have rewritten this paragraph [L279-289] for clarification.

*"Actually, "b" is the slope in a plot of log(-dQ/dt) vs log(Q)."*

This should be corrected throughout the ms and not just in this one line!

**Reply:** Our apologies. This time, we have checked throughout the manuscript and made the necessary corrections [L31].

*"Reply: We observed that when the drainage area larger than 800 km2, the point-cloud derived coefficients become similar to the third quantile of the coefficient distribution from individual segments [L197]."*

Explanation should be added to ms.

**Reply:** We had added the following sentence to the manuscript in [L201-202]: "Notably, when the catchment size exceeds approximately 500 km² (W19), the point-cloud-derived coefficients become similar to the third quantile of the coefficient distribution from individual segments."

*"described the calculation methods"*

Needs to be improved and in part corrected

**Reply:** We improved Table S3. We modified the gridlines of Table S3, correcting the unit of DD to [km km-2], changing the "meaning" column to "definition and meaning." We have also rephrased the calculation methods for each variable.

*"the superimposition of recession events on antecedent flows"*

Not clear, please rephrase

*"The negative correlation between b and peak flow does not necessarily imply a consistent response across all catchments."*

I don't understand this

**Reply:** We rephrased in [L328-332]: "Across all catchments, we observed an augmentation of exponent *b* with antecedent flows, but a decline with peak flow (Fig. 5). This augmentation can be attributed to the overlay of  onto antecedent flows, amplifying the value of *b* (Jachens et al., 2020). The inverse correlation between *b* and peak flow suggests that in the majority of catchments, the existence of active fast flow paths could potentially reduce the recession nonlinearity.".

Replace "than" with "compared to those derived from"

**Reply:** Revised as suggested [L235]. Thank you.

*"we have a deeper and longer groundwater flow system and thus drainage slowly"*

English needs to be fixed, also groundwater flow systems cannot really be long, only flow paths.

**Reply:** We rephrased it as "our groundwater flow paths possess greater depth and length, consequently leading to slower drainage rates." [L280-281]

**(in the track change document)**

L10: are these supposed to be two different things? Not clear.

**Reply:** We rephrased it as "landscape structures and rainstorms are recognized as drivers of recession response" [L10].

L10-11: needs to be rephrased.

L11: Logical flow not clear. Why "yet"?

**Reply:** We rephrased the first two opening sentences. Now it reads, "Streamflow recession reflects hydrological functioning, runoff dynamics, and storage status within catchments and landscape structures and rainstorms are recognized as drivers of recession response. However, the documented recession responses to landscape structure and rainstorms are inconsistent, and there are fewer studies that concurrently investigate the combined effects of these two factors on recession." [L9-12].

L13: be is the exponent in the equation, I don't think it is called "the nonlinearity"

**Reply:** Checked. In this version, when talking about equation or parameter, we used exponent b. For describing the recession behavior, nonlinearity is used.

L16: not clear, this is not quantitative and therefore it is not clear how something increases with structure. It is also completely unclear what you mean by landscape structure.

**Reply:** As suggested, we only used the term "landscape structure" to convey the general concept, but used specific variables like L/G and A to express the relationships. Now it reads, "Our finding that *L/G* and drainage area might regulate the contrasting response…." [L21].

L18: the exponent of the recession model and therefore

L19: replace "Without" with "Not"

L19-21: not clear. Are you saying we normally are just guessing recession model parameters for ungauged basins? Why and under which circumstances would we do that? Not clear what you are trying to say here.

**Reply:** We rewrote the abstract to increase its readability. Now it reads, "Our finding that *L/G* and drainage area might regulate the contrasting response of recession along rainfall amounts requires

additional validation in different regions since recession response is crucial when assessing regional recession in ungauged catchments under the influence of climate change." [L20-24]

L24: not clear. Runoff does not exist in plural form. What do you mean by different runoffs?
L24: the streamflow recession and its link with flow
**Reply:** We replaced "runoff" with "flow paths" and rephrased as suggested [L26]. Thank you.

L26: why is this equation different to the one in the abstract?
**Reply:** The recession parameters of the equation in the abstract were decorrelated, which differs from the equation in the original Brutsaert and Nieber (1977) version. For simplification, we eliminated the equation in the abstract.

L26: not the correct term
**Reply:** We replaced "streamflow declines" with "the rate of change in streamflow" [L28]

L26: describe the recession
**Reply:** Revised as suggested [L28].

L28-29: this is something that the reviewer corrected (see line 125), Why did you only correct it there but not here?
**Reply:** Apologies for only revising [L116] earlier. This time, we have checked throughout the manuscript and corrected it [L31].

L32-34: sentence needs to be fixed
**Reply:** We removed this sentence since the meaning of the sentence had already been explained in the previous statement.

L50: The exponent b slightly increases...
**Reply:** Revised as suggested [L43].

L52: downstream is not a noun
**Reply:** We replaced "the downstream" with "the downstream channel" [L44] and [L45].

L59-60: Most previous studies aggregated long-term data to a point-cloud, a collection of multiple recession curves *to* retrieve representative recession parameters
**Reply:** Revised as suggested [L51-52].

L67: streamflow

**Reply:** Revised as suggested [L57]

L69: I would not use this as a synonym for b

**Reply:** We replaced "nonlinearity" with "exponent" [L60]

L124: what is the difference to a?

**Reply:** $\hat{a} = ak^{b-1}$. We have elaborated on the rationale for this relationship in [L163-167]: "An important concern in recession parameter estimation is the dependence between $\hat{a}$ and $b$, which confounds the interpretation of parameters (Dralle et al., 2015). The decorrelation method assumes that the observed flow, $Q$, consists of a scale-free flow $\hat{Q}$ and a constant $k$ ($Q = k\hat{Q}$). Thus, the power law formula can be rewritten as $-dQ/dt = ak^{b-1}\hat{Q}^b$, where $a$ is the scale-free recession coefficient [$h^{-1}$]. For correcting $\hat{a}$ to $a$, the observed flow $Q$ was divided by a constant $Q_0$ (which is ideally equal to $1/k$, see detail in Dralle et al., 2015)".

**Supplementary Material**

**Table S1.** Summary of empirical power-law recession studies. The number of references corresponds to Table 1 in the main text. The parameter a and â represent decorrelated and un-decorrelated, respectively. $T_0$ represents recession timescale at the median flow. CTS and VTS denote constant and various time interval of sampling (Q, dQ/dt) pair, respectively.

| No | Reference | Data pool | Temporal scale | Location | Number of basins | Number of events | Basin area (km²) | Unit of flow | Initial time of recession segment (day after $Q_p$) | Sampling way (Q, dQ/dt) | $b$ | Target parameters |
|---|---|---|---|---|---|---|---|---|---|---|---|---|
| 1 | Mathias et al. (2016) | Point-cloud | Long-term | UK | 120 | n.a. | 1.1-1700 | $L\,T^{-1}$ | 0 | CTS | 1.68-1.99 | â, b |
| 2 | Patnaik et al. (2018) | Median | Long-term | Eastern USA | 212 | n.a. | n.a. | $L^3\,T^{-1}$ | 1 | CTS | 1-6 | b |
| 3 | Tashie et al. (2019) | Median | Monthly | North Carolina | 1 | 382 | 0.6 | $L\,T^{-1}$ | 1 | CTS | 4-20 | a, b |
| 4 | Bart and Hope (2014) | Events | Event | California | 4 | n.a. | 119-632 | $L\,T^{-1}$ | 7 | CTS | 1.8-2.1 | â |
| 5 | Biswal and Nagesh Kumar (2014) | Events | Event | USA | 67 | n.a. | 10-8858 | $L^3\,T^{-1}$ | 0 | CTS | 1.47-4.57 | â |
| 6 | Biswal and Marani (2014) | Events | Event | Eastern USA | 4 | n.a. | 41-583 | $L^3\,T^{-1}$ | 1 | CTS | 1.91-2.23 | â |
| 7 | Clark et al. (2009) | Point-cloud | Long-term/event | Georgia | 3 | n.a. | 0.001-0.41 | $L\,T^{-1}$ | 0 | VTS | 1-3 | b |
| 8 | Ghosh et al. (2016) | Events | Event | Georgia | 1 | 23 | 0.41 | $L\,T^{-1}$ | 0.25 | CTS | 2.5-7.8 | â, b |
| 9 | Patnaik et al. (2015) | Median/Events | Long-term/Event | USA | 358 | n.a. | 2-3247 | $L^3\,T^{-1}$ | 7 | CTS | n.a. | â |
| 10 | Millares et al. (2009) | Point-cloud | Long-term | Spain | 3 | n.a. | n.a. | $L^3\,T^{-1}$ | 0 | CTS | 1.15-1.30 | â |
| 11 | Sayama et al. (2011) | Point-cloud | Long-term | California | 17 | n.a. | 3-112 | $L\,T^{-1}$ | 0 | CTS | n.a. | b |
| 12 | Shaw and Riha (2012) | Events | Event | New York | 7 | 80 | 100-6415 | $L^3\,T^{-1}$ | 0 | VTS | 1.31-5.34 | â |
| 13 | Shaw et al. (2013) | Events | Event | New York | 9 | 72 | 287 | $L^3\,T^{-1}$ | 0 | VTS | 0.98-2.42 | â |
| 14 | Tague et al. (2004) | Point-cloud | Long-term | Oregon | 22 | n.a. | 7.3-1337 | $L^3\,T^{-1}$ | 0 | CTS | 1.38-3.16 | â, b |
| 15 | Tashie et al. (2020) | Events | Event | USA | 1027 | 155309 | n.a. | $L^3\,T^{-1}$ | 0 | CTS | 1.1-7.3 | b |
| 16 | Yan et al. (2022) | Point-cloud | Long-term | Eastern China | 382 | n.a. | 34-18211 | $L^3\,T^{-1}$ | 2 | CTS | 0.57-3 | â, b |
| 17 | Ye et al. (2014) | Point-cloud | Long-term | Eastern USA | 50 | n.a. | 66-9062 | $L\,T^{-1}$ | 3 | CTS | 0.99-1.91 | â, b |
| 18 | McMillan et al. (2014) | Median/Point-cloud | Long-term/monthly/event | New Zealand | 28 | n.a. | n.a. | $L\,T^{-1}$ | 0.5 | VTS | 1.5-4.0 | $T_0$, b |
| 19 | Biswal and Nagesh Kumar (2013) | Events | Event | USA | 39 | 5486 | 9.6-5457 | $L^3\,T^{-1}$ | 0 | CTS | 1.52-2.61 | b |
| 20 | Chen and Krajewski (2015) | Events | Event | Iowa | 25 | n.a. | 66-16854 | $L\,T^{-1}$ | 12 | CTS | 0.75-1.6 | â, b |
| 21 | Bogaart et al. (2016) | Point-cloud | Annual | Sweden | 316 | n.a. | 3-33000 | $L\,T^{-1}$ | 3 | CTS | 0.5-2.1 | â, b |

| | | | | | | | | | | |
|---|---|---|---|---|---|---|---|---|---|---|
| 22 Dralle et al. (2017) | Events | Event | California/Oregon | 16 | n.a. | 17-5457 | $L^3 T^{-1}$ | vary | CTS | 0.1-3.7 | a |
| 23 Santos et al. (2019) | Events | Annual/Event | Switzerland | 5 | n.a. | 50-352 | $L T^{-1}$ | vary | CTS | 1.73-2.4 | a, b |
| 24 Karlsen et al. (2019) | Events | Seasonal/Event | Northern Sweden | 14 | 163 | 12-6790 | $L T^{-1}$ | 2 | VTS | 1-10 | $T_0$, b |

**Table S2.** Landscape and landcover variables of the selected catchments.

| ID | HID | $H$ | $L$ | $G$ | $L/G$ | $A$ | $DD$ | $S_m$ | $HI$ | $ELO$ | $C_W$ | $C_F$ | $C_A$ |
|----|-----|-----|-----|-----|-------|-----|------|-------|------|-------|-------|-------|-------|
| | | (m) | (m) | (-) | (m) | (km²) | (km/km²) | (%) | (-) | (-) | (%) | (%) | (%) |
| W1 | 1140H085 | 91 | 256.1 | 0.38 | 699.3 | 110 | 0.994 | 1.33 | 0.395 | 0.386 | 1.0 | 90.7 | 4.8 |
| W2 | 1140H086 | 124 | 260.0 | 0.48 | 549.2 | 79 | 0.933 | 1.86 | 0.423 | 0.456 | 0.6 | 68.5 | 1.5 |
| W3 | 1300H013 | 169 | 291.2 | 0.57 | 526.7 | 147 | 0.875 | 7.63 | 0.381 | 0.686 | 1.0 | 89.9 | 4.3 |
| W4 | 1340H008 | 74 | 247.4 | 0.38 | 712.3 | 298 | 1.037 | 3.99 | 0.214 | 0.427 | 1.4 | 80.9 | 9.9 |
| W5 | 1350H001 | 127 | 260.8 | 0.51 | 557.7 | 244 | 1.073 | 4.56 | 0.266 | 0.503 | 0.8 | 83.3 | 10.4 |
| W6 | 1350H012 | 77 | 241.7 | 0.37 | 764.5 | 471 | 1.030 | 2.84 | 0.208 | 0.394 | 1.4 | 74.6 | 13.5 |
| W7 | 1420H034 | 208 | 286.4 | 0.72 | 404.8 | 105 | 0.856 | 10.19 | 0.355 | 0.648 | 0.9 | 92.1 | 3.3 |
| W8 | 1430H028 | 36 | 201.0 | 0.22 | 1109.3 | 265 | 1.191 | 1.18 | 0.203 | 0.545 | 2.4 | 41.1 | 29.4 |
| W9 | 1430H030 | 131 | 269.1 | 0.55 | 561.0 | 1043 | 0.962 | 2.36 | 0.285 | 0.399 | 1.1 | 69.0 | 20.6 |
| W10 | 1510H063 | 204 | 277.8 | 0.74 | 383.6 | 2089 | 0.924 | 2.22 | 0.432 | 0.421 | 0.5 | 84.8 | 4.3 |
| W11 | 1540H014 | 7 | 200.0 | 0.05 | 3200.0 | 83 | 1.285 | 2.85 | 0.097 | 0.304 | 0.0 | 25.0 | 7.7 |
| W12 | 1540H029 | 4 | 180.0 | 0.03 | 3600.0 | 220 | 1.539 | 1.14 | 0.103 | 0.424 | 3.0 | 18.8 | 53.2 |
| W13 | 1580H001 | 148 | 282.8 | 0.52 | 545.3 | 81 | 1.157 | 6.66 | 0.391 | 0.541 | 1.9 | 11.8 | 70.9 |
| W14 | 1660H010 | 23 | 208.8 | 0.12 | 1951.2 | 140 | 1.350 | 0.29 | 0.182 | 0.338 | 3.0 | 56.2 | 22.4 |
| W15 | 1730H031 | 211 | 280.7 | 0.75 | 375.9 | 812 | 0.915 | 3.09 | 0.426 | 0.321 | 0.7 | 85.5 | 3.1 |
| W16 | 2200H011 | 167 | 268.3 | 0.65 | 457.1 | 1573 | 0.919 | 2.36 | 0.383 | 0.433 | 2.6 | 59.2 | 19.4 |
| W17 | 2370H017 | 157 | 260.8 | 0.65 | 475.8 | 1527 | 0.945 | 2.91 | 0.329 | 0.459 | 1.9 | 79.7 | 9.5 |
| W18 | 2420H043 | 148 | 260.0 | 0.64 | 518.7 | 563 | 1.015 | 4.51 | 0.349 | 0.445 | 1.0 | 75.5 | 12.1 |
| W19 | 2560H001 | 188 | 269.1 | 0.69 | 424.9 | 450 | 0.934 | 5.25 | 0.335 | 0.473 | 1.9 | 88.8 | 2.3 |

Here, $H$ is the flow-path height [L], $L$ is the flow-path length [L], $G$ is the flow-path gradient [-], $A$ is the drainage area [L²], $DD$ is the drainage density [L/L²], $S_m$ is the gradient of mainstream, $HI$ is the hypsometric integral [-], ELO is the basin elongation [-], $C_W$, $C_F$, $C_A$ is the land cover of water, forest, and agriculture [-].

**Table S3.** Definition and calculation of hydrologic event and landscape variables.

| Variable | Definition and meaning | Calculation method |
|---|---|---|
| **Hydrologic event** | | |
| $AP_{7day}$ [mm] | 7-day antecedent precipitation could be used to present the saturation status of the watershed before the rainstorm. | Sum of rainfall amounts over the previous seven days leading up to the start of the rising limb. |
| $P$ [mm] | Total precipitation describes the magnitude of a rainstorm. | Sum of rainfall amounts throughout the defined rainfall period[a] |
| $D$ [hr] | Duration of precipitation indicates how long does the rainstorm last. | Length of time between the start and end of the defined rainfall period. |
| $I_{avg}$ [mm hr$^{-1}$] | Averaged precipitation intensity presents the magnitude of rainstorm intensity. | $P/D$ |
| $Q_{tot}$ [mm] | Total streamflow represents how much water is exported during a rainstorm | Sum of flow rates during the rainstorm. |
| $Q_{ant}$ [mm] | Antecedent streamflow. Recorded flow rate before the start of the rising limb. | |
| $Q_p$ [mm] | Peak flow. The highest recorded flow rate during a rainstorm. | |
| $Q_{tot}/P$ [-] | Ratio of total streamflow to precipitation, also called runoff coefficient. It indicates the efficiency of the conversion from rainfall to runoff. | |
| **Landscape** | | |
| $H$ [m] | Median of flow path heights within a catchment, which is related to potential energy of water. | Compute the elevation differences between hillslope cells and stream cell along the flow path. Then, determine the median of these difference across the catchment. |
| $L$ [m] | Median of flow path lengths within a catchment, which is related to flow accumulation from hillslopes. | Compute the distances between hillslope cells and stream cell along the flow path. Then, determine the median of these distances across the catchment. |
| $G$ [-] | Median of flow path gradients within a catchment, which could be regarded as a surrogate of flow velocity. | Calculate the gradients between hillslope cells and the stream cell along the flow path. Then, ascertain the median of these gradients across the catchment. |
| $L/G$ [m] | Median of ratios between flow-path length and gradient within a catchment, which is related to the mean residence time. | Compute the ratios of flow path length to gradient for each cell. Then, determine the median of these ratios across the catchment. |
| $A$ [km$^2$] | Drainage area, which could be linked to how much total water volume could be stored. | DEM cell size multiplied by the number of cells that can route to the outlet. |
| $DD$ [km km$^{-2}$] | Drainage density. It is related to how faster the catchment can drain water via stream. | Ratio of total stream length to the drainage area |
| $S_m$ [%] | Gradient of main stem, which is related to water velocity in main stem. | The changes in elevation along the main stem. |
| $HI$ [-] | Hypsometric integral. It represents how much a catchment can contain water storage. | Calculate the area under the hypsometric curve, which relates elevation and cumulative area |
| $ELO$ [-] | Basin elongation measures catchment shape and affects surface flow travel time. | Measure the ratio of the length of the longest axis of a catchment to the length of the perpendicular axis across it. |
| $C_W$ [%] | Water bodies coverage, which is negatively related to the recession exponent. | The area occupied by water bodies divided by drainage area. |
| $C_F$ [%] | Forests coverage, which is negatively related to the recession coefficient. | The area occupied by forest divided by drainage area. |
| $C_A$ [%] | Agriculture land coverage, which is related to the field capacity. | The area occupied by agriculture land divided by drainage area. |

[a]Rainfall period is defined as the elapsed time from 6 h before the rising flow to the peak flow.

[b]Flow path is defined as the trajectory taken by water from a hillslope grid point, as it follows the surface flow direction toward the channel.

**Table S4**. Descriptions of the selected catchments and events

[revised manuscript text omitted]

---

## Author Response (ED3)

**Reply to Editor's Comment**

Thank you for your corrections to the previous version of the manuscript. I spent quite some time going over your manuscript again and found that while your revision is an improvement, several issues still need to be clarified:

**Reply:** Thanks for your diligent review of our manuscript. We have made revisions to both the main manuscript and the supplementary material based on your feedback. We now proceed to respond to these comments point-by-point.

L/G: median L /median G or median L/G? There is a difference. In McGuire 2005 they use median L/median G. You say in Table S3 that you use median (L/G) – why? Explain what this means for your results. Especially as you are now using a different measure than previous studies.

**Reply:** Thank you for bringing up this issue. We have included additional sentences to emphasize the distinction between the approaches in L/G and our rationale [L89-92]:

"Please be aware that we defined the median of $l_{fp}/g_{fp}$ (*L/G*) in our study, differing from McGuire et al.'s (2005) previous study, which directly used the median $l_{fp}$ divided by the median $g_{fp}$. In the hydrological context, $l_{fp}/g_{fp}$ represents the residence time of each flow path and the median of $l_{fp}/g_{fp}$ characterizes the flow paths within a catchment, while the median of $l_{fp}/g_{fp}$ reflects catchment-wide residence time."

Also – you should state in the methods that you are using median values. Throughout the text it is not clear when you are talking about L or G or H if you are referring to the median or not. This needs to be clarified.

**Reply:** In order to distinguish whether the description refers to individual values or medians of flow path characteristics, we have employed two distinct labels. We have made revisions to certain sentences in [L84-89]:

"Specifically, flow path length ($l_{fp}$) is the route length from a cell to channel cell, flow path height ($h_{fp}$) is the elevation difference between the specific cell to channel cell, and flow path gradient ($g_{fp}$) is calculated as flow path height divided by flow path length. Each cell possesses its own value of $l_{fp}$, $h_{fp}$, $g_{fp}$, and $l_{fp}/g_{fp}$. Since the velocity of gravity-driven flow is typically proportional to the gradient ($v = l_{fp}/T \sim h_{fp}/l_{fp}$), this implies that the time (*T*) is proportional to $l_{fp}/g_{fp}$: $T \sim l_{fp}^2/h_{fp} = l_{fp}/g_{fp}$. Consequently, $l_{fp}/g_{fp}$ could serve as a potential proxy for residence time. Within a catchment, the medians of the $l_{fp}$, $h_{fp}$, and $g_{fp}$ distributions (*L*, *H*, and *G*) served as representative flow path characteristics."

Why is L/G a good proxy for residence time? If you base this on Darcy's law it would be good to explain this better than the currently slightly confusing sentence: "Note that G can also be regarded as a surrogate of flow velocity (most equations used for estimating flow velocity needs gradient to represent the conversion from potential to kinetic energy). Therefore, the composite ratio, L/G [m], can be a proxy for residence time (McGuire et al., 2005; Tetzlaff et al., 2009) and as a means to

comprehend the interplay between landscape features and climate impacts on residence time (Seybold et al., 2017)." Instead I would suggest something similar to this (if this is indeed what you mean): As the velocity of gravity-driven flow is usually proportional to the gradient: v = L/T ~ H/L this results in time T being proportional to L/G: T~L^2/H=L/G. Therefore, L/G could be a potential proxy for residence time. In terms of a catchment this relates to the ratio of the medians of L and G.

**Reply:** We appreciated the editor provide the thoughtful explanation of L/G as the proxy of residence time. We rephrased the sentences in [L86-88]:

"Since the velocity of gravity-driven flow is typically proportional to the gradient ($v = l_{fp}/T \sim h_{fp}/l_{fp}$), this implies that the time ($T$) is proportional to $l_{fp}/g_{fp}$: $T \sim l_{fp}^2/h_{fp} = l_{fp}/g_{fp}$. Consequently, $l_{fp}/g_{fp}$ could serve as a potential proxy for residence time."

What do you mean by composite ratio? What is the difference between composite ratio and ratio? Why is a simple ratio not enough? This needs to be explained.

**Reply:** Upon careful consideration, we have opted to directly remove the term "composite" in [L90], [L289] and [L302].

You are citing that Harman et al. 2009 found that heterogeneity between hillslopes increased with catchment area. However, they compare a single hillslope to a catchment of 10 ha and a catchment with a catchment area of 41 ha, so a maximum scale of not even 0.5 km². In your study, catchment areas only begin at 77 km². Does this relationship still hold at this very different spatial scale? This should be discussed.

**Reply:** We have added one sentence in [L266-270]:

"The weak correlation between recession nonlinearity and those variables might be explained by: First is the scale effect. Some of our catchments are much larger than 500 km², which far exceeds the extent of rainstorms (usually less than 200 km²). In these large catchments, the limited extent of rainstorms would not bring about a comprehensive recession response in the outflow hydrograph (Huang et al., 2012). Second, the drainage area cannot reflect the unknown number of aquifers (Ajami et al., 2011), making it unclear whether a positive relationship exists between nonlinearity and drainage area in our study."

Table 3 in the supplement – here the language can be simplified/clarified and there are also still some expressions that should be corrected:
- Definition of DD: fast not faster
- Water body coverage
- Forest coverage
- Agricultural land coverage

**Reply:** Thanks for the correction. We have enhanced the language in Table S3.

Please also see my comments to the supplementary material and within the pdf of your response.

**Reply:** Thank you for your thorough review of the supplementary material. Your advice has greatly

improved the supplement, and we have incorporated the revisions accordingly.

I am glad to see the major improvements in this manuscript over the course of the review and am hoping that we can resolve these issues with a last round of minor revisions. Given that there are still some issues with the language I have been in touch with the English copy-editing department of Copernicus and they will assist you with final improvements on this front. Unfortunately, even your revised abstract still needs work, as the sentences have the tendency to be convoluted and confusing. Looking forward to bringing this review process to a close and moving your manuscript forward towards publication!

**Reply:** We again revised the abstract [L10-22]:

"Streamflow recession, shaped by hydrological processes, runoff dynamics, and catchment storage, is heavily influenced by landscape structure and rainstorm. However, our understanding of how recession relates to landscape structure and rainstorm remains inconsistent, with limited research examining their combined impact. This study examines  in shaping recession responses,  291 sets of recession parameters obtained through the decorrelation process. The data originates from 19 subtropical mountainous rivers  a wide spectrum of rainfall amounts. Key findings indicate that the recession coefficient (*a*) increases while the exponent (*b*) decreases with the *L/G* ratio (the median of ratios between flow-path length and gradient), suggesting that longer and gentler hillslopes facilitate flow accumulation and aquifer connectivity, ultimately reducing nonlinearity. Additionally, in large catchments, the exponent (*b*) rises with increasing rainfall due to greater landscape heterogeneity, whereas in small catchments, it declines with rainfall, likely indicating catchment is prone to saturated and thus reduced runoff heterogeneity. Our  underscores the necessity for further validation across diverse regions "

**SPECIFIC COMMENTS**

(in Response)

*"defined as."* [L85]
Delete.
**Reply:** Revised as the suggestion.

*"Our variable H"* [L279]
Our variable H likely suggests... not clear. Do you mean: The fact that H is negatively correlated with the recession coefficient suggests that...? Also why the stress on it being your variable?
**Reply:** We rephrased this sentence based on the editor's comment [L283-285]: "The fact that *H* is

negatively correlated with the recession coefficient suggests that our groundwater flow paths possess greater depth and length, consequently leading to slower drainage rates."

"it does not necessarily correspond to hydraulic gradient due to the geologic and soil settings varying across regions" [L282]

I don't understand this. H/L is gradient, not H alone. Are you talking about the hydraulic gradients of GW level to stream? That can of course be different to the surface gradient to stream and depends on geology etc. Please clarify here what you mean.

**Reply:** We rephrased this sentence in [L285-288]: "While *H* is commonly believed to be positively correlated with the velocity of gravity-driven flow at a small spatial scale, the high heterogeneity in  geology or soil properties at a larger spatial scale (Karlsen et al., 2019) implies that a large *H* does not necessarily lead to a large recession coefficient."

"While the equivalent composite ratio can result from either L or G," [L288]

This sentence is not clear. equivalent to what? not sure what results from either L or G is supposed to mean, either.

**Reply:** We rephrased this sentence in [L291-293]: "Potentially, the relationship between recession parameters and *L/G* has the chance to establish a further linkage between recession parameters and water residence time.".

"Explain why this can serve as a proxy for residence time and what you mean by composite ratio."

see comment in separate document

**Reply:** Replied in the general comment, seeing revised [L84-92].

"recession event flow" [L330]

event recession flows (at least for me it seems like event recession makes more sense than recession event - here it is not clear how exactly a recession event is defined.)

**Reply:** Revised as the suggestion [L334].

"responses to" [L10]

relationships with

**Reply:** Revised as the suggestion.

"rainstorms" [L10]

characteristics

**Reply:** Revised as the suggestion.

(in the Supplementary Material)

Table S1: "un-decorrelated"

is this really a word? What about using "original values" or "values before decorrelation"?

**Reply:** We have replaced "un-decorrelated" with "original values." [Table S1]

"CTS and VTS denote constant and various time interval of sampling (Q, dQ/dt) pair, respectively."
this does not make sense. Do you mean variable?

**Reply:** We rephrased this sentence: "CTS and VTS represent constant and variable time intervals for sampling (Q, dQ/dt) pairs, respectively." [Table S1]

Table S2: footnote.
state if these are medians or means or what sort of summary statistics you are using here.

**Reply:** We rephrased this footnote: "Here, $H$ is the median of flow-path heights, $L$ is the median of flow-path lengths, $G$ is the median of flow-path gradients, L/G is the median of ratios between flow-path length and gradient. $A$ is the drainage area, $DD$ is the drainage density, $S_m$ is the gradient of main stem, $HI$ is the hypsometric integral, ELO is the basin elongation, $C_W$, $C_F$, $C_A$ is the coverage of water body, forest, and agricultural land, respectively." [Table S2]

Table S3:
this table still needs work language-wise.

**Reply:** We have enhanced the language in this table [Table S3].

Table S4:
remove the large space between the median and the max/min.
State in caption that max and min are given in brackets or provide separate headers for median, max and min

**Reply:** We removed the large space and also separated the headers of median, min, and max. [Table S4]

Reference
References

**Reply:** Revised as the suggestion.

**Supplementary Material**

**Table S1.** Summary of empirical power-law recession studies. The number of references corresponds to Table 1 in the main text. The parameter a and â represent decorrelated and original values, respectively. $T_0$ represents recession timescale at the median flow. CTS and VTS represent constant and variable time intervals for sampling (Q, dQ/dt) pairs, respectively.

| No | Reference | Data pool | Temporal scale | Location | Number of basins | Number of events | Basin area (km²) | Unit of flow | Initial time of recession segment (day after $Q_p$) | Sampling way (Q, dQ/dt) | $b$ | Target parameters |
|---|---|---|---|---|---|---|---|---|---|---|---|---|
| 1 | Mathias et al. (2016) | Point-cloud | Long-term | UK | 120 | n.a. | 1.1-1700 | $L\ T^{-1}$ | 0 | CTS | 1.68-1.99 | â, b |
| 2 | Patnaik et al. (2018) | Median | Long-term | Eastern USA | 212 | n.a. | n.a. | $L^3\ T^{-1}$ | 1 | CTS | 1-6 | b |
| 3 | Tashie et al. (2019) | Median | Monthly | North Carolina | 1 | 382 | 0.6 | $L\ T^{-1}$ | 1 | CTS | 4-20 | a, b |
| 4 | Bart and Hope (2014) | Events | Event | California | 4 | n.a. | 119-632 | $L\ T^{-1}$ | 7 | CTS | 1.8-2.1 | â |
| 5 | Biswal and Nagesh Kumar (2014) | Events | Event | USA | 67 | n.a. | 10-8858 | $L^3\ T^{-1}$ | 0 | CTS | 1.47-4.57 | â |
| 6 | Biswal and Marani (2014) | Events | Event | Eastern USA | 4 | n.a. | 41-583 | $L^3\ T^{-1}$ | 1 | CTS | 1.91-2.23 | â |
| 7 | Clark et al. (2009) | Point-cloud | Long-term/event | Georgia | 3 | n.a. | 0.001-0.41 | $L\ T^{-1}$ | 0 | VTS | 1-3 | b |
| 8 | Ghosh et al. (2016) | Events | Event | Georgia | 1 | 23 | 0.41 | $L\ T^{-1}$ | 0.25 | CTS | 2.5-7.8 | â, b |
| 9 | Patnaik et al. (2015) | Median/Events | Long-term/Event | USA | 358 | n.a. | 2-3247 | $L^3\ T^{-1}$ | 7 | CTS | n.a. | â |
| 10 | Millares et al. (2009) | Point-cloud | Long-term | Spain | 3 | n.a. | n.a. | $L^3\ T^{-1}$ | 0 | CTS | 1.15-1.30 | â |
| 11 | Sayama et al. (2011) | Point-cloud | Long-term | California | 17 | n.a. | 3-112 | $L\ T^{-1}$ | 0 | CTS | n.a. | b |
| 12 | Shaw and Riha (2012) | Events | Event | New York | 7 | 80 | 100-6415 | $L^3\ T^{-1}$ | 0 | VTS | 1.31-5.34 | â |
| 13 | Shaw et al. (2013) | Events | Event | New York | 9 | 72 | 287 | $L^3\ T^{-1}$ | 0 | VTS | 0.98-2.42 | â |
| 14 | Tague et al. (2004) | Point-cloud | Long-term | Oregon | 22 | n.a. | 7.3-1337 | $L^3\ T^{-1}$ | 0 | CTS | 1.38-3.16 | â, b |
| 15 | Tashie et al. (2020) | Events | Event | USA | 1027 | 155309 | n.a. | $L^3\ T^{-1}$ | 0 | CTS | 1.1-7.3 | b |
| 16 | Yan et al. (2022) | Point-cloud | Long-term | Eastern China | 382 | n.a. | 34-18211 | $L^3\ T^{-1}$ | 2 | CTS | 0.57-3 | â, b |
| 17 | Ye et al. (2014) | Point-cloud | Long-term | Eastern USA | 50 | n.a. | 66-9062 | $L\ T^{-1}$ | 3 | CTS | 0.99-1.91 | â, b |
| 18 | McMillan et al. (2014) | Median/Point-cloud | Long-term/monthly/event | New Zealand | 28 | n.a. | n.a. | $L\ T^{-1}$ | 0.5 | VTS | 1.5-4.0 | $T_0$, b |
| 19 | Biswal and Nagesh Kumar (2013) | Events | Event | USA | 39 | 5486 | 9.6-5457 | $L^3\ T^{-1}$ | 0 | CTS | 1.52-2.61 | b |
| 20 | Chen and Krajewski (2015) | Events | Event | Iowa | 25 | n.a. | 66-16854 | $L\ T^{-1}$ | 12 | CTS | 0.75-1.6 | â, b |
| 21 | Bogaart et al. (2016) | Point-cloud | Annual | Sweden | 316 | n.a. | 3-33000 | $L\ T^{-1}$ | 3 | CTS | 0.5-2.1 | â, b |

| | | | | | | | | | | |
|---|---|---|---|---|---|---|---|---|---|---|
| 22 | Dralle et al. (2017) | Events | Event | California/Oregon | 16 | n.a. | 17-5457 | $L^3 T^{-1}$ | vary | CTS | 0.1-3.7 | a |
| 23 | Santos et al. (2019) | Events | Annual/Event | Switzerland | 5 | n.a. | 50-352 | $L T^{-1}$ | vary | CTS | 1.73-2.4 | a, b |
| 24 | Karlsen et al. (2019) | Events | Seasonal/Event | Northern Sweden | 14 | 163 | 12-6790 | $L T^{-1}$ | 2 | VTS | 1-10 | $T_0$, b |

**Table S2.** Landscape and landcover variables of the selected catchments.

| ID | HID | *H* (m) | *L* (m) | *G* (-) | *L/G* (m) | *A* (km²) | *DD* (km/km²) | $S_m$ (%) | *HI* (-) | *ELO* (-) | $C_W$ (%) | $C_F$ (%) | $C_A$ (%) |
|---|---|---|---|---|---|---|---|---|---|---|---|---|---|
| W1 | 1140H085 | 91 | 256.1 | 0.38 | 699.3 | 110 | 0.994 | 1.33 | 0.395 | 0.386 | 1.0 | 90.7 | 4.8 |
| W2 | 1140H086 | 124 | 260.0 | 0.48 | 549.2 | 79 | 0.933 | 1.86 | 0.423 | 0.456 | 0.6 | 68.5 | 1.5 |
| W3 | 1300H013 | 169 | 291.2 | 0.57 | 526.7 | 147 | 0.875 | 7.63 | 0.381 | 0.686 | 1.0 | 89.9 | 4.3 |
| W4 | 1340H008 | 74 | 247.4 | 0.38 | 712.3 | 298 | 1.037 | 3.99 | 0.214 | 0.427 | 1.4 | 80.9 | 9.9 |
| W5 | 1350H001 | 127 | 260.8 | 0.51 | 557.7 | 244 | 1.073 | 4.56 | 0.266 | 0.503 | 0.8 | 83.3 | 10.4 |
| W6 | 1350H012 | 77 | 241.7 | 0.37 | 764.5 | 471 | 1.030 | 2.84 | 0.208 | 0.394 | 1.4 | 74.6 | 13.5 |
| W7 | 1420H034 | 208 | 286.4 | 0.72 | 404.8 | 105 | 0.856 | 10.19 | 0.355 | 0.648 | 0.9 | 92.1 | 3.3 |
| W8 | 1430H028 | 36 | 201.0 | 0.22 | 1109.3 | 265 | 1.191 | 1.18 | 0.203 | 0.545 | 2.4 | 41.1 | 29.4 |
| W9 | 1430H030 | 131 | 269.1 | 0.55 | 561.0 | 1043 | 0.962 | 2.36 | 0.285 | 0.399 | 1.1 | 69.0 | 20.6 |
| W10 | 1510H063 | 204 | 277.8 | 0.74 | 383.6 | 2089 | 0.924 | 2.22 | 0.432 | 0.421 | 0.5 | 84.8 | 4.3 |
| W11 | 1540H014 | 7 | 200.0 | 0.05 | 3200.0 | 83 | 1.285 | 2.85 | 0.097 | 0.304 | 0.0 | 25.0 | 7.7 |
| W12 | 1540H029 | 4 | 180.0 | 0.03 | 3600.0 | 220 | 1.539 | 1.14 | 0.103 | 0.424 | 3.0 | 18.8 | 53.2 |
| W13 | 1580H001 | 148 | 282.8 | 0.52 | 545.3 | 81 | 1.157 | 6.66 | 0.391 | 0.541 | 1.9 | 11.8 | 70.9 |
| W14 | 1660H010 | 23 | 208.8 | 0.12 | 1951.2 | 140 | 1.350 | 0.29 | 0.182 | 0.338 | 3.0 | 56.2 | 22.4 |
| W15 | 1730H031 | 211 | 280.7 | 0.75 | 375.9 | 812 | 0.915 | 3.09 | 0.426 | 0.321 | 0.7 | 85.5 | 3.1 |
| W16 | 2200H011 | 167 | 268.3 | 0.65 | 457.1 | 1573 | 0.919 | 2.36 | 0.383 | 0.433 | 2.6 | 59.2 | 19.4 |
| W17 | 2370H017 | 157 | 260.8 | 0.65 | 475.8 | 1527 | 0.945 | 2.91 | 0.329 | 0.459 | 1.9 | 79.7 | 9.5 |
| W18 | 2420H043 | 148 | 260.0 | 0.64 | 518.7 | 563 | 1.015 | 4.51 | 0.349 | 0.445 | 1.0 | 75.5 | 12.1 |
| W19 | 2560H001 | 188 | 269.1 | 0.69 | 424.9 | 450 | 0.934 | 5.25 | 0.335 | 0.473 | 1.9 | 88.8 | 2.3 |

Here, *H* is the median of flow-path heights, *L* is the median of flow-path lengths, *G* is the median of flow-path gradients, L/G is the median of ratios between flow-path length and gradient. *A* is the drainage area, *DD* is the drainage density, $S_m$ is the gradient of main stem, *HI* is the hypsometric integral, *ELO* is the basin elongation, $C_W$, $C_F$, $C_A$ is the coverage of water body, forest, and agricultural land, respectively.

**Table S3.** Definition and calculation of hydrologic event and landscape variables.

| Variable | Definition and meaning | Calculation method |
|---|---|---|
| **Hydrologic event** | | |
| $AP_{7day}$ [mm] | 7-day antecedent precipitation  used to  the saturation status of the watershed before the rainstorm. | Sum of rainfall amounts over the previous seven days leading up to the start of the rising limb. |
| $P$ [mm] | Total precipitation . | Sum of rainfall amounts throughout the defined rainfall period[a] |
| $D$ [hr] | Duration of precipitation . | Length of time between the start and end of the defined rainfall period. |
| $I_{avg}$ [mm hr$^{-1}$] | Averaged precipitation intensity . | $P/D$ |
| $Q_{tot}$ [mm] | Total streamflow represents how much water is exported during a rainstorm | Sum of flow rates during the rainstorm. |
| $Q_{ant}$ [mm] | Antecedent streamflow. Recorded flow rate before the start of the rising limb. | |
| $Q_p$ [mm] | Peak flow. The highest recorded flow rate during a rainstorm. | |
| $Q_{tot}/P$ [-] | Ratio of total streamflow to precipitation, also called runoff coefficient. It indicates the efficiency of the conversion from rainfall to runoff. | |
| **Landscape** | | |
| $H$ [m] | Median of flow path heights, which is related to the potential energy of water. | Compute the elevation differences between hillslope cells and stream cell along the flow path. Then, determine the median of these difference across the catchment. |
| $L$ [m] | Median of flow path lengths, which is related to flow accumulation from hillslopes. | Compute the distances between hillslope cells and stream cell along the flow path. Then, determine the median of these distances across the catchment. |
| $G$ [-] | Median of flow path gradients, which could be regarded as a surrogate of flow velocity. | Calculate the gradients between hillslope cells and the stream cell along the flow path. Then, ascertain the median of these gradients across the catchment. |
| $L/G$ [m] | Median of ratios between flow-path length and gradient, which is related to the mean residence time. | Calculate the ratios of flow path length to gradient for each cell, and subsequently, determine the median of these ratios across the entire catchment. |
| $A$ [km$^2$] | Drainage area, which could be linked to how much total water volume could be stored. | Total area of cells that can route to the outlet. |
| $DD$ [km km$^{-2}$] | Drainage density. It is related to how fast the catchment can drain water via stream. | Ratio of total stream length to the drainage area |
| $S_m$ [%] | Gradient of main stem, which is related to water velocity in main stem. | The changes in elevation along the main stem. |
| $HI$ [-] | Hypsometric integral. It represents how much a catchment can contain water storage. | Calculate the area under the hypsometric curve, which relates elevation and cumulative area |
| $ELO$ [-] | Basin elongation measures catchment shape and affects surface flow travel time. | Measure the ratio of the length of the longest axis of a catchment to the length of the perpendicular axis across it. |
| $C_W$ [%] | Water body coverage, which is negatively related to the recession exponent. | Percentage of the area of water bodies divided by drainage area. |
| $C_F$ [%] | Forest coverage, which is negatively related to the recession coefficient. | Percentage of the forest area divided by drainage area. |
| $C_A$ [%] | Agricultural land coverage, which is related to the field capacity. | Percentage of the agricultural area divided by drainage area. |

[a]Rainfall period is defined as the elapsed time from 6 h before the rising flow to the peak flow.

[b]Flow path is defined as the trajectory taken by water from a hillslope grid point, as it follows the surface flow direction toward the channel.

**Table S4**. Descriptions of the selected catchments and events

| ID | HID | N | $AP_{7day}$ (mm) | | | $P$ (mm) | | | $I_{avg}$ (mm h$^{-1}$) | | | $Q_{ant}$ (mm h$^{-1}$) | | | $Q_{tot}$ (mm) | | | $Q_{p}$ (mm h$^{-1}$) | | | $Q_{tot}/P$ (-) | | |
|---|---|---|---|---|---|---|---|---|---|---|---|---|---|---|---|---|---|---|---|---|---|---|---|
| | | | median | min | max | median | min | max | median | min | max | median | min | max | median | min | max | median | min | max | median | min | max |
| W1 | 1140H085 | 15 | 70 | 3 | 282 | 246 | 131 | 904 | 7.3 | 3.7 | 11.7 | 0.32 | 0.02 | 2.35 | 205 | 100 | 697 | 8.7 | 4.9 | 27.2 | 0.76 | 0.49 | 1.05 |
| W2 | 1140H086 | 18 | 60 | 1 | 294 | 272 | 98 | 854 | 7.1 | 3.1 | 17.5 | 0.24 | 0.02 | 1.91 | 190 | 99 | 650 | 8.8 | 4.6 | 27.1 | 0.80 | 0.52 | 1.04 |
| W3 | 1300H013 | 16 | 56 | 3 | 248 | 239 | 25 | 1012 | 10.0 | 5.6 | 23.6 | 0.34 | 0.07 | 2.16 | 111 | 26 | 537 | 8.6 | 0.9 | 37.5 | 0.52 | 0.26 | 1.02 |
| W4 | 1340H008 | 21 | 51 | 0 | 498 | 206 | 58 | 865 | 7.4 | 3.9 | 21.6 | 0.13 | 0.01 | 1.06 | 122 | 16 | 670 | 12.6 | 1.6 | 31.8 | 0.71 | 0.23 | 1.00 |
| W5 | 1350H001 | 20 | 53 | 0 | 272 | 352 | 127 | 1247 | 5.4 | 1.3 | 13.4 | 0.19 | 0.05 | 0.89 | 191 | 51 | 749 | 10.5 | 2.4 | 52.2 | 0.61 | 0.20 | 0.94 |
| W6 | 1350H012 | 18 | 45 | 9 | 489 | 336 | 155 | 596 | 5.9 | 3.5 | 11.3 | 0.14 | 0.01 | 0.86 | 221 | 55 | 424 | 8.0 | 2.1 | 32.9 | 0.54 | 0.24 | 1.06 |
| W7 | 1420H034 | 11 | 38 | 4 | 186 | 558 | 189 | 651 | 10.3 | 5.4 | 12.1 | 0.34 | 0.08 | 1.03 | 302 | 104 | 691 | 11.9 | 4.9 | 22.8 | 0.63 | 0.32 | 1.08 |
| W8 | 1430H028 | 26 | 81 | 4 | 355 | 343 | 89 | 934 | 6.5 | 3.6 | 13.8 | 0.23 | 0.12 | 0.46 | 138 | 38 | 458 | 13.6 | 4.0 | 65.4 | 0.41 | 0.21 | 0.70 |
| W9 | 1430H030 | 13 | 84 | 3 | 923 | 415 | 87 | 674 | 4.7 | 1.9 | 8.7 | 0.40 | 0.11 | 0.76 | 150 | 43 | 446 | 3.6 | 1.5 | 14.0 | 0.34 | 0.23 | 1.03 |
| W10 | 1510H063 | 15 | 31 | 8 | 102 | 471 | 105 | 1276 | 6.2 | 2.6 | 10.4 | 0.11 | 0.05 | 0.70 | 237 | 46 | 964 | 5.8 | 1.6 | 19.1 | 0.51 | 0.21 | 0.91 |
| W11 | 1540H014 | 9 | 80 | 17 | 187 | 164 | 85 | 364 | 7.1 | 3.0 | 10.3 | 0.25 | 0.03 | 0.63 | 137 | 30 | 304 | 13.0 | 3.2 | 22.2 | 0.73 | 0.36 | 1.10 |
| W12 | 1540H029 | 12 | 69 | 13 | 237 | 158 | 28 | 581 | 6.2 | 4.0 | 11.9 | 0.33 | 0.19 | 0.70 | 112 | 16 | 591 | 8.4 | 1.6 | 28.1 | 0.75 | 0.27 | 1.02 |
| W13 | 1580H001 | 24 | 65 | 2 | 396 | 712 | 61 | 2558 | 9.7 | 4.2 | 20.3 | 0.44 | 0.04 | 1.27 | 368 | 37 | 1736 | 24.9 | 1.6 | 84.5 | 0.56 | 0.25 | 1.08 |
| W14 | 1660H010 | 17 | 80 | 11 | 707 | 201 | 24 | 982 | 6.6 | 3.2 | 13.6 | 0.16 | 0.02 | 3.22 | 137 | 14 | 946 | 11.7 | 1.7 | 27.4 | 0.72 | 0.31 | 1.10 |
| W15 | 1730H031 | 10 | 106 | 26 | 317 | 507 | 186 | 820 | 7.6 | 4.8 | 17.0 | 0.28 | 0.11 | 0.66 | 254 | 101 | 628 | 9.8 | 2.2 | 28.8 | 0.67 | 0.38 | 1.00 |
| W16 | 2200H011 | 10 | 66 | 21 | 175 | 236 | 65 | 716 | 4.9 | 2.4 | 9.4 | 0.20 | 0.03 | 0.92 | 156 | 27 | 583 | 5.2 | 1.0 | 18.8 | 0.67 | 0.25 | 0.99 |
| W17 | 2370H017 | 10 | 28 | 4 | 124 | 456 | 225 | 840 | 5.2 | 4.4 | 10.8 | 0.10 | 0.05 | 0.68 | 369 | 59 | 512 | 10.9 | 2.6 | 21.3 | 0.76 | 0.22 | 1.11 |
| W18 | 2420H043 | 21 | 49 | 0 | 358 | 333 | 102 | 813 | 4.9 | 2.7 | 13.8 | 0.30 | 0.01 | 0.85 | 187 | 34 | 602 | 10.1 | 1.5 | 47.6 | 0.58 | 0.28 | 0.98 |
| W19 | 2560H001 | 5 | 58 | 48 | 59 | 255 | 196 | 484 | 5.0 | 4.1 | 9.5 | 0.12 | 0.03 | 0.46 | 109 | 82 | 277 | 4.9 | 2.0 | 11.5 | 0.43 | 0.42 | 0.57 |
| | Average | | 62 | | | 340 | | | 6.7 | | | 0.24 | | | 194 | | | 10.1 | | | 0.61 | | |

*ID is the identifier of catchments in this study, HID is the identifier of catchments named by the Taiwan Water Resource Agency, N is the number of events. Values in each column present the median and range of the events in the corresponding catchments. Numbers in parentheses indicate the lower and upper limit among the events in the specific catchment.

**Table S5.** Median, minimum, and maximum values of the recession coefficient and exponent for each catchment.

| ID | HID | $a$ [hr$^{-1}$] | | | $b$ [-] | | | $1/a$ [h] | | |
|----|-----|--------|-----|-----|--------|-----|-----|--------|-----|-----|
| | | median | min | max | median | min | max | median | min | max |
| W1 | 1140H085 | 0.033 | 0.019 | 0.067 | 1.73 | 1.30 | 2.38 | 30.0 | 14.9 | 53.7 |
| W2 | 1140H086 | 0.035 | 0.018 | 0.049 | 1.82 | 1.30 | 2.38 | 28.8 | 20.4 | 54.2 |
| W3 | 1300H013 | 0.046 | 0.011 | 0.156 | 1.94 | 1.00 | 2.74 | 21.9 | 6.4 | 93.8 |
| W4 | 1340H008 | 0.074 | 0.028 | 0.172 | 1.62 | 1.19 | 1.99 | 13.6 | 5.8 | 35.2 |
| W5 | 1350H001 | 0.022 | 0.010 | 0.094 | 1.96 | 1.62 | 2.53 | 45.0 | 10.7 | 95.5 |
| W6 | 1350H012 | 0.068 | 0.020 | 0.129 | 1.56 | 0.90 | 1.92 | 14.6 | 7.8 | 50.0 |
| W7 | 1420H034 | 0.016 | 0.010 | 0.041 | 1.92 | 1.58 | 2.37 | 62.5 | 24.3 | 102.2 |
| W8 | 1430H028 | 0.068 | 0.025 | 0.166 | 1.63 | 1.26 | 2.39 | 14.6 | 6.0 | 40.3 |
| W9 | 1430H030 | 0.026 | 0.010 | 0.102 | 2.34 | 1.37 | 2.98 | 37.9 | 9.8 | 99.4 |
| W10 | 1510H063 | 0.031 | 0.013 | 0.116 | 1.51 | 1.12 | 2.05 | 32.6 | 8.7 | 77.4 |
| W11 | 1540H014 | 0.110 | 0.048 | 0.144 | 1.30 | 0.95 | 1.60 | 9.1 | 6.9 | 21.0 |
| W12 | 1540H029 | 0.089 | 0.052 | 0.156 | 1.63 | 0.91 | 2.95 | 11.2 | 6.4 | 19.4 |
| W13 | 1580H001 | 0.031 | 0.003 | 0.273 | 1.67 | 1.19 | 4.39 | 32.2 | 3.7 | 303.8 |
| W14 | 1660H010 | 0.094 | 0.049 | 0.218 | 1.29 | 1.05 | 1.63 | 10.6 | 4.6 | 20.6 |
| W15 | 1730H031 | 0.025 | 0.009 | 0.087 | 1.71 | 1.25 | 2.39 | 40.1 | 11.5 | 108.8 |
| W16 | 2200H011 | 0.036 | 0.026 | 0.164 | 1.74 | 1.32 | 1.96 | 28.1 | 6.1 | 38.0 |
| W17 | 2370H017 | 0.029 | 0.015 | 0.087 | 1.67 | 1.16 | 1.95 | 34.6 | 11.6 | 64.9 |
| W18 | 2420H043 | 0.054 | 0.020 | 0.180 | 1.60 | 0.97 | 2.21 | 18.4 | 5.6 | 49.0 |
| W19 | 2560H001 | 0.055 | 0.021 | 0.202 | 1.30 | 1.05 | 1.72 | 18.1 | 5.0 | 47.1 |

**References**

25   Bart, R., & Hope, A.: Inter-seasonal variability in baseflow recession rates: The role of aquifer antecedent storage in central California watersheds. Journal of Hydrology, 519, 205-213, https://doi.org/10.1016/j.jhydrol.2014.07.020, 2014.

Biswal, B., & Marani, M.: 'Universal'recession curves and their geomorphological interpretation. Advances in Water Resources, 65, 34-42, https://doi.org/10.1016/j.advwatres.2014.01.004, 2014.

Biswal, B., & Nagesh Kumar, D.: A general geomorphological recession flow model for river basins. Water Resources Research, 49(8), 4900-4906,
30   https://doi.org/10.1002/wrcr.20379, 2013.

Biswal, B., & Nagesh Kumar, D.: Study of dynamic behaviour of recession curves. Hydrological Processes, 28(3), 784-792, https://doi.org/10.1002/hyp.9604, 2014.

Bogaart, P. W., van der Velde, Y., Lyon, S. W., and Dekker, S. C.: Streamflow recession patterns can help unravel the role of climate and humans in landscape co-evolution, Hydrol. Earth Syst. Sci., 20, 1413–1432, https://doi.org/10.5194/hess-20-1413-2016, 2016.

35   Chen, B., & Krajewski, W. F.: Recession analysis across scales: The impact of both random and nonrandom spatial variability on aggregated hydrologic response. Journal of Hydrology, 523, 97-106, https://doi.org/10.1016/j.jhydrol.2015.01.049, 2015.

Clark, M. P., Rupp, D. E., Woods, R. A., Tromp-van Meerveld, H. J., Peters, N. E., & Freer, J. E.: Consistency between hydrological models and field observations: linking processes at the hillslope scale to hydrological responses at the watershed scale. Hydrological Processes: An International Journal, 23(2), 311-319, https://doi.org/10.1002/hyp.7154, 2009

40   Dralle, D. N., Karst, N. J., Charalampous, K., Veenstra, A., and Thompson, S. E.: Event-scale power law recession analysis: quantifying methodological uncertainty, Hydrol. Earth Syst. Sci., 21, 65–81, https://doi.org/10.5194/hess-21-65-2017, 2017.

Ghosh, D. K., Wang, D., & Zhu, T.: On the transition of base flow recession from early stage to late stage. Advances in Water Resources, 88, 8-13, https://doi.org/10.1016/j.advwatres.2015.11.015, 2016.

Karlsen, R. H., Bishop, K., Grabs, T., Ottosson-Löfvenius, M., Laudon, H., & Seibert, J.: The role of landscape properties, storage and
45   evapotranspiration on variability in streamflow recessions in a boreal catchment. Journal of Hydrology, 570, 315-328, https://doi.org/10.1016/j.jhydrol.2018.12.065, 2019.

Mathias, S. A., McIntyre, N., Oughton, R. H.: A study of non-linearity in rainfall-runoff response using 120 UK catchments. Journal of Hydrology, 540, 423-436, https://doi.org/10.1016/j.jhydrol.2016.06.039, 2016.

McMillan, H., Gueguen, M., Grimon, E., Woods, R., Clark, M., & Rupp, D. E.: Spatial variability of hydrological processes and model structure
50   diagnostics in a 50 km2 catchment. Hydrological Processes, 28(18), 4896-4913, https://doi.org/10.1002/hyp.9988, 2014.

Millares, A., Polo, M. J., and Losada, M. A.: The hydrological response of baseflow in fractured mountain areas, Hydrol. Earth Syst. Sci., 13, 1261–1271, https://doi.org/10.5194/hess-13-1261-2009, 2009.

Patnaik, S., Biswal, B., Kumar, D. N., & Sivakumar, B.: Effect of catchment characteristics on the relationship between past discharge and the power law recession coefficient. Journal of Hydrology, 528, 321-328, https://doi.org/10.1016/j.jhydrol.2015.06.032, 2015.

55  Patnaik, S., Biswal, B., Nagesh Kumar, D., & Sivakumar, B.: Regional variation of recession flow power-law exponent. Hydrological Processes, 32(7), 866-872, https://doi.org/10.1002/hyp.11441, 2018.

Santos, A. C., Portela, M. M., Rinaldo, A., & Schaefli, B.: Estimation of streamflow recession parameters: New insights from an analytic streamflow distribution model. Hydrological processes, 33(11), 1595-1609, https://doi.org/10.1002/hyp.13425, 2019.

Sayama, T., McDonnell, J. J., Dhakal, A., & Sullivan, K.: How much water can a watershed store?. Hydrological Processes, 25(25), 3899-3908,
60  https://doi.org/10.1002/hyp.8288, 2011.

Shaw, S. B., & Riha, S. J.: Examining individual recession events instead of a data cloud: Using a modified interpretation of dQ/dt–Q streamflow recession in glaciated watersheds to better inform models of low flow. Journal of hydrology, 434, 46-54, https://doi.org/10.1016/j.jhydrol.2012.02.034, 2012.

Shaw, S. B., McHardy, T. M., & Riha, S. J.: Evaluating the influence of watershed moisture storage on variations in base flow recession rates during
65  prolonged rain-free periods in medium-sized catchments in New York and Illinois, USA. Water Resources Research, 49(9), 6022-6028, https://doi.org/10.1002/wrcr.20507, 2013

Tague, C., & Grant, G. E.: A geological framework for interpreting the low-flow regimes of Cascade streams, Willamette River Basin, Oregon. Water Resources Research, 40(4), https://doi.org/10.1029/2003WR002629, 2004.

Tashie, A., Pavelsky, T., & Band, L. E.: An empirical reevaluation of streamflow recession analysis at the continental scale. Water Resources Research,
70  56(1), e2019WR025448, https://doi.org/10.1029/2019WR025448, 2020.

Tashie, A., Scaife, C. I., & Band, L. E.: Transpiration and subsurface controls of streamflow recession characteristics. Hydrological Processes, 33(19), 2561-2575, https://doi.org/10.1002/hyp.13530, 2019.

Yan, H., Hu, H., Liu, Y., Tudaji, M., Yang, T., Wei, Z., Chen, Z.: Characterizing the groundwater storage–discharge relationship of small catchments in China. Hydrology Research, 53(5), 782-794, https://doi.org/10.2166/nh.2022.023, 2022.

75  Ye, S., Li, H. Y., Huang, M., Ali, M., Leng, G., Leung, L. R., Sivapalan, M. Regionalization of subsurface stormflow parameters of hydrologic models: Derivation from regional analysis of streamflow recession curves. Journal of Hydrology, 519, 670-682, https://doi.org/10.1016/j.jhydrol.2014.07.017, 2014.

---

## Author Response (AR6)

**Reply to Editor's Comment**

There are still quite some issues with the writing/language of this manuscript that really affect the readers ability to understand what you want to say.

I have made some corrections and suggestions/remarks in the attached pdf.

Please work with the language editing service of Copernicus to improve the writing.

Otherwise all involved have worked really hard to improve this manuscript (reviewers and authors) and I am hoping it will find interest in the community!

**Reply:** Thank you for your feedback and for taking the time to review our manuscript. We appreciate your efforts in helping us improve the quality of our work. Your guidance is invaluable in making our research accessible and understandable to the readers. We will certainly work with the language editing service of Copernicus to further refine our manuscript's writing. It's essential for us to ensure that our work is presented in the best possible way.

**SPECIFIC COMMENTS**

*"Streamflow recession, shaped by hydrological processes, runoff dynamics, and catchment storage, is heavily influenced by landscape structure and rainstorm* characteristics*. However, our understanding of how recession relates to landscape structure and rainstorm* characteristics *remains inconsistent, with limited research examining their combined impact. This study examines* this interplay  *in shaping recession responses, based on*  *291 sets of recession parameters obtained through the decorrelation process. The data originates from 19 subtropical mountainous* rivers [rivers do not display rainfall amounts. I made a suggestion on how to rephrase the sentence.] and covers events with  *a wide spectrum of rainfall amounts. Key findings indicate that the recession coefficient (a) increases while the exponent (b) decreases with the L/G ratio (the median of ratios between flow-path length and gradient), suggesting that longer and gentler hillslopes facilitate flow accumulation and aquifer connectivity, ultimately reducing nonlinearity.*

**Reply:** Revised as the suggestion.

*Additionally, in large catchments, the exponent (b)* rises [] *with increasing rainfall due to greater landscape heterogeneity, whereas in small catchments, it declines with rainfall, likely indicating* that the *catchment is* prone to saturated [this is not clear. The phrasing is incorrect and it is also not clear why a small catchment should be saturated.] *and thus reduced runoff heterogeneity.*

**Reply:** We rephrased this sentence: "Conversely, in small catchments, it declines with rainfall, indicating that these catchments have less landscape heterogeneity and thus reduced runoff heterogeneity."

*Our* findings  underscores *the necessity for further validation* of how L/G and drainage area regulate recession responses to varying rainfall levels *across diverse regions*

*drainage area regulate recession responses to varying rainfall levels, given the pivotal role of assessing recession responses in understanding regional recession patterns within ungauged catchments, particularly within the context of climate change.* [This last part is very convoluted: assessing recession responses in understanding regional recession patterns... is very confusing. Also, the sentence is much too long. I suggest to simplify or leave it out. I made suggestion above how to rephrase the sentence]*"* [L10-22]

**Reply:** Revised as the suggestion.

*"Specifically, flow path length ($l_{fp}$) is the route length from a cell to **the nearest** channel cell, flow path height ($h_{fp}$) is the elevation difference between the specific cell to **the nearest** channel cell, and flow path gradient ($g_{fp}$) is calculated as flow path height divided by flow path length."* [L84-86]

**Reply:** Revised as the suggestion.

*"In the hydrological context, $l_{fp}/g_{fp}$ represents the residence time of each flow path and the median of $l_{fp}/g_{fp}$ characterizes the flow paths within a catchment, while the median of $l_{fp}/g_{fp}$ reflects catchment-wide residence time."* [L90-92]

I don't understand this. You are both times describing the same ratio, but linking it with a "while" which mean you are talking about different things. Something is wrong here, the sentence does not work.

**Reply:** We apologize for the incorrect sentence. The sentence now reads: "In the hydrological context, $l_{fp}/g_{fp}$ represents the residence time of each flow path, while the median of $l_{fp}/g_{fp}$ reflects catchment-wide residence time.".

*"The weak correlation between recession nonlinearity and those variables might be explained by: First is the scale effect."* [L267]

This sentence does not work

**Reply:** We rephrased this sentence: "The weak correlation between recession nonlinearity and those variables may be attributed to two factors. First, there is the scale effect."

*"The fact that H is negatively correlated with the recession coefficient suggests that our [] groundwater flow paths possess greater depth and length, consequently leading to slower drainage rates."* [L283-285]

**Reply:** We removed the term "our".

*"While H is commonly believed to be positively correlated with the velocity of gravity-driven flow at a small spatial scale, the high heterogeneity in  geology or soil properties at a larger spatial scale (Karlsen et al., 2019) implies that a large H does not necessarily lead to a large recession*

*coefficient.*" [L285-287]

**Reply:** Revised as the suggestion.

(In table S3 in the supplementary material)

*7-day antecedent precipitation*,  *used to*  assess *the saturation status of the watershed before the rainstorm.*

**Reply:** Revised as the suggestion.

*Total precipitation*

**Reply:** Revised as the suggestion.

*Duration of precipitation*

**Reply:** Revised as the suggestion.

*Averaged precipitation intensity*

**Reply:** Revised as the suggestion.

*Total streamflow represents how much water is exported during a rainstorm* [this is total specific discharge as you provide it in mm.]

**Reply:** Revised as the suggestion.

*Rainfall period* [I think this is not the rainfall period, but the window over which you cumulate rainfall. It would be good to also mention this in the text (the manuscript). Why do you cut off at peak flow? Rainfall continuing after the peak would affect the shape of the recession.]

**Reply:** We mentioned this in the main text [L100-102]:

"The cumulative rainfall window was defined as the elapsed time from 6 h before the rising flow to the peak flow. We do not consider rainfall amount after peak flow because there is typically less rainfall occurring after the peak flow."

*Length of time between the start and end of the defined rainfall period* [does this also follow the definition below in the footnote? Or does this refer to the actual start and end of rainfall? The latter would make much more sense.].

**Reply:** We revised as "Length of the cumulative rainfall window."

*Sum of flow rates during the rainstorm.* [this is not correct.]

**Reply:** We revised as "Total discharge divided by the drainage area."

**Remarks from the preceding review file validation**

Checking your paper, I noticed that your tables contain coloured cells. Please note that this will not be possible in the final revised version of the paper due to HTML conversion of the paper. When revising the final version, you can use footnotes or italic/bold font. For now, the process will continue, but please note that the final version cannot be published by using coloured tables.

**Reply:** We have updated the table by using symbols (+, -, and x) to indicate positive, negative, and no correlation with the factors, in place of using colored cells in Table 1. As for Table 2, we replace the gray colored cells with bold fonts.